# EFFICIENT AND TRUSTWORTHY CAUSAL DISCOVERY WITH LATENT VARIABLES AND COMPLEX RELATIONS

**Xiu-Chuan Li    Tongliang Liu**[*]

Sydney AI Centre, University of Sydney

## ABSTRACT

Most traditional causal discovery methods assume that all task-relevant variables are observed, an assumption often violated in practice. Although some recent works allow the presence of latent variables, they typically assume the absence of certain special causal relations to ensure a degree of simplicity, which might also be invalid in real-world scenarios. This paper tackles a challenging and important setting where latent and observed variables are interconnected through complex causal relations. Under a pure children assumption ensuring that latent variables leave adequate footprints in observed variables, we develop novel theoretical results, leading to an efficient causal discovery algorithm which is the first one capable of handling the setting with both latent variables and complex relations within polynomial time. Our algorithm first sequentially identifies latent variables from leaves to roots and then sequentially infers causal relations from roots to leaves. Moreover, we prove trustworthiness of our algorithm, meaning that when the assumption is invalid, it can raise an error signal rather than draw an incorrect causal conclusion, thus preventing potential damage to downstream tasks. We demonstrate the efficacy of our algorithm through experiments. Our work significantly enhances efficiency and reliability of causal discovery in complex systems. Our code is available at: https://github.com/XiuchuanLi/ICLR2025-ETCD

## 1 INTRODUCTION

Causality is a fundamental notion in natural and social sciences, which plays a crucial role in explanation, prediction, decision making and control (Zhang et al., 2018). Particularly, causality has driven significant progress in machine learning (Yao et al., 2021; 2023; Huang et al., 2023; Hong et al., 2024; Lin et al., 2024; Sun et al., 2024). Uncovering causality through analysis of observational data, commonly known as causal discovery, has garnered significant attention. Most traditional causal discovery methods (Spirtes & Glymour, 1991; Chickering, 2002; Shimizu et al., 2006) assume that all task-relevant variables are ob-

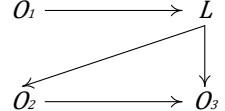

Figure 1: A causal structure violating all of the purity, measurement, and no-triangle assumptions.

served, but in practice, we often fail to collect and measure all of them in practice. Some previous works can reveal causal relations between observed variables in the presence of latent variables, but their results are not informative of the number or the causal relations of latent variables. By utilizing linear models, some recent works can display latent variables and their causal relations explicitly in their results. However, they often assume the absence of certain special causal relations to ensure a degree of simplicity, including the *measurement assumption* (Silva et al., 2006; Xie et al., 2022) positing the absence of edges from observed variables to latent ones, the *purity assumption* (Cai et al., 2019; Xie et al., 2020) positing the absence of edges between observed variables, and the *no-triangle assumption* (Huang et al., 2022; Dong et al., 2024) positing the absence of triangles formed by three mutually adjacent variables. Unfortunately, these assumptions fail in many real-world scenarios. Consider the causal structure shown as Fig. 1 where $L$ and $O$ respectively denote latent and observed variables. Clearly, $O_2 \rightarrow O_3$ violates the purity assumption, $O_1 \rightarrow L$ violates the measurement assumption, and $L, O_2, O_3$ violates the no-triangle assumption. This structure can be found in business contexts, where $O_1, L, O_2, O_3$ refer to advertisement spending, consumer interest, product

---

[*]Correspondence to Tongliang Liu (tongliang.liu@sydney.edu.au).

views, and product sales respectively. In this paper, given observational data generated by a linear non-Gaussian acyclic model (LiNGAM) with latent variables where latent and observed variables are interconnected through complex causal relations, that is, none of the above three assumptions is required, we aim to correctly identify the underlying complete causal structure, which is a directed acyclic graph (DAG) that explicitly represents both observed and latent variables along with their causal relations.

Adams et al. (2021) first investigate the challenging setting with both latent variables and complex relations. They develop a causal discovery algorithm under the assumption which is exactly sufficient and necessary for identifiability of LiNGAM with latent variables, but it requires the number of latent variables as prior knowledge and lacks robustness, hence is not advisable in practice. Subsequently, Jin et al. (2024) introduce a stronger assumption that latent variables have pure children, ensuring latent variables leave adequate footprints in observed variables. Under this assumption, they propose the first practical algorithm capable of handling this challenging setting, which recovers the causal graph in a bottom-up manner, progressing from leaves to roots. Unfortunately, its has exponential time complexity with respect to the number of variables, substantially limiting its applicability. In this paper, under a similar assumption also involving pure children, we propose a far more efficient algorithm with only cubic time complexity. Our algorithm follows a bottom-up then top-down pattern. In stage 1, it sequentially identifies latent variables through their pure children, progressing from leaves to roots. In stage 2, for variables not recognized as others' pure children in stage 1, it sequentially infers their causal relations, progressing from roots to leaves.

As mentioned above, we make a pure children assumption to enable a practical causal discovery algorithm. When this assumption holds, our algorithm can correctly identify the underlying complete causal structure. But when this assumption fails, the recovered causal graph may be incorrect, which is potentially harmful in practical applications. For instance, in financial markets, a plausible but incorrect causal conclusion might mislead investors to make poor investment choices and cause significant financial losses. To overcome this limitation, we additionally prove trustworthiness of our algorithm, meaning that it can raise an error signal rather than return an incorrect causal structure when the pure children assumption is invalid. Specifically, if this assumption is violated, we prove at the end of stage 1, there exists an unidentified latent variable or an identified latent variable whose recognized pure children are not actually its pure children. In stage 2, this hidden risk will be triggered, prompting the error signal. To the best of our knowledge, there is a lack of similar results in the literature of causal discovery with latent variables[1].

The major innovations of our work are summarized as follows.

- We investigate an understudied and challenging setting in causal discovery where latent and observed variables are interconnected through complex causal relations.
- Under a pure children assumption, we develop a series of novel theoretical results, leading to an efficient causal discovery algorithm, which can handle the setting with both latent variables and complex relations within polynomial time.
- We prove trustworthiness of our algorithm, meaning that when the pure children assumption is invalid, it can raise an error signal rather than return an incorrect result.

In summary, our work significantly enhances efficiency and reliability of causal discovery in complex systems. It may both inspire further research in causal discovery and benefit research in natural and social sciences. Due to space limit, we defer detailed discussion on related works to App. A.

## 2 PRELIMINARY

We focus on the linear non-Gaussian acyclic model (LiNGAM) with latent variables whose graph structure $\mathcal{G}_0$ is a DAG. Its vertex set is $\mathbf{V}_0 = \mathbf{L} \cup \mathbf{O}_0$ where $\mathbf{L}$ and $\mathbf{O}_0$ respectively denote the set of latent and observed variables. We augment $\mathcal{G}_0$ to $\mathcal{G}$ by creating two children for every $O \in \mathbf{O}_0$, each of which is $O$ plus an independent non-Gaussian noise. We denote the set of such created variables

---

[1]Many works employ similar pure children assumptions. While the vast majority of them limit their scope to the scenario where the assumption holds, Silva et al. (2006) systematically analyze the case where the assumption fails, rigorously characterizing the potential mistakes. Although this is an insightful result, it does not directly enable us to determine whether the recovered causal graph is correct. Instead, we prove that an error signal can be explicitly raised if the assumption fails.

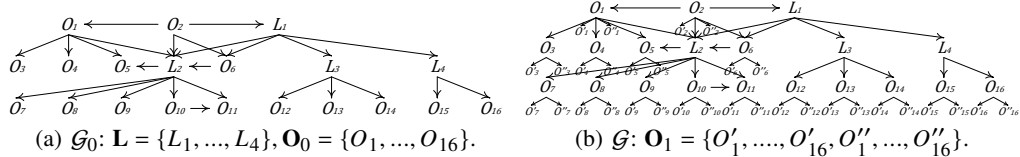

(a) $\mathcal{G}_0$: $\mathbf{L} = \{L_1, ..., L_4\}, \mathbf{O}_0 = \{O_1, ..., O_{16}\}$.  (b) $\mathcal{G}$: $\mathbf{O}_1 = \{O'_1, ...., O'_{16}, O''_1, ..., O''_{16}\}$.

Figure 2: An illustrative example of augmenting (a) $\mathcal{G}_0$ to (b) $\mathcal{G}$.

by $\mathbf{O}_1$[2] and let $\mathbf{O} = \mathbf{O}_0 \cup \mathbf{O}_1$, $\mathbf{V} = \mathbf{L} \cup \mathbf{O}$. Trivially, identifying $\mathcal{G}_0$ is equivalent to identifying $\mathcal{G}$. An example is shown as Fig. 2. Because of the linearity, each variable $V_i \in \mathbf{V}$ follows

$$V_i = \sum_{V_j \in \mathbf{V}} a_{ij} V_j + \epsilon_{V_i}, \tag{1}$$

where $\epsilon_{V_i}$ refers to an exogenous noise. All exogenous noises have non-Gaussian distributions and are independent of each other. $a_{ij} \neq 0$ iff $V_j$ is a parent of $V_i$. Eq. (1) can also be written as

$$V_i = \sum_{V_j \in \mathbf{V}} m_{ij} \epsilon_{V_j}. \tag{2}$$

where $M = (I - A)^{-1}$ with $M$ and $A$ being matrices composed of $m_{ij}$ and $a_{ij}$ respectively. By convention, we assume the distribution over $\mathbf{V}$ is both Markov and bottleneck-faithful (Adams et al., 2021) to $\mathcal{G}$. Given $V \in \mathbf{V}$, we denote its parents, children, neighbors, ancestors, and descendants by $\mathrm{Pa}(V), \mathrm{Ch}(V), \mathrm{Ne}(V), \mathrm{An}(V)$, and $\mathrm{De}(V)$. Particularly, a variable's ancestors/descendants do not include itself. We call a variable's ancestors/descendants plus itself its generalized ancestors/descendants, denoted by $\mathrm{GAn}(V)$ and $\mathrm{GDe}(V)$. We abbreviate $\bigcup_{V \in \mathbf{V}'} \mathrm{Pa}(V)$ to $\bigcup \mathrm{Pa}(\mathbf{V}')$.

## 3 EFFICIENT CAUSAL DISCOVERY

In this section, we develop a series of novel theoretical results under Asmp. 1, leading to an efficient causal discovery algorithm with only cubic time complexity.

**Definition 1.** *(Pure child) We say $V_2$ is a pure child of $V_1$, denoted by $V_2 \in \mathrm{PCh}(V_1)$, if (i) $\mathrm{Pa}(V_2) = \{V_1\}$ and (ii) $\forall V \in \mathrm{De}(V_2), |\mathrm{Pa}(V)| = 1$.*

**Example.** In Fig. 2(a), $\mathrm{PCh}^{\mathcal{G}_0}(L_2) = \{O_7, O_8, O_9\}$. $O_{11} \notin \mathrm{PCh}^{\mathcal{G}_0}(L_2)$ as $\mathrm{Pa}^{\mathcal{G}_0}(O_{11}) = \{L_2, O_{10}\} \neq \{L_2\}$. $O_{10} \notin \mathrm{PCh}^{\mathcal{G}_0}(L_2)$ as $O_{11} \in \mathrm{De}^{\mathcal{G}_0}(O_{10})$ but $|\mathrm{Pa}^{\mathcal{G}_0}(O_{11})| = 2$.

**Remark.** This concept has been widely used in related works, but there is no consensus on its exact definition. For instance, a pure child in Silva et al. (2006); Kummerfeld & Ramsey (2016); Xie et al. (2020); Li et al. (2024) must be an observed variable with no child, which is more restrictive than ours. In Jin et al. (2024), a variable's pure child can have a descendant with multiple parents provided that these parents do not include the variable itself, which is less restrictive than ours.

**Assumption 1.** $\forall L \in \mathbf{L}, |\mathrm{PCh}^{\mathcal{G}_0}(L)| \geq 2$ *and* $|\mathrm{Ne}^{\mathcal{G}_0}(L)| \geq 3$.

**Example.** The graph in Fig. 2(a) satisfies this assumption. For instance, $\mathrm{PCh}^{\mathcal{G}_0}(L_1) = \{L_3, L_4\}$ and $\mathrm{Ne}^{\mathcal{G}_0}(L_1) = \{L_2, L_3, L_4, O_2, O_6\}$.

**Remark.** This assumption allows each latent variable to leave footprints in observed variables adequate for identification. It naturally holds in the scenarios with many directly measured variables such as topic model (Arora et al., 2013). Similar assumptions involving pure children were also made in many previous works (Silva et al., 2006; Kummerfeld & Ramsey, 2016; Cai et al., 2019; Xie et al., 2020; Zeng et al., 2021; Chen et al., 2022; Xie et al., 2022; Huang et al., 2022; Chen et al., 2023; Dong et al., 2024; Jin et al., 2024; Xie et al., 2024).

---

[2]While the values of observed variables are directly accessible for causal discovery, the causal relations of latent variables can only be inferred indirectly through their pure children (Def. 1). By introducing $\mathbf{O}_1$ to create pure children for each observed variable, we can uniformly handle both latent and observed variables through analyzing their pure children, thereby eliminating the need to repeatedly distinguish between treatments of latent and observed variables, which keeps our core methodology clear.

Figure 3: Left: Initial $\mathcal{H}_1$, where $\mathbf{V}_a$ and $\mathbf{V}_b$ are marked in red and black respectively; Right: Initial $\mathcal{H}_2$, which is exactly $\mathcal{G}_0$, where $\mathbf{V}_a$ and $\mathbf{V}_c$ are marked in red and black respectively.

### 3.1 STAGE 1: IDENTIFYING LATENT VARIABLES

**§ High-level Overview.** In stage 1, we identify latent variables through their pure children. Concretely, we initialize an active set $\mathbf{V}_a$ as $\mathbf{O}_0$. First, we locate some variables that serve as pure children of others from the active set (Thms. 1 and 2). Second, we identify these pure children's parents (Thm. 3). Third, we update the active set by replacing these pure children with their parents. Repeating this process until the active set cannot be updated, all latent variables can be identified finally (Thm. 4). Clearly, we follow a bottom-up pattern in stage 1, progressing from leaves to roots.

**§ Initialization.** During stage 1, we maintain two sets of variables $\mathbf{V}_a, \mathbf{V}_b$ and a graph $\mathcal{H}_1$ over $\mathbf{V}_a \cup \mathbf{V}_b$. Initially, we let $\mathbf{V}_a, \mathbf{V}_b$ be $\mathbf{O}_0, \mathbf{O}_1$ respectively and let $V_i \in \mathrm{Pa}^{\mathcal{H}_1}(V_j)$ iff $V_i \in \mathbf{O}_0, V_j \in \mathbf{O}_1$, and $V_i \in \mathrm{Pa}^{\mathcal{G}}(V_j)$. For $\mathcal{G}$ shown as Fig. 2(b), the initial $\mathcal{H}_1$ is displayed on the left of Fig. 3. Subsequently, $\mathbf{V}_a, \mathbf{V}_b$, and $\mathcal{H}_1$ will be updated following the rules described later in § Update. Intuitively, $\mathbf{V}_a$ consists of identified variables whose causal relations (i.e., both incoming and outgoing edges of the variable in the underlying causal graph) are not fully identified, $\mathbf{V}_b$ consists of identified variables whose causal relations are fully identified, and $\mathcal{H}_1$ consists of all identified causal relations. Clearly, Cond. 1 holds initially, and we will show later Cond. 1 always holds throughout stage 1.

**Condition 1.** *(1)* $\forall V \in \mathbf{V}_b$, $|\mathrm{Pa}^{\mathcal{H}_1}(V)| = 1$ *and* $\mathrm{Ch}^{\mathcal{H}_1}(V) = \mathrm{PCh}^{\mathcal{G}}(V)$*; (2)* $\forall V \in \mathbf{V}_a$, $\mathrm{Pa}^{\mathcal{H}_1}(V) = \emptyset$, $|\mathrm{Ch}^{\mathcal{H}_1}(V)| \geq 2$*, and* $\mathrm{Ch}^{\mathcal{H}_1}(V) \subseteq \mathrm{PCh}^{\mathcal{G}}(V)$.

For ease of exposition, we denote $\mathbf{V}\backslash(\mathbf{V}_a \cup \mathbf{V}_b)$ by $\mathbf{V}_c$ and the induced subgraph of $\mathcal{G}$ over $\mathbf{V}_a \cup \mathbf{V}_c$ by $\mathcal{H}_2$. For $\mathcal{G}$ shown as Fig. 2(b), the initial $\mathbf{V}_c$ is exactly $\mathbf{L}$ and the initial $\mathcal{H}_2$ is displayed on the right of Fig. 3, which is exactly $\mathcal{G}_0$. Intuitively, while $\mathbf{V}_a \cup \mathbf{V}_b$ consists of all identified variables, $\mathbf{V}_c$ consists of all unidentified variables. While $\mathcal{H}_1$ consists of all identified causal relations, $\mathcal{H}_2$ consists of all unidentified causal relations.

**§ Locating Pure Children.** Ideally, we want to locate pure children in a single step, but this is infeasible because of the existence of complex causal relations. Instead, we first locate identifiable pairs (Def. 2) from $\mathbf{V}_a$ (Thm. 1) and then locate pure children from these identifiable pairs (Thm. 2).

**Definition 2.** *(Identifiable pair, IP) We say* $\{V_1, V_2\} \subseteq \mathbf{V}_a$ *is an IP, denoted by* $\{V_1, V_2\} \in \mathbb{S}$*, if*

*(1)* $\mathrm{Pa}^{\mathcal{H}_2}(V_2) = \{V_1\}$*,* $\mathrm{Ch}^{\mathcal{H}_2}(V_2) = \emptyset$*, and* $\mathrm{Ne}^{\mathcal{H}_2}(V_1)\backslash\{V_2\} \neq \emptyset$*. We denote this by* $\{V_1, V_2\} \in \mathbb{S}_1$*; or*

*(2)* $\exists V_0 \in \mathbf{V}_a \cup \mathbf{V}_c\backslash\{V_1, V_2\}$ *s.t.* $\mathrm{Pa}^{\mathcal{H}_2}(V_1) = \mathrm{Pa}^{\mathcal{H}_2}(V_2) = \{V_0\}$*,* $\mathrm{Ch}^{\mathcal{H}_2}(V_1) = \mathrm{Ch}^{\mathcal{H}_2}(V_2) = \emptyset$*, and* $V_0 \in \mathbf{V}_a$ *or* $\mathrm{Ne}^{\mathcal{H}_2}(V_0)\backslash\{V_1, V_2\} \neq \emptyset$*. We denote this by* $\{V_1, V_2\} \in \mathbb{S}_2$*; or*

*(3)* $\exists V_0 \in \mathbf{V}_a \cup \mathbf{V}_c\backslash\{V_1, V_2\}$ *s.t.* $\mathrm{Pa}^{\mathcal{H}_2}(V_1) = \{V_0\}$*,* $\mathrm{Ch}^{\mathcal{H}_2}(V_1) = \{V_2\}$*,* $\mathrm{Pa}^{\mathcal{H}_2}(V_2) = \{V_0, V_1\}$*, and* $\mathrm{Ch}^{\mathcal{H}_2}(V_2) = \emptyset$*. We denote this by* $\{V_1, V_2\} \in \mathbb{S}_3$.

**Example.** In the right sub-figure of Fig. 3, $\mathbb{S}_1 = \{\{O_1, O_3\}, \{O_1, O_4\}\}$, $\mathbb{S}_2 = \{\{O_3, O_4\}, \{O_7, O_8\}, \{O_7, O_9\}, \{O_8, O_9\}\{O_{12}, O_{13}\}, \{O_{12}, O_{14}\}, \{O_{13}, O_{14}\}, \{O_{15}, O_{16}\}\}$, $\mathbb{S}_3 = \{\{O_{10}, O_{11}\}\}$.

**Intuition.** Although $\mathcal{H}_2$ is unknown, we will show later that identifiable pairs can still be located from $\mathbf{V}_a$ via statistical analysis (Thm. 1), this is what "identifiable" means. The connection between identifiable pairs and pure children are as follows:

(1) If $\{V_1, V_2\} \in \mathbb{S}_1$, $V_2$ is $V_1$'s pure child or $V_1$ is $V_2$'s pure child.

(2) If $\{V_1, V_2\} \in \mathbb{S}_2$, $V_1$ and $V_2$ are both pure children of $V_0$.

(3) If $\{V_1, V_2\} \in \mathbb{S}_3$, neither $V_1$ nor $V_2$ is a pure child of any other variable.

**Definition 3.** *(Pseudo-residual (Cai et al., 2019)) Given three variables* $V_1, V_2, V_3$ *s.t.* $\mathrm{Cov}(V_2, V_3) \neq 0$*, the pseudo-residual of* $V_1, V_2$ *relative to* $V_3$ *is defined as*

$$R(V_1, V_2 | V_3) = V_1 - \frac{\mathrm{Cov}(V_1, V_3)}{\mathrm{Cov}(V_2, V_3)} V_2. \tag{3}$$

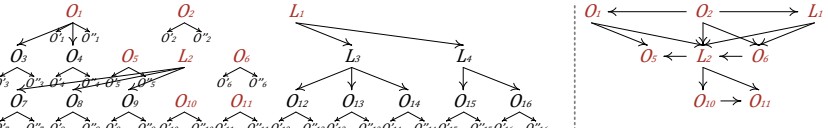

Figure 4: Left: $\mathcal{H}_1$ after the first update, where $\mathbf{V}_a$ and $\mathbf{V}_b$ are marked in red and black respectively; Right: $\mathcal{H}_2$ after the first iteration, where $\mathbf{V}_a$ and $\mathbf{V}_c$ are marked in red and black respectively.

Figure 5: Left: $\mathcal{H}_1$ at the end of stage 1, where $\mathbf{V}_a$ and $\mathbf{V}_b$ are marked in red and black respectively; Right: $\mathcal{H}_2$ at the end of stage 1, where $\mathbf{V}_a$ and $\mathbf{V}_c$ are marked in red and black respectively.

**Intuition.** Pseudo-residual is a simple variant of the conventional residual. The former reduces to the latter when $V_2 = V_3$. Before Cai et al. (2019), similar concepts have already been used by earlier works (Drton & Richardson, 2004; Chen et al., 2017).

**Theorem 1.** $\forall \{V_i, V_j\} \subseteq \mathbf{V}_a$, $\{V_i, V_j\} \in \mathbb{S}$ iff there exists $V_k \in \mathbf{V}_a \backslash \{V_i, V_j\}$ s.t. $\text{Cov}(V_i, V_j)$ $\text{Cov}(V_i, V_k) \text{Cov}(V_j, V_k) \neq 0$ and for each such $V_k$, $\text{R}(V_i, V_j | V_k) \perp\!\!\!\perp \mathbf{V}_a \backslash \{V_i, V_j\}$.

**Remark.** This theorem provides a method for locating identifiable pairs from $\mathbf{V}_a$ via statistical analysis. In principle, we need to do independence test for each $V_k$. But in fact, it is sufficient to only consider any single $V_k$ (see Prop. 1 in App. C.2.1), reducing the time complexity.

**Definition 4.** *(Quintuple constraint)* We say $(V_1, V_2, V_3, V_4, V_5)$ satisfies the quintuple constraint if there exist $\alpha, \beta$ s.t. $V_1 + \alpha V_3 + \beta V_4 \perp\!\!\!\perp V_2$ and $\text{Cov}(V_1 + \alpha V_3 + \beta V_4, V_5) = 0$.

**Theorem 2.** $\forall \{V_i, V_j\} \in \mathbb{S}$, let $\{V_{i_1}, V_{i_2}\} \subseteq \text{Ch}^{\mathcal{H}_1}(V_i)$.

(1) $\text{R}(V_{i_1}, V_j | V_{i_2}) \perp\!\!\!\perp V_{i_2}$ iff $\{V_i, V_j\} \in \mathbb{S}_1$ and $V_i \in \text{Pa}^{\mathcal{G}}(V_j)$.

(2) Suppose $\{V_i, V_j\} \notin \mathbb{S}_1$, $\exists \{V_{i'}, V_{j'}\} \in \mathbb{S} \backslash \{\{V_i, V_j\}\}$ s.t. $\{V_{i'}, V_{j'}\} \cap \{V_i, V_j\} \neq \emptyset$ only if (but not if) $\{V_i, V_j\} \in \mathbb{S}_2$.

(3) Suppose $\{V_i, V_j\} \notin \mathbb{S}_1$, $\exists \{V_k, V_l\} \subseteq \bigcup \text{Ch}^{\mathcal{H}_1}(\mathbf{V}_a \backslash \{V_i, V_j\})$ s.t. $(V_{i_1}, V_{i_2}, V_j, V_k, V_l)$ satisfies the quintuple constraint iff $\{V_i, V_j\} \in \mathbb{S}_3$ and $V_i \in \text{Pa}^{\mathcal{G}}(V_j)$.

**Remark.** This theorem provides a method to divide $\mathbb{S}$ into $\mathbb{S}_1, \mathbb{S}_2, \mathbb{S}_3$ via statistical analysis, from which we can easily locate pure children. Given $\{V_i, V_j\} \in \mathbb{S}$, we first check whether $\{V_i, V_j\} \in \mathbb{S}_1$ based on (1). If it is and $V_i \in \text{Pa}^{\mathcal{G}}(V_j)$, then $V_j$ is $V_i$'s pure child. Otherwise, we further check whether $\{V_i, V_j\} \in \mathbb{S}_2$ based on (2,3). If it is, then both $V_i$ and $V_j$ are pure children of another unknown variable; otherwise, they are not.

**§ Identifying Pure Children's Parents.** As mentioned in Rem. of Thm. 2, for any pair in $\mathbb{S}_2$, its parent is unknown. To avoid duplicate identification of latent variables, we need to check whether its parent is in $\mathbf{V}_a$ or $\mathbf{V}_c$ (Thm. 3(1)) and whether it shares the parent with other pairs in $\mathbb{S}_2$ (Thm. 3(2)).

**Theorem 3.** *(1)* $\forall \{V_i, V_j\} \in \mathbb{S}_2$, $\bigcap \text{Pa}^{\mathcal{G}}(\{V_i, V_j\}) \subseteq \mathbf{V}_a$ iff $\exists \{V_{i'}, V_{j'}\} \in \mathbb{S}_1$ s.t. $\{V_i, V_j\} \cap \{V_{i'}, V_{j'}\} \neq \emptyset$. *(2)* $\forall \{\{V_i, V_j\}, \{V_{i'}, V_{j'}\}\} \subseteq \mathbb{S}_2$, $\bigcap \text{Pa}^{\mathcal{G}}(\{V_i, V_j\}) = \bigcap \text{Pa}^{\mathcal{G}}(\{V_{i'}, V_{j'}\})$ iff $\exists \{V_{i''}, V_{j''}\} \in \mathbb{S}_2$ s.t. $\{V_i, V_j\} \cap \{V_{i''}, V_{j''}\} \neq \emptyset$ and $\{V_{i'}, V_{j'}\} \cap \{V_{i''}, V_{j''}\} \neq \emptyset$.

**Example.** In Fig. 3, an example of (1) is that $\{V_i, V_j\} = \{O_3, O_4\}$ and $\{V_{i'}, V_{j'}\} = \{O_1, O_3\}$; an example of (2) is that $\{V_i, V_j\} = \{O_7, O_8\}$ and $\{V_{i'}, V_{j'}\} = \{V_{i''}, V_{j''}\} = \{O_7, O_9\}$.

**§ Update.** For pairs in $\mathbb{S}_1$, we move the children in them from $\mathbf{V}_a$ to $\mathbf{V}_b$ and add edges from parents to children into $\mathcal{H}_1$. For pairs in $\mathbb{S}_2$ whose parents are in $\mathbf{V}_c$ rather than $\mathbf{V}_a$, we merge those pairs that share a common parent into a set. In particular, if a pair does not share a common parent with any other, it is itself a set. For each set, we move it from $\mathbf{V}_a$ to $\mathbf{V}_b$, create and add a new latent variable into $\mathbf{V}_a$ as the its parent, and add edges from its parent to its all variables into $\mathcal{H}_1$. For initialization shown as Fig. 3, the updated result is shown as Fig. 4.

---

**Algorithm 1:** Stage 1: Identifying latent variables (overview)

---
**Input:** Observed variables $\mathbf{O}_0$ and $\mathbf{O}_1$
**Output:** $\mathbf{V}_a$, $\mathbf{V}_b$, and $\mathcal{H}_1$

1   Initializing $\mathbf{V}_a$, $\mathbf{V}_b$, and $\mathcal{H}_1$ following § Initialization in Sec. 3.1.
2   **while** *the current $\mathbf{V}_a$ is not identical to the previous $\mathbf{V}_a$* **do**
3     Locating identifiable pairs $\mathbb{S}$ from $\mathbf{V}_a$ based on Thm. 1.
4     Locating pure children from identifiable pairs $\mathbb{S}$ based on Thm. 2.
5     Identifying pure children's parents based on Thm. 3.
6     Updating $\mathbf{V}_a$, $\mathbf{V}_b$, and $\mathcal{H}_1$ following § Update in Sec. 3.1.
7   **end**

---

§ **Repeating This Process.** After update, Cond. 1 is still valid (see Prop. 2 in App. C.2.1). By induction, we can repeat the above process[3] until $\mathbf{V}_a$ cannot be updated. Finally, all latent variables can be identified (Thm. 4). For $\mathcal{G}$ shown as Fig. 2(b), the final result of stage 1 is shown as Fig. 5. An overview of our algorithm in stage 1 is shown as Alg. 1 while a detailed version is deferred to Alg. 3 in App. E. It has $O(R|\mathbf{O}_0|^3)$ complexity where $R$ is the number of iterations.

**Theorem 4.** *If $\mathbb{S}_1 \cup \mathbb{S}_2 = \emptyset$, $\mathbf{V}_c = \emptyset$.*

**Remark.** If both $\mathbb{S}_1$ and $\mathbb{S}_2$ are empty, then $\mathbf{V}_a$ cannot be updated, that is, stage 1 comes to an end. At this moment, $\mathbf{V}_c = \emptyset$, which means that all latent variables are identified.

### 3.2 STAGE 2: INFERRING CAUSAL RELATIONS

§ **High-level Overview.** For variables recognized as others' pure children in stage 1 (i.e., $\mathbf{V}_b$ at the end of stage 1), their causal relations are fully identified in $\mathcal{H}_1$. The aim of stage 2 is to infer those unidentified causal relations, i.e., to recover $\mathcal{H}_2$ among $\mathbf{V}_a \cup \mathbf{V}_c$ at the end of stage 1, when $\mathbf{V}_c = \emptyset$ based on Thm. 4. We initialize an active set $\mathbf{U}_a$ as $\mathbf{V}_a$ at the end of stage 1. First, we identify a root variable in the active set (Thm. 5). Second, we estimate the root variable's effects on others (Thm. 6). Third, we remove the root variable from the active set and also remove its effects on others. Repeating this process until there is no variable in the active set, we can estimate all causal effects and then recover $\mathcal{H}_2$. Finally, we can obtain $\mathcal{G}$ by combining $\mathcal{H}_1$ with $\mathcal{H}_2$. Clearly, we follow a top-down pattern in stage 2, progressing from roots to leaves.

§ **Initialization.** We initialize $\mathbf{U}_a$ as $\mathbf{V}_a$ at the end of stage 1. For the sake of uniform processing, we view observed variables in $\mathbf{U}_a$ as latent variables, and assign each $V_i \in \mathbf{U}_a$ with two observed surrogates $X_{2i-1}$ and $X_{2i}$, which can be any variable in $\text{GDe}^{\mathcal{H}_1}(V_{i_1}) \cap \mathbf{O}$ and $\text{GDe}^{\mathcal{H}_1}(V_{i_2}) \cap \mathbf{O}$ where $\{V_{i_1}, V_{i_2}\} \subseteq \text{Ch}^{\mathcal{H}_1}(V_i)$. Taking $\mathcal{H}_1$ shown on the left of Fig. 5 as an example, two observed surrogates of $L_1$ can be $O_{12}$ and $O_{15}$ while two observed surrogates of $O_1$ can be $O'_1$ and $O''_1$. Clearly, Cond. 2 holds initially ($e'_{X_j} = 0$ for any $j$). We will show later Cond. 2 always holds throughout stage 2.

**Condition 2.** *(1) $\forall V \in \mathbf{U}_a$, $\text{De}^{\mathcal{H}_2}(V) \subseteq \mathbf{U}_a$. (2) $\forall V_i \in \mathbf{U}_a$, $X_{2i-1}$, $X_{2i}$ can be written as*

$$X_{2i-1} = c_{2i-1} \sum_{V_j \in \mathbf{U}_a} m_{ij}\epsilon_{V_j} + e_{X_{2i-1}} + e'_{X_{2i-1}}, \quad X_{2i} = c_{2i} \sum_{V_j \in \mathbf{U}_a} m_{ij}\epsilon_{V_j} + e_{X_{2i}} + e'_{X_{2i}}, \quad (4)$$

*where $\epsilon_{V_1}, ..., \epsilon_{V_n}, e_{X_1}, ..., e_{X_{2n}}, (e'_{X_1}, e'_{X_3}, ..., e'_{X_{2n-1}}), (e'_{X_2}, e'_{X_4}, ..., e'_{X_{2n}})$ are mutually independent. Without loss of generality, we assume each $c_{2i-1}$ is positive and each $\epsilon_{V_j}$ has variance 1.*

§ **Identifying a Root Variable.** This can be accomplished based on Thm. 5.

**Theorem 5.** *$\forall V_i \in \mathbf{U}_a$, $\text{An}^{\mathcal{H}_2}(V_i) \cap \mathbf{U}_a = \emptyset$ iff $\forall V_j \in \mathbf{U}_a \backslash \{V_i\}$, $\text{R}(X_{2j-1}, X_{2i-1}|X_{2i}) \perp\!\!\!\perp X_{2i}$.*

§ **Estimating the Root Variable's Effects.** This can be accomplished based on Thm. 6.

---

[3]Readers may have such a question: how to test independence/correlation involving latent variables. For any latent variable $L \in \mathbf{V}_a \cup \mathbf{V}_b$, let $O$ be its any observed descendant in $\mathcal{H}_1$. With the fact implied by Cond. 1 that $O$ can be expressed as scaled $L$ plus a noise independent of all variables except for $L$'s descendants in $\mathcal{H}_1$ (this can also be implied by Cond. 3 in Sec. 4.1), any independence/correlation in our theoretical results involving $L$ holds iff it holds when $L$ is replaced with $O$. Therefore, when we need to test independence/correlation involving a latent variable, we can directly replace it with its any observed descendant in $\mathcal{H}_1$.

---

**Algorithm 2:** Stage 2: Inferring causal relations (overview)

---

**Input:** $\mathbf{V}_a$, $\mathbf{V}_b$, and $\mathcal{H}_1$ output by Alg. 1
**Output:** a complete causal structure $\mathcal{G}$

1 Initialize $\mathbf{U}_a$ following § Initialization in Sec. 3.2.
2 **while** $\mathbf{U}_a \neq \emptyset$ **do**
3      Identifying a root variable $V_i \in \mathbf{U}_a$ based on Thm. 5.
4      Estimating $V_i$'s effects on other variables based on Thm. 6.
5      Removing $V_i$ from $\mathbf{U}_a$ and updating $X_{2j-1}, X_{2j}$ for each $V_j \in \mathbf{U}_a$ following Eq. (6).
6 **end**
7 $\mathcal{G} := \mathcal{H}_1 \cup \mathcal{H}_2$ where $\mathcal{H}_2$ is recovered from the estimated effects.

---

**Theorem 6.** *If $V_i$ satisfies Thm. 5, then* $\mathrm{Cov}(X_{2i-1}, X_{2i}) = c_{2i-1}c_{2i}$ *and* $\forall V_j \in \mathbf{U}_a \backslash \{V_i\}$,

$$\mathrm{sgn}(m_{ji}) = \mathrm{sgn}(\frac{\mathrm{Cov}(X_{2i-1}, X_{2j})}{\mathrm{Cov}(X_{2i-1}, X_{2j})}), \quad \mathrm{Cov}(X_{2i-1}, X_{2j})\mathrm{Cov}(X_{2i}, X_{2j-1}) = c_{2i-1}c_{2i}c_{2j-1}c_{2j}m_{ji}^2. \quad (5)$$

*Besides,* $\forall V_k \in \mathbf{V}_a \backslash \mathbf{U}_a, m_{ki} = 0$.

> **Remark.** At the current iteration, $m_{ji}$ cannot be determined since $c_{2j-1}c_{2j}$ is still unknown. Later, when $V_j$ becomes the root variable, $c_{2j-1}c_{2j}$ is known and $m_{ji}$ can be determined.

**§ Removal.** We remove the identified root variable $V_i$ from $\mathbf{U}_a$ and eliminate its effects on $X_{2j-1}, X_{2j}$ following Eq. (6) for each $V_j \in \mathbf{U}_a$.

$$X_{2j-1} := \mathrm{R}(X_{2j-1}, X_{2i-1}|X_{2i}), \quad X_{2j} := X_{2j}. \quad (6)$$

**§ Repeating This Process.** After removal, Cond. 2 is still valid (see Prop. 3 in App. C.2.2). By induction, we can repeat the above process until there is no variable in $\mathbf{U}_a$. Finally, all causal effects can be estimated, from which $\mathcal{H}_2$ can be recovered following Eqs. (1) and (2). Combining $\mathcal{H}_1$ with $\mathcal{H}_2$, we can obtain $\mathcal{G}$. An overview of our algorithm in stage 2 is shown as Alg. 2 while a detailed version is deferred to Alg. 4 in App. E. It has $O(|\mathbf{V}_a|^3)$ time complexity.

### 3.3 SUMMARY

**Theorem 7.** *Suppose the observed variables are generated by a LiNGAM with latent variables satisfying the bottleneck-faithfulness assumption and Asmp. 1, in the limit of infinite data, our algorithm correctly identifies the underlying complete causal structure.*

## 4 TRUSTWORTHY CAUSAL DISCOVERY

In this section, we prove that our algorithm is trustworthy in the sense that it can raise an error signal rather than return an incorrect result if Asmp. 1 is invalid. This is quite challenging since we need to precisely characterize the behavior of our algorithm when Asmp. 1 is violated, that is, we have to carefully examine, modify, and re-prove all theoretical results in Sec. 3 in the case without Asmp. 1.

**Definition 5.** *(Paired pseudo-pure children) We say $\{V_2, V_3\}$ is a pair of pseudo-pure children of $V_1$, denoted by $\{V_2, V_3\} \in \mathrm{P}^3\mathrm{Ch}(V_1)$, if (i) $\bigcap \mathrm{Pa}(\{V_2, V_3\}) = \{V_1\}$, (ii) $\bigcup \mathrm{Pa}(\{V_2, V_3\}) \backslash \{V_1, V_2, V_3\} = \emptyset$, (iii) $V_2 \in \mathrm{Ne}(V_3)$, and (iv) $\forall V \in \bigcup \mathrm{De}(\{V_2, V_3\}) \backslash \{V_2, V_3\}, |\mathrm{Pa}(V)| = 1$.*

> **Example.** In Fig. 2(a), $\mathrm{P}^3\mathrm{Ch}(L_2) = \{\{O_{10}, O_{11}\}\}$. In Fig. 6(a), $\{L, V_2\} \notin \mathrm{P}^3\mathrm{Ch}(V_1)$ since $V_4 \in \bigcup \mathrm{De}(\{L, V_2\}) \backslash \{L, V_2\}$ but $|\mathrm{Pa}(V_4)| = 2$.

> **Intuition.** With the edge between $V_2$ and $V_3$ removed, they both become pure children of $V_1$.

**Definition 6.** *(Pathological variable, PV) Given a latent variable L, we say a L is a type-I PV (I-PV) if $\mathrm{Pa}(L) = \{V_1\}$ and $\mathrm{Ch}(L) = \{V_2, V_3, V_4\}$ where $\mathrm{Pa}(V_2) = \{V_1, L\}$, $|\mathrm{Pa}(V)| = 1$ for each $V \in \mathrm{De}(V_2)$, and $\mathrm{P}^3\mathrm{Ch}(L) = \{\{V_3, V_4\}\}$. We say L is a type-II PV (II-PV) if $\mathrm{Pa}(L) = \emptyset$, $\mathrm{Ch}(L) = \{V_1, V_2, V_3, V_4\}$, and $\mathrm{P}^3\mathrm{Ch}(L) = \{\{V_1, V_2\}, \{V_3, V_4\}\}$. L is a PV if it is either a I-PV or a II-PV.*

**Remark.** If a I-PV shown as Fig. 6(a) exists in $\mathcal{G}_0$, running our algorithm, there might be $V_1 \in \mathbf{V}_a$ and $\mathrm{Ch}^{\mathcal{H}_1}(V_1) = \{L, V_2\}$ sometime; if a II-PV shown as Fig. 6(b) exists in $\mathcal{G}_0$, there might be $L \in \mathbf{V}_a$ and $\mathrm{Ch}^{\mathcal{H}_1}(L) = \{V_1, V_2\}$ sometime. In both cases, neither $\mathrm{Ch}^{\mathcal{H}_1}(L) \subseteq \mathrm{PCh}^{\mathcal{G}}(L)$ nor $\{\mathrm{Ch}^{\mathcal{H}_1}(L)\} = \mathrm{P}^3\mathrm{Ch}^{\mathcal{G}}(L)$.

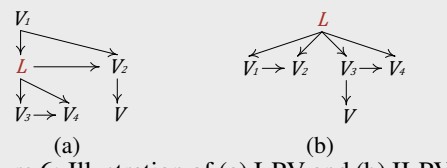

(a)           (b)

Figure 6: Illustration of (a) I-PV and (b) II-PV.

A pathological variable must satisfy many restrictive conditions, including but not limited to (1) it is a latent variable; (2) it has just the right number of parents and children; (3) its each descendant has the just right number of parents.

**Assumption 2.** *(1)* $\forall L \in \mathbf{L}, V \in \mathbf{V}_0 \backslash \{L\}$, $\mathrm{Ch}^{\mathcal{G}_0}(L) \not\subseteq \mathrm{Ch}^{\mathcal{G}_0}(V) \cup \{V\}$. *(2)* $\forall V \in \mathbf{V}_0$, $\mathrm{Ch}^{\mathcal{G}_0}(V)$ *is the unique minimal bottleneck (see Def. 7 in App. C) from* $\mathrm{Ch}^{\mathcal{G}_0}(V)$ *to* $\mathbf{O}_0$. *(3)* $\forall L \in \mathbf{L}$, *L is not a PV.*

**Remark.** It is infeasible to achieve trustworthiness without any assumptions. For instance, suppose $\mathcal{G}_0$ is shown as Fig. 7(a), then the observational distribution (i.e. $p(O_1, O_2, O_3)$) can also be explained by the causal graph shown as Fig. 7(b) that satisfies Asmp. 1, so Fig. 7(b) will be returned as the result and no error signal will be raised. To avoid such cases, we blend (1) and (2) from Adams et al. (2021) which are necessary

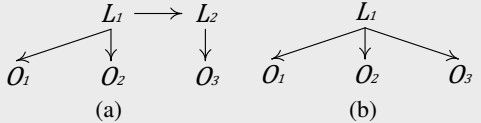

(a)           (b)

Figure 7: The observational distribution generated by (a) can also be explained by (b).

for identifiability of LiNGAM with latent variables. Besides, (3) is a technical assumption that significantly eases the readability and accessibility of the proof as it ensures that for every $V \in \mathbf{V}_a$, either $\mathrm{Ch}^{\mathcal{H}_1}(V) \subseteq \mathrm{PCh}^{\mathcal{G}}(V)$ or $\{\mathrm{Ch}^{\mathcal{H}_1}(V)\} = \mathrm{P}^3\mathrm{Ch}^{\mathcal{G}}(V)$. Considering that (3) is rather weak since a PV must satisfy many restrictive conditions, it does not seriously damage the generalizability of our results.

## 4.1 STAGE 1: IDENTIFYING LATENT VARIABLES

**§ High-level Overview.** We provide a variant of each theoretical result in Sec. 3.1. At the end of Sec. 3.1, all latent variables are identified and each variable's recognized children are all its actual pure children. But at the end of this section, there is an unidentified latent variable or an identified latent variable whose recognized pure children are not actually its pure children. This potential risk will be triggered in stage 2 such that an error signal can be raised.

**Condition 3.** *(1)* $\forall V \in \mathbf{V}_b$, $|\mathrm{Pa}^{\mathcal{H}_1}(V)| = 1$ *and* $\mathrm{Ch}^{\mathcal{H}_1}(V) = \mathrm{PCh}^{\mathcal{G}}(V)$; *(2)* $\forall V \in \mathbf{V}_a$, $\mathrm{Pa}^{\mathcal{H}_1}(V) = \emptyset$ *and* $|\mathrm{Ch}^{\mathcal{H}_1}(V)| \geq 2$. *If* $\mathrm{Ch}^{\mathcal{H}_1}(V) \not\subseteq \mathrm{PCh}^{\mathcal{G}}(V)$, *then* $\{\mathrm{Ch}^{\mathcal{H}_1}(V)\} = \mathrm{P}^3\mathrm{Ch}^{\mathcal{G}}(V)$ *and V satisfies some other conditions (more details are deferred to App. C.3.1).*

**Remark.** It is a variant of Cond. 1. Note that $\{\mathrm{Ch}^{\mathcal{H}_1}(V)\} = \mathrm{P}^3\mathrm{Ch}^{\mathcal{G}}(V)$ implies that $|\mathrm{Ch}^{\mathcal{H}_1}(V)| = 2$.

**Theorem 8.** $\forall \{V_i, V_j\} \subseteq \mathbf{V}_a$, $\{V_i, V_j\} \in \mathbb{S}$ *iff there exists* $V_k \in \mathbf{V}_a \backslash \{V_i, V_j\}$ *s.t.* $\mathrm{Cov}(V_i, V_j)$ $\mathrm{Cov}(V_i, V_k)\mathrm{Cov}(V_j, V_k) \neq 0$ *and for each such* $V_k$, $\mathrm{R}(V_i, V_j | V_k) \perp\!\!\!\perp \mathbf{V}_a \backslash \{V_i, V_j\}$.

**Remark.** It is identical to Thm. 1, so the identifiable pairs can still be located from observed variables correctly and exhaustively.

**Theorem 9.** $\forall \{V_i, V_j\} \in \mathbb{S}$, *let* $\{V_{i_1}, V_{i_2}\} \subseteq \mathrm{Ch}^{\mathcal{H}_1}(V_i)$.

*(1)* $\mathrm{R}(V_{i_1}, V_j | V_{i_2}) \perp\!\!\!\perp V_{i_2}$ *iff* $\{V_i, V_j\} \in \mathbb{S}_1$ *and* $V_i \in \mathrm{Pa}^{\mathcal{G}}(V_j)$.

*(2)* *Suppose* $\{V_i, V_j\} \notin \mathbb{S}_1$, $\exists \{V_{i'}, V_{j'}\} \in \mathbb{S} \backslash \{\{V_i, V_j\}\}$ *s.t.* $\{V_{i'}, V_{j'}\} \cap \{V_i, V_j\} \neq \emptyset$ *only if (but not if)* $\{V_i, V_j\} \in \mathbb{S}_2$.

*(3)* *Suppose* $\{V_i, V_j\} \notin \mathbb{S}_1$, $\exists \{V_k, V_l\} \subseteq \bigcup \mathrm{Ch}^{\mathcal{H}_1}(\mathbf{V}_a \backslash \{V_i, V_j\})$ *s.t.* $(V_{i_1}, V_{i_2}, V_j, V_k, V_l)$ *satisfies the quintuple constraint only if (but not if)* $\{V_i, V_j\} \in \mathbb{S}_3$ *and* $V_i \in \mathrm{Pa}^{\mathcal{G}}(V_j)$.

**Remark.** (1,2) here are identical to (1,2) in Thm. 2 while (3) here is different from (3) in Thm. 2. Denote the result of our algorithm at this step by $\tilde{\mathbb{S}}_1, \tilde{\mathbb{S}}_2, \tilde{\mathbb{S}}_3$, this means that all pairs in $\mathbb{S}_1$ will

be incorporated into $\tilde{\mathbb{S}}_1$, all pairs in $\mathbb{S}_2$ will be incorporated into $\tilde{\mathbb{S}}_2$, but some pairs in $\mathbb{S}_3$ will be incorporated into $\tilde{\mathbb{S}}_2$ rather than $\tilde{\mathbb{S}}_3$. Formally, there is $\tilde{\mathbb{S}}_1 = \mathbb{S}_1$, $\tilde{\mathbb{S}}_2 \supseteq \mathbb{S}_2$, and $\tilde{\mathbb{S}}_3 \subseteq \mathbb{S}_3$.

**Theorem 10.** *(1)* $\forall \{V_i, V_j\} \in \tilde{\mathbb{S}}_2$, $\bigcap \mathrm{Pa}^{\mathcal{G}}(\{V_i, V_j\}) \subseteq \mathbf{V}_a$ *iff* $\exists \{V_{i'}, V_{j'}\} \in \tilde{\mathbb{S}}_1$ *s.t.* $\{V_i, V_j\} \cap \{V_{i'}, V_{j'}\} \neq \emptyset$. *(2)* $\forall \{\{V_i, V_j\}, \{V_{i'}, V_{j'}\}\} \subseteq \tilde{\mathbb{S}}_2$, $\bigcap \mathrm{Pa}^{\mathcal{G}}(\{V_i, V_j\}) = \bigcap \mathrm{Pa}^{\mathcal{G}}(\{V_{i'}, V_{j'}\})$ *iff* $\exists \{V_{i''}, V_{j''}\} \in \tilde{\mathbb{S}}_2$ *s.t.* $\{V_i, V_j\} \cap \{V_{i''}, V_{j''}\} \neq \emptyset$ *and* $\{V_{i'}, V_{j'}\} \cap \{V_{i''}, V_{j''}\} \neq \emptyset$.

**Remark.** Similar to Thm. 3, it also guarantees no duplicate identification of latent variables.

**Theorem 11.** *If Asmp. 1 is invalid, when* $\tilde{\mathbb{S}}_1 \cup \tilde{\mathbb{S}}_2 = \emptyset$, $\mathbf{V}_c \neq \emptyset$ *or there exists* $L \in \mathbf{V}_a$ *s.t.* $\mathrm{Ch}^{\mathcal{H}_1}(L) \nsubseteq \mathrm{PCh}^{\mathcal{G}}(L)$.

**Remark.** It is a variant of Thm. 4, which means that at the end of this section, there is an unidentified latent variable (e.g., Fig. 8(b)) or an identified latent variable whose recognized pure children (i.e., children in $\mathcal{H}_1$) are not actually its pure children (e.g., Fig. 8(d)).

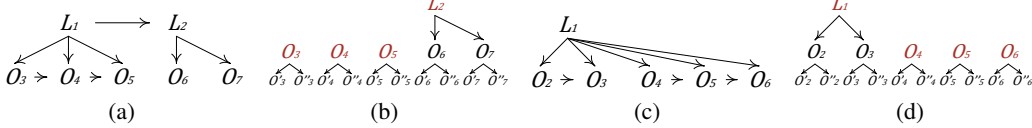

(a)    (b)    (c)    (d)

Figure 8: Suppose $\mathcal{G}_0$ is shown as (a), $\mathcal{H}_1$ at the end of stage 1 is shown as (b), where $\mathbf{V}_c = \{L_1\} \neq \emptyset$. Suppose $\mathcal{G}_0$ is shown as (c), $\mathcal{H}_1$ at the end of stage 1 is shown as (d), where $L_1 \in \mathbf{V}_a, \mathrm{Ch}^{\mathcal{H}_1}(L_1) = \{O_2, O_3\} \nsubseteq \mathrm{PCh}^{\mathcal{G}}(L_1)$.

### 4.2 Stage 2: Inferring Causal Relations

**§ High-level Overview.** We provide a variant of Thm. 5 as Thm. 12. In Sec. 3.2, there always exists a variable satisfying Thm. 5 at each iteration. But in this section, there exists no variable satisfying Thm. 12 at some iteration, when an error signal can be raised.

**Condition 4.** *(1)* $\forall V \in \mathbf{V}_a \backslash \mathbf{U}_a$, $\mathrm{Ch}^{\mathcal{H}_1}(V_i) \subseteq \mathrm{PCh}^{\mathcal{G}}(V_i)$. *(2)* $\forall V \in \mathbf{U}_a \cup \mathbf{V}_c$, $\mathrm{De}^{\mathcal{H}_2}(V) \subseteq \mathbf{U}_a \cup \mathbf{V}_c$. *(3)* $\forall V_i \in \mathbf{U}_a$, $X_{2i-1}, X_{2i}$ *can be written as*

$$X_{2i-1} = c_{2i-1} \sum_{V_j \in \mathbf{U}_a \cup \mathbf{V}_c} m_{ij} \epsilon_{V_j} + e_{X_{2i-1}} + e'_{X_{2i-1}}, \quad X_{2i} = c_{2i} \sum_{V_j \in \mathbf{U}_a \cup \mathbf{V}_c} m_{ij} \epsilon_{V_j} + e_{X_{2i}} + e'_{X_{2i}}, \quad (7)$$

*where* $\epsilon_{V_1}, ..., \epsilon_{V_n}, (e_{X_1}, e_{X_2})..., (e_{X_{2n-1}}, e_{X_{2n}}), (e'_{X_1}, e'_{X_3}, ..., e'_{X_{2n-1}}), (e'_{X_2}, e'_{X_4}, ..., e'_{X_{2n}})$ *are mutually independent and* $\forall j$ *s.t.* $\mathrm{Ch}^{\mathcal{H}_1}(V_j) \subseteq \mathrm{PCh}^{\mathcal{G}}(V_j)$, $e_{X_{2j-1}} \perp\!\!\!\perp e_{X_{2j}}$. *Without loss of generality, we assume each* $c_{2i-1}$ *is positive and each* $\epsilon_{V_j}$ *has variance 1.*

**Remark.** It is a variant of Cond. 2.

**Theorem 12.** $\forall V_i \in \mathbf{U}_a$, $\mathrm{An}^{\mathcal{H}_2}(V_i) \cap (\mathbf{U}_a \cup \mathbf{V}_c) = \emptyset$ *and* $\mathrm{Ch}^{\mathcal{H}_1}(V_i) \subseteq \mathrm{PCh}^{\mathcal{G}}(V_i)$ *iff* $\forall V_j \in \mathbf{U}_a \backslash \{V_i\}$, $R(X_{2j-1}, X_{2i-1}|X_{2i}) \perp\!\!\!\perp X_{2i}$.

**Remark.** It is a variant of Thm. 5, which implies that if at some iteration, there exists no $V_i \in \mathbf{U}_a$ s.t. $\mathrm{An}^{\mathcal{H}_2}(V_i) \cap (\mathbf{U}_a \cup \mathbf{V}_c) = \emptyset$ and $\mathrm{Ch}^{\mathcal{H}_1}(V_i) \subseteq \mathrm{PCh}^{\mathcal{G}}(V_i)$, we cannot find a $V_i \in \mathbf{U}_a$ satisfying the independence condition, when an error signal can be raised. In fact, this must happen before $\mathbf{U}_a$ becomes an empty set, more details are provided in the proof Thm. 13.

### 4.3 Summary

**Theorem 13.** *Suppose the observed variables are generated by a LiNGAM with latent variables satisfying the bottleneck-faithfulness assumption and Asmp. 2, if Asmp. 1 is invalid, in the limit of infinite data, our algorithm raises an error signal.*

## 5 Experiment

We first use four causal graphs shown as Fig. 9 to generate synthetic data. For each graph, we draw 10 sample sets of size 2k, 5k, 10k respectively. Each causal strength is sampled from a uniform

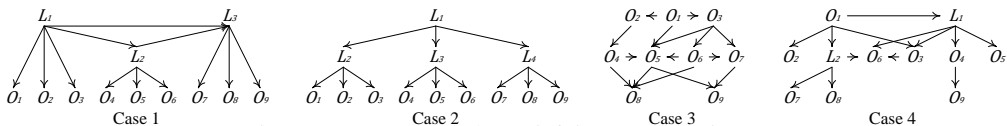

Figure 9: Causal graphs satisfying Assumption 1.

Table 1: Comparison on synthetic data. ↑ means higher is better while ↓ means lower is better.

| | | Error in Latent Variables ↓ | | | Correct-Ordering Rate ↑ | | | F1-Score ↑ | | | Running Time(s) ↓ | | |
|---|---|---|---|---|---|---|---|---|---|---|---|---|---|
| | | 2k | 5k | 10k | 2k | 5k | 10k | 2k | 5k | 10k | 2k | 5k | 10k |
| Case 1 | GIN | **0.0±0.0** | **0.0±0.0** | **0.0±0.0** | **1.00±0.00** | **1.00±0.00** | **1.00±0.00** | **1.00±0.00** | **1.00±0.00** | **1.00±0.00** | **1.21±0.18** | **1.39±0.13** | **1.63±0.16** |
| | LaHME | 0.3±0.5 | 0.1±0.3 | 0.2±0.4 | 0.83±0.26 | 0.94±0.17 | 0.89±0.23 | 0.93±0.11 | 0.98±0.08 | 0.95±0.10 | 1.39±0.15 | 1.61±0.11 | 1.87±0.14 |
| | Po-LiNGAM | 0.2±0.4 | 0.1±0.3 | **0.0±0.0** | 0.92±0.16 | 0.98±0.07 | **1.00±0.00** | 0.70±0.19 | 0.93±0.17 | 0.99±0.04 | 61.69±21.19 | 65.24±20.98 | 67.68±13.64 |
| | Ours | **0.0±0.0** | 0.1±0.3 | **0.0±0.0** | 0.92±0.13 | 0.91±0.18 | **1.00±0.00** | 0.98±0.03 | 0.97±0.07 | **1.00±0.00** | 1.80±0.19 | 2.18±0.23 | 2.49±0.11 |
| Case 2 | GIN | 1.0±0.0 | 1.0±0.0 | 1.0±0.0 | 0.43±0.00 | 0.43±0.02 | 0.43±0.00 | 0.75±0.00 | 0.74±0.02 | 0.75±0.00 | **1.29±0.11** | **1.58±0.15** | **1.71±0.18** |
| | LaHME | **0.0±0.0** | **0.0±0.0** | **0.0±0.0** | **1.00±0.00** | **1.00±0.00** | **1.00±0.00** | **1.00±0.00** | **1.00±0.00** | **1.00±0.00** | 1.38±0.13 | 1.63±0.24 | 1.81±0.18 |
| | PO-LiNGAM | 0.6±0.5 | 0.4±0.5 | 0.1±0.3 | 0.73±0.26 | 0.77±0.28 | 0.94±0.17 | 0.77±0.16 | 0.90±0.10 | 0.98±0.06 | 36.54±11.98 | 38.89±10.87 | 37.56±9.16 |
| | Ours | **0.0±0.0** | **0.0±0.0** | **0.0±0.0** | **1.00±0.00** | **1.00±0.00** | **1.00±0.00** | **1.00±0.00** | **1.00±0.00** | **1.00±0.00** | 1.56±0.03 | 1.81±0.05 | 2.26±0.17 |
| Case 3 | GIN | **0.0±0.0** | **0.0±0.0** | **0.0±0.0** | 0.00±0.00 | 0.00±0.00 | 0.00±0.00 | 0.00±0.00 | 0.00±0.00 | 0.00±0.00 | **0.54±0.07** | **0.58±0.06** | **0.68±0.06** |
| | LaHME | 1.0±0.0 | 0.9±0.3 | 1.0±0.0 | 0.00±0.00 | 0.00±0.00 | 0.00±0.00 | 0.00±0.00 | 0.00±0.00 | 0.00±0.00 | 7.83±0.79 | 8.71±0.52 | 10.38±0.49 |
| | PO-LiNGAM | **0.0±0.0** | **0.0±0.0** | **0.0±0.0** | 0.75±0.22 | 0.79±0.27 | 0.91±0.21 | 0.50±0.15 | 0.63±0.18 | 0.89±0.19 | 56.88±15.50 | 75.78±12.07 | 89.10±8.05 |
| | Ours | **0.0±0.0** | **0.0±0.0** | **0.0±0.0** | 0.98±0.04 | **1.00±0.00** | **1.00±0.00** | 0.92±0.10 | 0.98±0.03 | 0.99±0.01 | 3.64±0.10 | 4.20±0.13 | 5.02±0.14 |
| Case 4 | GIN | 0.9±0.3 | 0.9±0.3 | 0.9±0.3 | 0.19±0.03 | 0.20±0.00 | 0.19±0.03 | 0.28±0.02 | 0.29±0.01 | 0.27±0.05 | **0.91±0.10** | **0.94±0.10** | **1.11±0.11** |
| | LaHME | 1.8±0.6 | 2.0±0.0 | 2.0±0.0 | 0.22±0.02 | 0.20±0.00 | 0.20±0.00 | 0.34±0.03 | 0.32±0.00 | 0.32±0.01 | 2.27±0.34 | 2.61±0.37 | 3.22±0.79 |
| | PO-LiNGAM | 0.9±0.5 | 0.4±0.5 | **0.0±0.0** | 0.63±0.31 | 0.71±0.35 | **1.00±0.00** | 0.53±0.24 | 0.73±0.29 | **1.00±0.00** | 36.76±2.13 | 44.31±9.78 | 45.65±4.39 |
| | Ours | 0.3±0.5 | **0.0±0.0** | **0.0±0.0** | **0.91±0.15** | **1.00±0.00** | **1.00±0.00** | **0.87±0.19** | **1.00±0.01** | **1.00±0.00** | 4.30±0.25 | 4.90±0.13 | 5.94±0.12 |

distribution over $[-2.0, -0.5] \cup [0.5, 2.0]$ and each noise is generated from the seventh power of uniform distribution. We compare our methods with GIN (Xie et al., 2020), LaHME (Xie et al., 2022), and PO-LiNGAM (Jin et al., 2024). We use 3 metrics to evaluate the performance, including (i) *Error in Latent Variables*, the absolute difference between the estimated number of latent variables and the ground-truth one; (ii) *Correct-Ordering Rate*, the number of correctly estimated causal ordering divided by the number of causal ordering in the ground-truth graph; (iii) *F1-Score* of causal edges. The results are summarized in Tab. 1, where we also report the running time. In particular, we set the size of the largest atomic unit in GIN and PO-LiNGAM to 1 for a fair comparison.

With sufficient samples, all methods can handle case 1 properly. GIN does not perform well in case 2 where some latent variable has no observed pure child. Both GIN and LaHME are not suitable for case 3 and case 4 where the purity or measurement assumption is invalid. While PO-LiNGAM and our algorithm can both handle all cases properly, ours is far more efficient. PO-LiNGAM alternates between inferring causal relations and inferring causal relations from leaves to roots, whereas ours first identifying latent variables from leaves to roots and then infers causal relations from roots to leaves. The efficiency gap arises from distinct approaches for inferring causal relations. Take case 3 as an example, PO-LiNGAM first identifies $O_9$ as a leaf node by finding a subset $\mathbf{P} \subseteq \mathbf{O}_0 \backslash \{O_9\}$ s.t. a particular linear combination of $\mathbf{P} \cup \{O_9\}$ is independent of $\mathbf{O}_0 \backslash \{O_9\}$, where $\mathbf{P}$ is exactly $O_9$'s parents $\{O_5, O_7\}$. In contrast, our algorithm first identifies $O_1$ as a root node because for any $O_i \in \mathbf{O}_0 \backslash \{O_1\}$, $R(X_{2i-1}, X_1|X_2) \perp\!\!\!\perp X_2$. Clearly, PO-LiNGAM needs to traverse the power set of $\mathbf{O}_0 \backslash \{O_9\}$ while the ours only needs to traverse $\mathbf{O}_0 \backslash \{O_1\}$ itself.

In addition, we also do experiments on data generated by the graphs shown in Fig. 10, where Asmp. 1 is invalid. On 10 sample sets sized 10k for each case, while other algorithms all yield incorrect results, ours raises an error signal 8 times in case 5 and 7 times in case 6.

Figure 10: Causal graphs violating Asmp. 1.

We also apply our proposed algorithm to real-world data, more details are deferred to App. D.

## 6 CONCLUSION

In this paper, we focus on the setting where latent variables and observed variables are interconnected through complex causal relations. Under a pure children assumption, we propose an efficient algorithm, which is the first one capable of handling the setting with both latent variables and complex relations within polynomial time. Also, we prove trustworthiness of our algorithm. To the best of our knowledge, there is no similar result in the literature of causal discovery with latent variables.

**Limitations.** First, although Asmp. 1 allows the presence of complex causal relations, it is still somewhat restrictive, we will attempt to relax it without compromising efficiency significantly in our future work. Second, this work does not accommodate non-stationary (Liu & Kuang, 2023) and cyclic (Sethuraman et al., 2023) causal relations, on which we defer the research to our future work.

## ACKNOWLEDGMENTS

Tongliang Liu is partially supported by the following Australian Research Council projects: FT220100318, DP220102121, LP220100527, LP220200949, IC190100031. Xiu-Chuan Li is partially supported by ARC FT220100318.

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

## A    RELATED WORKS

Traditional causal discovery methods mostly assume that all task-relevant variables are observed (Spirtes & Glymour, 1991; Colombo & Maathuis, 2014; Chickering, 2002; Shimizu et al., 2006; 2011; Hoyer et al., 2009; Peters et al., 2014; Mooij et al., 2016). Unfortunately, latent variables are pervasive in practice, in which case these methods usually introduce spurious causal relations. This limitation has inspired extensive works on causal discovery with latent variables. Some early works can reveal causal relations between observed variables in the presence of latent variables, but their results are not informative of the number or the causal relations of latent variables. Specifically, some works (Hoyer et al., 2008; Tashiro et al., 2014; Maeda & Shimizu, 2020; Yang et al., 2022; Cai et al., 2023) explicitly assume that latent variables are all root variables, which means that there exists no causal relation between any two latent variables. Others (Spirtes et al., 1995; Claassen et al., 2013; Claassen & Bucur, 2022) allow the presence of causally-related latent variables, but they cannot identify latent variables, which means that both the number and the causal relations of latent variables are unknown. By utilizing linear models, others can identify latent variables and infer their causal relations (Silva et al., 2006; Anandkumar et al., 2013; Kummerfeld & Ramsey, 2016; Cai et al., 2019; Xie et al., 2020; Zeng et al., 2021; Adams et al., 2021; Chen et al., 2022; Xie et al., 2022; Huang et al., 2022; Chen et al., 2023; Dong et al., 2024; Jin et al., 2024; Li et al., 2024; Xie et al., 2024). Among the latter line of works, the *measurement assumption* is employed by all except Adams et al. (2021); Dong et al. (2024); Jin et al. (2024) and the *purity assumption* is used by all except Silva et al. (2006); Kummerfeld & Ramsey (2016); Adams et al. (2021); Dong et al. (2024); Jin et al. (2024); Li et al. (2024); Xie et al. (2024). In addition, Huang et al. (2022) employ the *non-triangle assumption* that any three variables are not mutually adjacent while Dong et al. (2024) weaken this assumption slightly by allowing three mutually adjacent variables only if they are all observed variables. That is, only Adams et al. (2021) and Jin et al. (2024) can handle setting where latent variables and observed variables are interconnected through complex causal relations, which are of particular relevance to our work.

Adams et al. (2021) are the first one investigating the important and challenging setting with both latent variables and complex relations, they present the sufficient and necessary condition for identifiability of LiNGAMs with latent variables, which is a really profound theoretical contribution. Also, using this condition as the assumption, they develop a causal discovery method, which is unfortunately inadvisable in practice as acknowledged by themselves. First, because it is based on overcomplete independent component analysis (OICA) which needs to know the number of source signals (Podosinnikova et al., 2019; Ding et al., 2019), it requires the number of latent variables as prior knowledge and is computationally intractable. Given the mixing matrix returned by OICA, it still needs to test which submatrices' singular values are exact zeros, which is rather sensitive to noise. Subsequently, Jin et al. (2024) strike a delicate balance between theoretical identifiability and practical feasibility. Specifically, under a stronger assumption involving pure children similar to many previous works (Cai et al., 2019; Xie et al., 2020; 2022; Huang et al., 2022; Dong et al., 2024), they propose the first practical algorithm capable of handling the setting with both latent variables and complex relations. But this algorithm has exponential time complexity with respect to the number of variables, seriously limiting its applicability. To overcome this limitation, under a similar assumption also involving pure children that is moderately more restrictive than Jin et al. (2024)'s[4], we propose an efficient algorithm which is the first one capable of handling the challenging setting within only polynomial time. Our algorithm differs significantly from theirs. Specifically, their algorithm follows a bottom-up pattern, which alternates between inferring causal relations and identifying latent variables, progressing from leaves to roots. Instead, ours follows a bottom-up then top-down pattern, which first sequentially identifies all latent variables, progressing from leaves to roots, and then sequentially infers causal relations, progressing from roots to leaves. Also, we prove trustworthiness of our algorithm, which means that it can raise an error signal rather than return an incorrect result when the pure children assumption is invalid. To the best of our knowledge, there is a complete lack of similar results in the literature of causal discovery with latent variables.

---

[4]On the one hand, our definition of pure child is more restrictive than Jin et al. (2024)'s as stated in Rem. of Def. 1, so our assumption is also more restrictive than theirs. On the other hand, the gap between these two assumptions is narrower than that between Adams et al. (2021)'s and Jin et al. (2024)'s because Adams et al. (2021) have absolutely no need for pure children, so we say our assumption is "moderately" more restrictive than Jin et al. (2024)'s.

Besides our work and that of Jin et al. (2024), many previous studies that allow the presence of latent variables but not complex relations also make similar pure children assumptions (Silva et al., 2006; Kummerfeld & Ramsey, 2016; Cai et al., 2019; Xie et al., 2020; Zeng et al., 2021; Chen et al., 2022; Xie et al., 2022; Huang et al., 2022; Chen et al., 2023; Dong et al., 2024; Xie et al., 2024). While the vast majority of them limit their scope to the scenario where the assumption is satisfied, Silva et al. (2006) systematically analyze the case where the assumption fails, rigorously characterizing the potential mistakes. For instance, their algorithm may fail to identify some latent variables and may erroneously introduce an edge from a latent into an observed variable that does not exist actually. Although this is an insightful result, it does not directly enable us to determine whether the recovered causal graph is correct. Instead, we prove that an error signal can be explicitly raised if the assumption fails.

While the works discussed above all focus on the linear case, there are also some studies investigating nonlinear problems, but most assume access to counterfactual data (Brehmer et al., 2022; Ahuja et al., 2022) or interventional data (Ahuja et al., 2023; Jiang & Aragam, 2023; Buchholz et al., 2023; Zhang et al., 2023). Notably, without structural restrictions such as the pure children assumption, even linear causal models satisfying the measurement assumption are unidentifiable without comprehensive interventional data obtained by intervening on each latent variable individually (Squires et al., 2023). To the best of our knowledge, in the presence of latent variables, only Kivva et al. (2021) and Kong et al. (2023) can handle non-linear problems through only observational data, but they both make rather strong assumptions. Specifically, Kivva et al. (2021) require that all latent variables are discrete. Kong et al. (2023) require that the mapping from all exogenous noises to observed variables is invertible. We leave further research on nonlinear problems to our future work.

## B  NOTATIONS

We summarize notations in Tab. 2

## C  PROOF

**Definition 7.** *(Unique minimal bottleneck) We say* $\mathbf{B}$ *is a bottleneck from* $\mathbf{J}$ *to* $\mathbf{K}$ *(* $\mathbf{J}, \mathbf{K}, \mathbf{B}$ *need not be mutually disjoint) if* $\forall J \in \mathbf{J}$ *and* $K \in \mathbf{K}$, *each directed path from* $J$ *to* $K$ *includes some* $B \in \mathbf{B}$. *Given a bottleneck* $\mathbf{B}_1$ *from* $\mathbf{J}$ *to* $\mathbf{K}$, *if any other bottleneck* $\mathbf{B} \neq \mathbf{B}_1$ *satisfying* $|\mathbf{B}| \geq |\mathbf{B}_1|$, *we say* $\mathbf{B}_1$ *is a minimal bottleneck. Furthermore, if* $|\mathbf{B}| > |\mathbf{B}_1|$, *we say* $\mathbf{B}_1$ *is the unique minimal bottleneck.*

**Assumption 3.** *For every* $\mathbf{J} \subseteq \mathbf{V}, \mathbf{K} \subseteq \mathbf{V}$, *if* $\mathbf{B}$ *is a minimal bottleneck from* $\mathbf{J}$ *to* $\mathbf{K}$, *then the rank of* $M_{\mathbf{K}}^{\mathbf{J}}$ *is exactly* $|\mathbf{B}|$, *where* $M_{\mathbf{K}}^{\mathbf{J}}$ *is a submatrix of* $M$ *with columns indexed by* $\mathbf{J}$ *and rows indexed by* $\mathbf{K}$ *and* $M$ *is a matrix composed of* $m_{ij}$ *in Eq. (2).*

> **Intuition.** This assumption implies that
>
> (1) $m_{ij} \neq 0$ iff $V_j \in \mathrm{GAn}(V_i)$.
>
> (2) Suppose $m_{ik}m_{jk}m_{il}m_{jl} \neq 0$, $m_{ik}/m_{jk} \neq m_{il}/m_{jl}$ iff there exist two non-intersecting paths from $\{V_k, V_l\}$ to $\{V_i, V_j\}$.
>
> **Remark.** In the following proof, rather than working directly with the bottleneck faithfulness assumption itself, we only utilize these two properties derived from it.

For ease of exposition, given $V \in \mathbf{V}$ and $\mathbf{V}' \subseteq \mathbf{V}$, we abbreviate $\mathrm{Pa}(V) \cap \mathbf{V}'$ as $\mathrm{Pa}_{\mathbf{V}'}(V)$ in the following.

### C.1  IMPORTANT LEMMAS

In this section, we summarize several important properties of the pseudo-residual (Lem. 1) and the quintuple constraint (Lem. 2), which serve as the cornerstones of the proofs of many following theoretical results.

Table 2: Summary of notations.

| Notation | Description | First appeared |
|---|---|---|
| $\mathcal{G}_0$ | Original ground-truth causal graph | Sec. 2 |
| $\mathbf{L}$ | Latent variables | Sec. 2 |
| $\mathbf{O}_0$ | Original observed variables | Sec. 2 |
| $\mathbf{V}_0$ | $\mathbf{L} \cup \mathbf{O}_0$ | Sec. 2 |
| $\mathcal{G}$ | Augmented ground-truth causal graph | Sec. 2 |
| $\mathbf{O}_1$ | Created observed variables | Sec. 2 |
| $\mathbf{O}$ | $\mathbf{O}_0 \cup \mathbf{O}_1$ | Sec. 2 |
| $\mathbf{V}$ | $\mathbf{V}_0 \cup \mathbf{O}_1$ | Sec. 2 |
| $\mathrm{Pa}(V)$ | Parents of $V$ | Sec. 2 |
| $\mathrm{Ch}(V)$ | Children of $V$ | Sec. 2 |
| $\mathrm{Ne}(V)$ | Neighbors of $V$ | Sec. 2 |
| $\mathrm{An}(V)$ | Ancestors of $V$ | Sec. 2 |
| $\mathrm{De}(V)$ | Descendants of $V$ | Sec. 2 |
| $\mathrm{GAn}(V)$ | Generalized ancestors of $V$, that is, $\mathrm{An}(V) \cup \{V\}$ | Sec. 2 |
| $\mathrm{GDe}(V)$ | Generalized descendants of $V$, that is, $\mathrm{De}(V) \cup \{V\}$ | Sec. 2 |
| $\mathrm{PCh}(V)$ | Pure children of $V$ | Def. 1 in Sec. 3 |
| $\mathrm{P^3Ch}(V)$ | Paired pseudo-pure children of $V$ | Def. 5 in Sec. 4 |
| $\mathrm{PDe}(V)$ | Pure descendants of $V$ | Def. 8 in App. C.2.1 |
| $\mathrm{P^2De}(V)$ | Pseudo-pure descendants of $V$ | Def. 9 in App. C.3.1 |
| $\mathbf{V}_a, \mathbf{V}_b$ | Two sets of variables maintained during stage 1 with specific initialization and update rules | Sec. 3.1 |
| $\mathbf{V}_c$ | $\mathbf{V} \backslash (\mathbf{V}_a \cup \mathbf{V}_b)$ | Sec. 3.1 |
| $\mathcal{H}_1$ | A graph over $\mathbf{V}_a \cup \mathbf{V}_b$ maintained during stage 1 with specific initialization and update rules | Sec. 3.1 |
| $\mathcal{H}_2$ | Induced subgraph of $\mathcal{G}$ over $\mathbf{V}_c \cup \mathbf{V}_a$ | Sec. 3.1 |
| $\mathbb{S}$ | Identifiable pairs in $\mathbf{V}_a$ | Def. 2 in Sec. 3.1 |
| $\mathbb{S}_1, \mathbb{S}_2, \mathbb{S}_3$ | Subsets of $\mathbb{S}$ | Def. 2 in Sec. 3.1 |
| $\tilde{\mathbb{S}}_1, \tilde{\mathbb{S}}_2, \tilde{\mathbb{S}}_3$ | Subsets of $\mathbb{S}$ return by our algorithm when Asmp. 1 is invalid | Rem. of Thm. 9 in Sec. 4.1 |
| $\mathrm{R}(V_1, V_2 \mid V_3)$ | Pseudo-residual of $V_1, V_2$ relative to $V_3$ | Def. 3 in Sec. 3.1 |
| $\mathbf{U}_a$ | Active set in stage 2 | Sec. 3.2 |

**Darmois-Skitovitch (D-S) Theorem.** (Kagan et al., 1973) Suppose two random variables $V_1$ and $V_2$ are both linear combinations of independent random variables $\{n_i\}_i$:

$$V_1 = \sum_i \alpha_i n_i, \quad V_2 = \sum_i \beta_i n_i. \tag{8}$$

Then, if $V_1 \perp\!\!\!\perp V_2$, each $n_i$ for which $\alpha_i \beta_i \neq 0$ follows Gaussian distribution. That is, if there exists a non-Gaussian $n_j$ s.t. $\alpha_j \beta_j \neq 0$, $V_1 \not\!\perp\!\!\!\perp V_2$.

**Definition 3.** (Pseudo-residual) Given three variables $V_1, V_2, V_3$ s.t. $\mathrm{Cov}(V_2, V_3) \neq 0$, the pseudo-residual of $V_1, V_2$ relative to $V_3$ is defined as

$$\mathrm{R}(V_1, V_2 \mid V_3) = V_1 - \frac{\mathrm{Cov}(V_1, V_3)}{\mathrm{Cov}(V_2, V_3)} V_2. \tag{9}$$

**Lemma 1.** *Given $V_1, V_2, V_3, V_4, V_5$ (it is possible that $V_3 = V_4 = V_5$) where $\mathrm{Cov}(V_1, V_2)\mathrm{Cov}(V_1, V_3)$ $\mathrm{Cov}(V_2, V_3) \neq 0$,*

*(1) If $V_1 = \lambda_1 e + e'_1$ and $V_2 = \lambda_2 e + e'_2$ where $(e'_1, e'_2) \perp\!\!\!\perp (e, V_3, V_4)$, $\mathrm{Var}(e) \neq 0$, and $\lambda_1 \lambda_2 \neq 0$, then $\mathrm{R}(V_1, V_2 \mid V_3) \perp\!\!\!\perp V_4$.*

*(2) If there exists $V_i$ s.t. only one of $m_{1i}$ and $m_{2i}$ is non-zero and $m_{4i} \neq 0$, then $\mathrm{R}(V_1, V_2 \mid V_3) \not\!\perp\!\!\!\perp V_4$.*

*(3) If there exists $V_i, V_j$ s.t. $m_{1i}m_{1j}m_{2i}m_{2j}m_{4i}m_{5j} \neq 0$ and $m_{1i}/m_{1j} \neq m_{2i}/m_{2j}$, then $\mathrm{R}(V_1, V_2 \mid V_3) \not\!\perp\!\!\!\perp V_4$ or $\mathrm{R}(V_1, V_2 \mid V_3) \not\!\perp\!\!\!\perp V_5$.*

**Remark.** (1) provides a sufficient condition for independence involving the pseudo-residual to hold while (2, 3) provides two sufficient conditions for independence involving the pseudo-residual to not hold.

*Proof.* The proofs of are as follows.

(1) $R(V_1, V_2|V_3) = (\lambda_1 e + e'_1) - \frac{\text{Cov}(\lambda_1 e + e'_1, V_3)}{\text{Cov}(\lambda_2 e + e'_2, V_3)}(\lambda_2 e + e'_2) = (\lambda_1 e + e'_1) - \frac{\lambda_1}{\lambda_2}(\lambda_2 e + e'_2) = e'_1 - \frac{\lambda_1}{\lambda_2}e'_2 \perp\!\!\!\perp V_4$.

(2) As only one of $m_{1i}$ and $m_{2i}$ is non-zero, $R(V_1, V_2|V_3)$ contains $\epsilon_{V_i}$. Since $m_{4i} \neq 0$, based on D-S Theorem, $R(V_1, V_2|V_3) \not\perp\!\!\!\perp V_4$.

(3) As $m_{1i}m_{1j}m_{2i}m_{2j} \neq 0$ and $m_{1i}/m_{1j} \neq m_{2i}/m_{2j}$, $R(V_1, V_2|V_3)$ contains $\epsilon_{V_i}$ or $\epsilon_{V_j}$. Since $m_{4i}m_{5j} \neq 0$, based on D-S Theorem, $R(V_1, V_2|V_3) \not\perp\!\!\!\perp V_4$ or $R(V_1, V_2|V_3) \not\perp\!\!\!\perp V_5$.

$\square$

**Definition 4.** (Quintuple constraint) We say $(V_1, V_2, V_3, V_4, V_5)$ satisfies the quintuple constraint if there exist $\alpha, \beta$ s.t. $V_1 + \alpha V_3 + \beta V_4 \perp\!\!\!\perp V_2$ and $\text{Cov}(V_1 + \alpha V_3 + \beta V_4, V_5) = 0$.

**Lemma 2.** *Given $V_1, V_2, V_3, V_4, V_5$,*

*(1) If there exists $V_i$ s.t. $m_{1i}m_{2i} \neq 0$ and $m_{3i} = m_{4i} = 0$, then $(V_1, V_2, V_3, V_4, V_5)$ does not satisfy the quintuple constraint.*

*(2) Suppose $e_i, e_j, e'_1, e'_2, e'_3, e'_4$ are mutually independent and $V_1, V_2, V_3, V_4$ can be written as*

$$V_1 = \lambda_1 e_i + \gamma_1 e_j + e'_1, \quad V_2 = \lambda_2 e_i + \gamma_2 e_j + e'_2, \quad V_3 = \lambda_3 e_i + \gamma_3 e_j + e'_3, \quad V_4 = \lambda_4 e_i + e'_4, \quad (10)$$

*where $\text{Var}(e_i)\text{Var}(e_j) \neq 0$, $\lambda_1\lambda_2\lambda_3\lambda_4 \neq 0$, and $\gamma_1\gamma_2\gamma_3 \neq 0$. (2.a) If $V_5 \perp\!\!\!\perp (e_j, e'_1, e'_2, e'_3, e'_4)$ and $\text{Cov}(V_5, e_i) \neq 0$, then $(V_1, V_2, V_3, V_4, V_5)$ satisfies the quintuple constraint; (2.b) If $V_5 \perp\!\!\!\perp (e_j, e'_1, e'_2, e'_3)$, $\text{Cov}(V_5, e_i)\text{Cov}(V_5, e'_4) \neq 0$, and $\lambda_1/\lambda_3 \neq \gamma_1/\gamma_3$, then $(V_1, V_2, V_3, V_4, V_5)$ does not satisfy the quintuple constraint.*

**Remark.** (2.a) provides a sufficient condition for the quintuple constraint to hold while (1, 2.b) provides two sufficient conditions for the quintuple constraint to not hold.

*Proof.* The proofs are as follows.

(1) As $m_{1i} \neq 0$ and $m_{3i} = m_{4i} = 0$, $V_1 + \alpha V_3 + \beta V_4$ contains $\epsilon_{V_i}$. Since $m_{2i} \neq 0$, based on D-S Theorem, for any $\alpha, \beta$, $V_1 + \alpha V_3 + \beta V_4 \not\perp\!\!\!\perp V_2$.

(2) If $V_5 \perp\!\!\!\perp (e_j, e'_1, e'_2, e'_3, e'_4)$ and $\text{Cov}(V_5, e_i) \neq 0$, let $\text{Cov}(V_1 + \alpha V_3 + \beta V_4, V_2) = 0$ and $\text{Cov}(V_1 + \alpha V_3 + \beta V_4, V_5) = 0$, we have

$$\lambda_1 + \alpha\lambda_3 + \beta\lambda_4 = 0, \quad \gamma_1 + \alpha\gamma_3 = 0,$$

then

$$V_1 + \alpha V_3 + \beta V_4 = (\lambda_1 + \alpha\lambda_3 + \beta\lambda_4)e_i + (\gamma_1 + \alpha\gamma_3)e_j + e'_1 + \alpha e'_3 + \beta e'_4 = e'_1 + \alpha e'_3 + \beta e'_4 \perp\!\!\!\perp V_2.$$

If $V_5 \perp\!\!\!\perp (e_j, e'_1, e'_2, e'_3)$, $\text{Cov}(V_5, e_i)\text{Cov}(V_5, e'_4) \neq 0$, and $\lambda_1/\lambda_3 \neq \gamma_1/\gamma_3$, let $\text{Cov}(V_1 + \alpha V_3 + \beta V_4, V_2) = 0$ and $\text{Cov}(V_1 + \alpha V_3 + \beta V_4, V_5) = 0$, we have

$$\lambda_1 + \alpha\lambda_3 + \beta\lambda_4 \neq 0, \quad \gamma_1 + \alpha\gamma_3 \neq 0,$$

then

$$V_1 + \alpha V_3 + \beta V_4 = (\lambda_1 + \alpha\lambda_3 + \beta\lambda_4)e_i + (\gamma_1 + \alpha\gamma_3)e_j + e'_1 + \alpha e'_3 + \beta e'_4 \quad (11)$$

contains $e_i$ and $e_j$, so $V_1 + \alpha V_3 + \beta V_4 \not\perp\!\!\!\perp V_2$.

$\square$

## C.2 PROOF OF THEORETICAL RESULTS IN SEC. 3

**Definition 1.** (Pure child) We say $V_2$ is a pure child of $V_1$, denoted by $V_2 \in \mathrm{PCh}(V_1)$, if (i) $\mathrm{Pa}(V_2) = \{V_1\}$ and (ii) $\forall V \in \mathrm{De}(V_2), |\mathrm{Pa}(V)| = 1$.

> **Remark.** Based on this definition, if $V_2 \in \mathrm{PCh}(V_1)$, then $\mathrm{Ch}(V_2) = \mathrm{PCh}(V_2)$.

**Assumption 1.** $\forall L \in \mathbf{L}, |\mathrm{PCh}^{\mathcal{G}_0}(L)| \geq 2$ and $|\mathrm{Ne}^{\mathcal{G}_0}(L)| \geq 3$.

Trivially, if Asmp. 1 holds, then $\forall L \in \mathbf{L}, |\mathrm{PCh}^{\mathcal{G}}(L)| \geq 2$ and $|\mathrm{Ne}^{\mathcal{G}}(L)| \geq 3$.

### C.2.1 PROOF OF THEORETICAL RESULTS IN SEC. 3.1

**Condition 1.** (1) $\forall V \in \mathbf{V}_b, |\mathrm{Pa}^{\mathcal{H}_1}(V)| = 1$ and $\mathrm{Ch}^{\mathcal{H}_1}(V) = \mathrm{PCh}^{\mathcal{G}}(V)$; (2) $\forall V \in \mathbf{V}_a, \mathrm{Pa}^{\mathcal{H}_1}(V) = \emptyset$, $|\mathrm{Ch}^{\mathcal{H}_1}(V)| \geq 2$, and $\mathrm{Ch}^{\mathcal{H}_1}(V) \subseteq \mathrm{PCh}^{\mathcal{G}}(V)$.

Before proving theoretical results in the main text one by one, we first introduce two corollaries (Cors. 1 and 2) readily derived from Cond. 1.

**Corollary 1.** *(1)* $\forall V \in \mathbf{V}_b, \mathrm{Ch}^{\mathcal{G}}(V) = \mathrm{Ch}^{\mathcal{H}_1}(V)$ *and* $\mathrm{Pa}^{\mathcal{G}}(V) = \mathrm{Pa}^{\mathcal{H}_1}(V)$; *(2)* $\forall V \in \mathbf{V}_c, \mathrm{Ch}^{\mathcal{G}}(V) = \mathrm{Ch}^{\mathcal{H}_2}(V)$ *and* $\mathrm{Pa}^{\mathcal{G}}(V) = \mathrm{Pa}^{\mathcal{H}_2}(V)$; *(3)* $\forall V \in \mathbf{V}_a, \mathrm{Ch}^{\mathcal{G}}(V) = \mathrm{Ch}^{\mathcal{H}_1}(V) \cup \mathrm{Ch}^{\mathcal{H}_2}(V)$ *and* $\mathrm{Pa}^{\mathcal{G}}(V) = \mathrm{Pa}^{\mathcal{H}_2}(V)$.

> **Remark.** This corollary reveals the properties of variables in $\mathbf{V}_b$, $\mathbf{V}_c$, and $\mathbf{V}_a$. (1) means that for each variable in $\mathbf{V}_b$, its parents and children in the underlying causal graph $\mathcal{G}$ are exactly its parents and children in $\mathcal{H}_1$. (2) means that for each variable in $\mathbf{V}_c$, its parents and children in the underlying causal graph $\mathcal{G}$ are exactly its parents and children in $\mathcal{H}_2$. (3) means that for each variable in $\mathbf{V}_a$, its children in the underlying causal graph $\mathcal{G}$ are the union of its children in $\mathcal{H}_1$ and its children in $\mathcal{H}_2$ while its parents in $\mathcal{G}$ are exactly its parents in $\mathcal{H}_2$. This corollary is widely used in the following proofs. To maintain fluency, we will use it without further citation.

*Proof.* First, if $V_i \in \mathbf{V}_b$, then $\mathrm{Ch}^{\mathcal{H}_1}(V_i) = \mathrm{PCh}^{\mathcal{G}}(V_i)$ and there exists $V_j \in \mathbf{V}_a \cup \mathbf{V}_b$ s.t. $\mathrm{Pa}^{\mathcal{H}_1}(V_i) = \{V_j\}$ based on Cond. 1(1). Moreover, since (i) $\mathrm{Ch}^{\mathcal{H}_1}(V_j) = \mathrm{PCh}^{\mathcal{G}}(V_j)$ if $V_j \in \mathbf{V}_b$ based on Cond. 1(1) and (ii) $\mathrm{Ch}^{\mathcal{H}_1}(V_j) \subseteq \mathrm{PCh}^{\mathcal{G}}(V_j)$ if $V_j \in \mathbf{V}_a$ based on Cond. 1(2), there is always $V_i \in \mathrm{PCh}^{\mathcal{G}}(V_j)$. Therefore, we can conclude that $\mathrm{Pa}^{\mathcal{G}}(V_i) = \{V_j\} = \mathrm{Pa}^{\mathcal{H}_1}(V_i)$ and $\mathrm{Ch}^{\mathcal{G}}(V_i) = \mathrm{PCh}^{\mathcal{G}}(V_i) = \mathrm{Ch}^{\mathcal{H}_1}(V_i)$, this completes the proof of (1).

Second, if $V_i \in \mathbf{V}_c$, based on (1) that was just proven, $\mathrm{Ch}^{\mathcal{G}}(V_i) \cap \mathbf{V}_b = \emptyset$ and $\mathrm{Pa}^{\mathcal{G}}(V_i) \cap \mathbf{V}_b = \emptyset$, which is equivalent to $\mathrm{Ch}^{\mathcal{G}}(V_i) = \mathrm{Ch}^{\mathcal{H}_2}(V_i)$ and $\mathrm{Pa}^{\mathcal{G}}(V_i) = \mathrm{Pa}^{\mathcal{H}_2}(V_i)$ based on the definition of $\mathcal{H}_2$, this completes the proof of (2).

Third, if $V_i \in \mathbf{V}_a$, then $\mathrm{Ch}^{\mathcal{G}}(V_i) = \mathrm{Ch}^{\mathcal{G}}_{\mathbf{V}_b}(V_i) \cup \mathrm{Ch}^{\mathcal{G}}_{\mathbf{V}_c \cup \mathbf{V}_a}(V_i) = \mathrm{Ch}^{\mathcal{H}_1}(V) \cup \mathrm{Ch}^{\mathcal{H}_2}(V)$. Besides, since $\mathrm{Pa}^{\mathcal{H}_1}(V) = \emptyset$ based on Cond. 1(2), $\mathrm{Pa}^{\mathcal{G}}(V) = \mathrm{Pa}^{\mathcal{H}_2}(V)$. This completes the proof of (3). □

**Definition 8.** *(Pure descendant) We say $V_2$ is a pure descendant of $V_1$, denoted by $V_2 \in \mathrm{PDe}(V_1)$, if $V_2 \in \bigcup \mathrm{GDe}(\mathrm{PCh}(V_1))$*

> **Example.** In Fig. 2(a), $\mathrm{PDe}(L_1) = \{L_3, L_4, O_{12}, O_{13}, O_{14}, O_{15}, O_{16}\}$.

> **Remark.** Based on this definition, if $V_2 \in \mathrm{PDe}(V_1)$, then $\mathrm{Ch}(V_2) = \mathrm{PCh}(V_2)$ and $\mathrm{De}(V_2) = \mathrm{PDe}(V_2)$.

**Corollary 2.** $\forall V \in \mathbf{V}_c, |\mathrm{PDe}^{\mathcal{G}}_{\mathbf{V}_a}(V)| \geq 2$. *If* $|\mathrm{De}^{\mathcal{G}}_{\mathbf{V}_a}(V)| = 2$, *then* $\mathrm{De}^{\mathcal{G}}_{\mathbf{V}_a}(V) = \mathrm{PCh}^{\mathcal{G}}(V)$.

> **Remark.** This corollary means that each variable in $\mathbf{V}_c$ has at least two pure descendants in $\mathbf{V}_a$, and if it has exactly two descendants in $\mathbf{V}_a$, these two descendants are exactly its all pure children. Intuitively, this enables us to identify variables in $\mathbf{V}_c$ through analyzing variables in $\mathbf{V}_a$.

*Proof.* Note that $\mathbf{V}_c \subseteq \mathbf{L}$, $\forall V \in \mathbf{V}_c$, we have $|\mathrm{PCh}^{\mathcal{G}}(V)| \geq 2$ and $\mathrm{PCh}^{\mathcal{G}}(V) \subseteq \mathbf{V}_a \cup \mathbf{V}_c$. If $|\mathrm{PCh}^{\mathcal{G}}_{\mathbf{V}_a}(V)| \geq 2$, we have $|\mathrm{PDe}^{\mathcal{G}}_{\mathbf{V}_a}(V)| \geq 2$ naturally. Otherwise, $\mathrm{PDe}^{\mathcal{G}}_{\mathbf{V}_c}(V) \supseteq \mathrm{PCh}^{\mathcal{G}}_{\mathbf{V}_c}(V) \neq \emptyset$. Let

$V_i \in \mathrm{PDe}_{\mathbf{V}_c}^{\mathcal{G}}(V)$ s.t. $\mathrm{Ch}^{\mathcal{H}_2}(V_i) \subseteq \mathbf{V}_a$, we have $|\mathrm{PCh}_{\mathbf{V}_a}^{\mathcal{G}}(V_i)| \geq 2$, so $|\mathrm{PDe}_{\mathbf{V}_a}^{\mathcal{G}}(V)| \geq |\mathrm{PCh}_{\mathbf{V}_a}^{\mathcal{G}}(V_i)| \geq 2$. Therefore, $\forall V \in \mathbf{V}_c$, there is always $|\mathrm{PDe}_{\mathbf{V}_a}^{\mathcal{G}}(V)| \geq 2$.

Following the above analysis, given $V \in \mathbf{V}_c$, for any $V' \in \mathrm{PCh}_{\mathbf{V}_c}^{\mathcal{G}}(V)$, we have $|\mathrm{PDe}_{\mathbf{V}_a}^{\mathcal{G}}(V')| \geq 2$, so

$$|\mathrm{De}_{\mathbf{V}_a}^{\mathcal{G}}(V)| \geq 2|\mathrm{PCh}_{\mathbf{V}_c}^{\mathcal{G}}(V)| + |\mathrm{PCh}_{\mathbf{V}_a}^{\mathcal{G}}(V)|. \tag{12}$$

Note that

$$|\mathrm{PCh}^{\mathcal{G}}(V)| = |\mathrm{PCh}_{\mathbf{V}_c}^{\mathcal{G}}(V)| + |\mathrm{PCh}_{\mathbf{V}_a}^{\mathcal{G}}(V)| \geq 2, \tag{13}$$

if $|\mathrm{De}_{\mathbf{V}_a}^{\mathcal{G}}(V)| = 2$, then $\mathrm{PCh}_{\mathbf{V}_c}^{\mathcal{G}}(V) = \emptyset$ and $\mathrm{De}_{\mathbf{V}_a}^{\mathcal{G}}(V) = \mathrm{PCh}_{\mathbf{V}_a}^{\mathcal{G}}(V)$, that is, $\mathrm{De}_{\mathbf{V}_a}^{\mathcal{G}}(V) = \mathrm{PCh}^{\mathcal{G}}(V)$. $\quad\square$

**Proposition 1.** $\forall \{V_i, V_j, V_k, V_l\} \subseteq \mathbf{V}_a$ where $\mathrm{Cov}(V_i, V_j)\mathrm{Cov}(V_i, V_k)\mathrm{Cov}(V_j, V_k) \neq 0$ and $\mathrm{Cov}(V_i, V_j)\mathrm{Cov}(V_i, V_l)\mathrm{Cov}(V_j, V_l) \neq 0$, $\mathrm{R}(V_i, V_j|V_k) \perp\!\!\!\perp \mathbf{V}_a \backslash \{V_i, V_j\}$ iff $\mathrm{R}(V_i, V_j|V_l) \perp\!\!\!\perp \mathbf{V}_a \backslash \{V_i, V_j\}$.

> **Remark.** Given $\{V_i, V_j\} \subseteq \mathbf{V}_a$, denote $\{V \in \mathbf{V}_a \backslash \{V_i, V_j\} | \mathrm{Cov}(V_i, V_j)\mathrm{Cov}(V, V_i)\mathrm{Cov}(V, V_j) \neq 0\}$ by $\mathbf{V}_{ij}$, this proposition means that there exists no $\{V_k, V_l\} \subseteq \mathbf{V}_{ij}$ s.t. $\mathrm{R}(V_i, V_j|V_k) \perp\!\!\!\perp \mathbf{V}_a \backslash \{V_i, V_j\}$ and $\mathrm{R}(V_i, V_j|V_l) \not\perp\!\!\!\perp \mathbf{V}_a \backslash \{V_i, V_j\}$. Therefore, if we want to know whether for each $V \in \mathbf{V}_{ij}$, $\mathrm{R}(V_i, V_j|V) \perp\!\!\!\perp \mathbf{V}_a \backslash \{V_i, V_j\}$, we only need to consider any single $V_k \in \mathbf{V}_{ij}$.

*Proof.* If $\mathrm{R}(V_i, V_j|V_k) \perp\!\!\!\perp \mathbf{V}_a \backslash \{V_i, V_j\}$, then $\mathrm{R}(V_i, V_j|V_k) \perp\!\!\!\perp V_l$, which means that $\mathrm{Cov}(\mathrm{R}(V_i, V_j|V_k), V_l) = 0$, that is, $\frac{\mathrm{Cov}(V_i, V_k)}{\mathrm{Cov}(V_j, V_k)} = \frac{\mathrm{Cov}(V_i, V_l)}{\mathrm{Cov}(V_j, V_l)}$, so $\mathrm{R}(V_i, V_j|V_l) = \mathrm{R}(V_i, V_j|V_k) \perp\!\!\!\perp \mathbf{V}_a \backslash \{V_i, V_j\}$. Similarly, if $\mathrm{R}(V_i, V_j|V_l) \perp\!\!\!\perp \mathbf{V}_a \backslash \{V_i, V_j\}$, there is also $\mathrm{R}(V_i, V_j|V_k) \perp\!\!\!\perp \mathbf{V}_a \backslash \{V_i, V_j\}$. $\quad\square$

**Theorem 1.** $\forall \{V_i, V_j\} \subseteq \mathbf{V}_a$, $\{V_i, V_j\} \in \mathbb{S}$ iff there exists $V_k \in \mathbf{V}_a \backslash \{V_i, V_j\}$ s.t. $\mathrm{Cov}(V_i, V_j)\mathrm{Cov}(V_i, V_k)\mathrm{Cov}(V_j, V_k) \neq 0$ and for each such $V_k$, $\mathrm{R}(V_i, V_j|V_k) \perp\!\!\!\perp \mathbf{V}_a \backslash \{V_i, V_j\}$.

> **Proof Sketch.** If $\{V_i, V_j\} \in \mathbb{S}$, we can prove correlation and independence based on Lem. 1(1). Otherwise, for each possible case, we can prove either non-correlation or dependence based on Lem. 1(2,3).

*Proof.* "Only if".

(1) Suppose $\{V_i, V_j\} \in \mathbb{S}_1$. Let $V_i \in \mathrm{Pa}^{\mathcal{H}_2}(V_j)$ without loss of generality and $V_k \in \mathrm{Ne}^{\mathcal{H}_2}(V_i) \backslash \{V_j\}$, we have $\mathrm{GDe}_{\mathbf{V}_a}^{\mathcal{H}_2}(V_k) \backslash \{V_i, V_j\} \neq \emptyset^5$ and let $V_l \in \mathrm{GDe}_{\mathbf{V}_a}^{\mathcal{H}_2}(V_k) \backslash \{V_i, V_j\}$. Clearly, $V_i, V_j, V_l$ are correlated to each other. Also, $V_i = V_i + 0$, $V_j = a_{ji}V_i + \epsilon_{V_j}$ where $\forall V \in \mathbf{V}_a \backslash \{V_i, V_j\}$, $(0, \epsilon_{V_j}) \perp\!\!\!\perp (V_i, V_l, V)$, so $\mathrm{R}(V_i, V_j|V_l) \perp\!\!\!\perp \mathbf{V}_a \backslash \{V_i, V_j\}$ based on Lem. 1(1). Combined with Prop. 1, we reach the conclusion.

(2) Suppose $\{V_i, V_j\} \in \mathbb{S}_2$. Let $\mathrm{Pa}^{\mathcal{H}_2}(V_i) = \mathrm{Pa}^{\mathcal{H}_2}(V_j) = \{V_k\}$. If $V_k \in \mathbf{V}_a$, we let $V_m = V_k$. Otherwise, let $V_l \in \mathrm{Ne}^{\mathcal{H}_2}(V_k) \backslash \{V_i, V_j\}$. Similarly to fn. 5, we have $\mathrm{GDe}_{\mathbf{V}_a}^{\mathcal{H}_2}(V_l) \backslash \{V_i, V_j\} \neq \emptyset$ and let $V_m \in \mathrm{GDe}_{\mathbf{V}_a}^{\mathcal{H}_2}(V_l) \backslash \{V_i, V_j\}$. Clearly, $V_i, V_j, V_m$ are correlated to each other. Also, $V_i = a_{ik}V_k + \epsilon_{V_i}$, $V_j = a_{jk}V_k + \epsilon_{V_j}$ where $\forall V \in \mathbf{V}_a \backslash \{V_i, V_j\}$, $(\epsilon_{V_i}, \epsilon_{V_j}) \perp\!\!\!\perp (V_k, V_m, V)$, so $\mathrm{R}(V_i, V_j|V_m) \perp\!\!\!\perp \mathbf{V}_a \backslash \{V_i, V_j\}$ based on Lem. 1(1). Combined with Prop. 1, we reach the conclusion.

(3) Suppose $\{V_i, V_j\} \in \mathbb{S}_3$. Let $V_i \in \mathrm{Pa}^{\mathcal{H}_2}(V_j)$ without loss of generality and $\mathrm{Pa}^{\mathcal{H}_2}(V_i) = \{V_k\}$. Similarly to fn. 5, we also have $\mathrm{GDe}_{\mathbf{V}_a}^{\mathcal{H}_2}(V_k) \backslash \{V_i, V_j\} \neq \emptyset$ and let $V_l \in \mathrm{GDe}_{\mathbf{V}_a}^{\mathcal{H}_2}(V_k) \backslash \{V_i, V_j\}$. Clearly, $V_i, V_j, V_l$ are correlated to each other. Also, $V_i = a_{ik}V_k + \epsilon_{V_i}$, $V_j = (a_{ik}a_{ji} + a_{jk})V_k + (a_{ji}\epsilon_{V_i} + \epsilon_{V_j})$ where $\forall V \in \mathbf{V}_a \backslash \{V_i, V_j\}$, $(\epsilon_{V_i}, a_{ji}\epsilon_{V_i} + \epsilon_{V_j}) \perp\!\!\!\perp (V_k, V_l, V)$, so $\mathrm{R}(V_i, V_j|V_l) \perp\!\!\!\perp \mathbf{V}_a \backslash \{V_i, V_j\}$ based on Lem. 1(1). Combined with Prop. 1, we reach the conclusion.

---

[5]There are three possible cases. (i) If $V_k \in \mathbf{V}_a$, we have $V_k \in \mathrm{GDe}_{\mathbf{V}_a}^{\mathcal{H}_2}(V_k) \backslash \{V_i, V_j\} \neq \emptyset$. (ii) If $V_k \in \mathbf{V}_c$ and $\{V_i, V_j\} \not\subseteq \mathrm{PDe}^{\mathcal{G}}(V_k)$, based on Cor. 2, we have $\mathrm{GDe}_{\mathbf{V}_a}^{\mathcal{H}_2}(V_k) \backslash \{V_i, V_j\} \neq \emptyset$. (iii) If $V_k \in \mathbf{V}_c$ and $\{V_i, V_j\} \subseteq \mathrm{PDe}^{\mathcal{G}}(V_k)$, since $V_j \notin \mathrm{PCh}^{\mathcal{G}}(V_k)$, based on Cor. 2, we have $\mathrm{GDe}_{\mathbf{V}_a}^{\mathcal{H}_2}(V_k) \backslash \{V_i, V_j\} \neq \emptyset$.

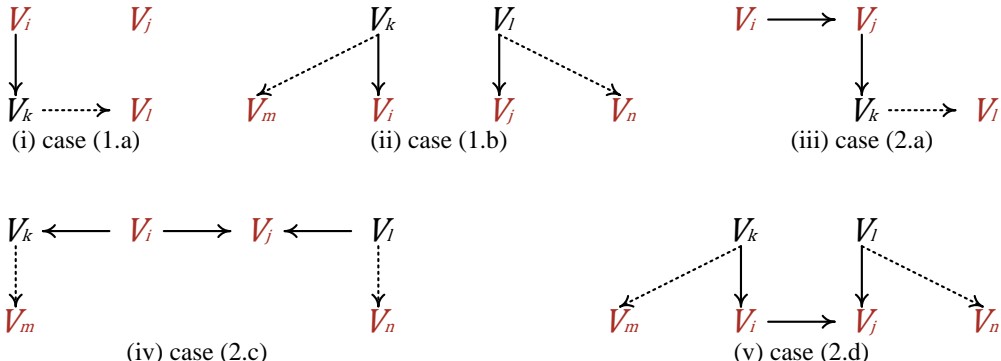

Figure 11: Illustration of "if" part in proof of Thm. 1. A dotted arrow from $V_1$ to $V_2$ means that $V_2 \in \mathrm{GDe}^{\mathcal{H}_2}(V_1)$. $V$ is marked in red if $V \in \mathbf{V}_a$.

"If". We prove this part by contradiction. Suppose $\{V_i, V_j\} \notin \mathbb{S}$.

(1) Suppose $V_i \notin \mathrm{Ne}^{\mathcal{H}_2}(V_j)$, all possible cases are as follows.

   (a) Suppose $\mathrm{Ch}^{\mathcal{H}_2}(V_i) \neq \emptyset$ or $\mathrm{Ch}^{\mathcal{H}_2}(V_j) \neq \emptyset$, we take the former as an example without loss of generality. Let $V_k \in \mathrm{Ch}^{\mathcal{H}_2}(V_i)$. Similarly to fn. 5, we have $\mathrm{GDe}_{\mathbf{V}_a}^{\mathcal{H}_2}(V_k) \backslash \{V_i, V_j\} \neq \emptyset$ and let $V_l \in \mathrm{GDe}_{\mathbf{V}_a}^{\mathcal{H}_2}(V_k) \backslash \{V_i, V_j\}$. An illustrative example is shown as Fig. 11(i). If $V_j \notin \mathrm{De}^{\mathcal{H}_2}(V_i)$, $m_{ii}m_{li} \neq 0$ and $m_{ji} = 0$, so for any $V \in \mathbf{V}_a \backslash \{V_i, V_j\}$ s.t. $V, V_i, V_j$ are correlated to each other, $\mathrm{R}(V_i, V_j | V) \not\perp V_l$ based on Lem. 1(2), which leads to contradiction. Otherwise, if $V_j \in \mathrm{De}^{\mathcal{H}_2}(V_i)$, without loss of generality, we assume $V_j \in \mathrm{De}^{\mathcal{H}_2}(V_k)$, in which case $m_{jk}m_{lk} \neq 0$ and $m_{ik} = 0$, so for any $V \in \mathbf{V}_a \backslash \{V_i, V_j\}$ s.t. $V, V_i, V_j$ are correlated to each other, $\mathrm{R}(V_i, V_j | V) \not\perp V_l$ based on Lem. 1(2), which leads to contradiction.

   (b) Suppose $\mathrm{Ch}^{\mathcal{H}_2}(V_i) = \mathrm{Ch}^{\mathcal{H}_2}(V_j) = \emptyset$, $\mathrm{Pa}^{\mathcal{H}_2}(V_i) \neq \emptyset$, and $\mathrm{Pa}^{\mathcal{H}_2}(V_j) \neq \emptyset$. Since $\{V_i, V_j\} \notin \mathbb{S}_2$, there exist $\{V_k, V_l\} \subseteq \mathbf{V}_a \cup \mathbf{V}_c \backslash \{V_i, V_j\}$ s.t. $V_k \in \mathrm{Pa}^{\mathcal{H}_2}(V_i)$ and $V_l \in \mathrm{Pa}^{\mathcal{H}_2}(V_j)$. Similarly to fn. 5, we have $\mathrm{GDe}_{\mathbf{V}_a}^{\mathcal{H}_2}(V_k) \backslash \{V_i, V_j\} \neq \emptyset$, $\mathrm{GDe}_{\mathbf{V}_a}^{\mathcal{H}_2}(V_l) \backslash \{V_i, V_j\} \neq \emptyset$ and let $V_m \in \mathrm{GDe}_{\mathbf{V}_a}^{\mathcal{H}_2}(V_k) \backslash \{V_i, V_j\}$, $V_n \in \mathrm{GDe}_{\mathbf{V}_a}^{\mathcal{H}_2}(V_l) \backslash \{V_i, V_j\}$ (It is possible that $V_m = V_n$). An illustrative example is shown as Fig. 11(ii). If $V_i \notin \mathrm{De}^{\mathcal{H}_2}(V_l)$ or $V_j \notin \mathrm{De}^{\mathcal{H}_2}(V_k)$, we take the former as an example without loss of generality, then $m_{jl}m_{nl} \neq 0$ and $m_{il} = 0$, so for any $V \in \mathbf{V}_a \backslash \{V_i, V_j\}$ s.t. $V, V_i, V_j$ are correlated to each other, $\mathrm{R}(V_i, V_j | V) \not\perp V_n$ based on Lem. 1(2), which leads to contradiction. Otherwise, $m_{ik}m_{jk}m_{il}m_{jl} \neq 0$. Since there exist two non-intersecting directed paths from $\{V_k, V_l\}$ to $\{V_i, V_j\}$ (e.g., $V_k \to V_i$ and $V_l \to V_j$), $m_{ik}/m_{il} \neq m_{jk}/m_{jl}$. Also, $m_{mk}m_{nl} \neq 0$. So for any $V \in \mathbf{V}_a \backslash \{V_i, V_j\}$ s.t. $V, V_i, V_j$ are correlated to each other, $\mathrm{R}(V_i, V_j | V) \not\perp V_m$ or $\mathrm{R}(V_i, V_j | V) \not\perp V_n$ based on Lem. 1(3), which leads to contradiction.

   (c) Suppose $\mathrm{Ne}^{\mathcal{H}_2}(V_i) = \emptyset$ or $\mathrm{Ne}^{\mathcal{H}_2}(V_j) = \emptyset$, we take the former as an example without loss of generality. Clearly, $\mathrm{Cov}(V_i, V_j) = 0$, which leads to contradiction.

(2) Assume $V_i \in \mathrm{Pa}^{\mathcal{H}_2}(V_j)$ or $V_j \in \mathrm{Pa}^{\mathcal{H}_2}(V_i)$, we take the former as an example without loss of generality, all possible cases are as follows.

   (a) Suppose $\mathrm{Ch}^{\mathcal{H}_2}(V_j) \neq \emptyset$. Let $V_k \in \mathrm{Ch}^{\mathcal{H}_2}(V_j)$. Similarly to fn. 5, we have $\mathrm{GDe}_{\mathbf{V}_a}^{\mathcal{H}_2}(V_k) \backslash \{V_i, V_j\} \neq \emptyset$ and let $V_l \in \mathrm{GDe}_{\mathbf{V}_a}^{\mathcal{H}_2}(V_k) \backslash \{V_i, V_j\}$. An illustrative example is shown as Fig. 11(iii). Clearly, $m_{jj}m_{lj} \neq 0$ and $m_{ij} = 0$, so for any $V \in \mathbf{V}_a \backslash \{V_i, V_j\}$ s.t. $V, V_i, V_j$ are correlated to each other, $\mathrm{R}(V_i, V_j | V) \not\perp V_l$ based on Lem. 1(2), which leads to contradiction.

(b) Suppose $\text{Pa}^{\mathcal{H}_2}(V_j)\backslash\{V_i\} = \emptyset$ and $\text{Ch}^{\mathcal{H}_2}(V_j) = \emptyset$. Since $\{V_i, V_j\} \notin \mathbb{S}_1$, we have $\text{Ne}^{\mathcal{H}_2}(V_i)\backslash\{V_j\} = \emptyset$. Clearly, for any $V \in \mathbf{V}_a\backslash\{V_i, V_j\}$, $\text{Cov}(V, V_i) = 0$ which leads to contradiction.

(c) Suppose $\text{Ch}^{\mathcal{H}_2}(V_i)\backslash\{V_j\} \neq \emptyset$, $\text{Pa}^{\mathcal{H}_2}(V_j)\backslash\{V_i\} \neq \emptyset$, and $\text{Ch}^{\mathcal{H}_2}(V_j) = \emptyset$. Let $V_k \in \text{Ch}^{\mathcal{H}_2}(V_i)\backslash\{V_j\}$ and $V_l \in \text{Pa}^{\mathcal{H}_2}(V_j)\backslash\{V_i\}$ (It is possible that $V_k = V_l$). Similarly to fn. 5, we have $\text{GDe}_{\mathbf{V}_a}^{\mathcal{H}_2}(V_k)\backslash\{V_i, V_j\} \neq \emptyset$, $\text{GDe}_{\mathbf{V}_a}^{\mathcal{H}_2}(V_l)\backslash\{V_i, V_j\} \neq \emptyset$ and let $V_m \in \text{GDe}_{\mathbf{V}_a}^{\mathcal{H}_2}(V_k)\backslash\{V_i, V_j\}$, $V_n \in \text{GDe}_{\mathbf{V}_a}^{\mathcal{H}_2}(V_l)\backslash\{V_i, V_j\}$ (It is possible that $V_m = V_n$). An illustrative example is shown as Fig. 11(iv). If $V_i \notin \text{De}^{\mathcal{H}_2}(V_l)$, $m_{jl}m_{nl} \neq 0$ and $m_{il} = 0$, so for any $V \in \mathbf{V}_a\backslash\{V_i, V_j\}$ s.t. $V, V_i, V_j$ are correlated to each other, $\text{R}(V_i, V_j|V) \not\perp V_n$ based on Lem. 1(2), which leads to contradiction. Otherwise, $m_{ii}m_{il}m_{ji}m_{jl} \neq 0$, since there exist two non-intersecting directed paths from $\{V_i, V_l\}$ to $\{V_i, V_j\}$ (e.g., $V_i$ and $V_l \rightarrow V_j$), $m_{ii}/m_{il} \neq m_{ji}/m_{jl}$. Also, $m_{mi}m_{nl} \neq 0$, so for any $V \in \mathbf{V}_a\backslash\{V_i, V_j\}$ s.t. $V, V_i, V_j$ are correlated to each other, $\text{R}(V_i, V_j|V) \not\perp V_m$ or $\text{R}(V_i, V_j|V) \not\perp V_n$ based on Lem. 1(3), which leads to contradiction.

(d) Suppose $\text{Pa}^{\mathcal{H}_2}(V_i) \neq \emptyset$, $\text{Ch}^{\mathcal{H}_2}(V_i)\backslash\{V_j\} = \emptyset$, $\text{Pa}^{\mathcal{H}_2}(V_j)\backslash\{V_i\} \neq \emptyset$, and $\text{Ch}^{\mathcal{H}_2}(V_j) = \emptyset$. Since $\{V_i, V_j\} \notin \mathbb{S}_3$, there exist $\{V_k, V_l\} \subseteq \mathbf{V}_a \cup \mathbf{V}_c\backslash\{V_i, V_j\}$ s.t. $V_k \in \text{Pa}^{\mathcal{H}_2}(V_i)$ and $V_l \in \text{Pa}^{\mathcal{H}_2}(V_j)$. Similarly to fn. 5, we have $\text{GDe}_{\mathbf{V}_a}^{\mathcal{H}_2}(V_k)\backslash\{V_i, V_j\} \neq \emptyset$, $\text{GDe}_{\mathbf{V}_a}^{\mathcal{H}_2}(V_l)\backslash\{V_i, V_j\} \neq \emptyset$ and let $V_m \in \text{GDe}_{\mathbf{V}_a}^{\mathcal{H}_2}(V_k)\backslash\{V_i, V_j\}$, $V_n \in \text{GDe}_{\mathbf{V}_a}^{\mathcal{H}_2}(V_l)\backslash\{V_i, V_j\}$ (It is possible that $V_m = V_n$). An illustrative example is shown as Fig. 11(v). Then the proof is similar to case (1.b).

(e) Suppose $\text{Pa}^{\mathcal{H}_2}(V_i) = \emptyset$, $\text{Ch}^{\mathcal{H}_2}(V_i)\backslash\{V_j\} = \emptyset$, $\text{Pa}^{\mathcal{H}_2}(V_j)\backslash\{V_i\} \neq \emptyset$, and $\text{Ch}^{\mathcal{H}_2}(V_j) = \emptyset$. Clearly, for any $V \in \mathbf{V}_a\backslash\{V_i, V_j\}$, $\text{Cov}(V, V_i) = 0$, which leads to contradiction.

$\square$

**Theorem 2.** $\forall\{V_i, V_j\} \in \mathbb{S}$, let $\{V_{i_1}, V_{i_2}\} \subseteq \text{Ch}^{\mathcal{H}_1}(V_i)$.

(1) $\text{R}(V_{i_1}, V_j|V_{i_2}) \perp V_{i_2}$ iff $\{V_i, V_j\} \in \mathbb{S}_1$ and $V_i \in \text{Pa}^{\mathcal{G}}(V_j)$.

(2) Suppose $\{V_i, V_j\} \notin \mathbb{S}_1$, $\exists\{V_{i'}, V_{j'}\} \in \mathbb{S}\backslash\{\{V_i, V_j\}\}$ s.t. $\{V_{i'}, V_{j'}\} \cap \{V_i, V_j\} \neq \emptyset$ only if (but not if) $\{V_i, V_j\} \in \mathbb{S}_2$.

(3) Suppose $\{V_i, V_j\} \notin \mathbb{S}_1$, $\exists\{V_k, V_l\} \subseteq \bigcup \text{Ch}^{\mathcal{H}_1}(\mathbf{V}_a\backslash\{V_i, V_j\})$ s.t. $(V_{i_1}, V_{i_2}, V_j, V_k, V_l)$ satisfies the quintuple constraint iff $\{V_i, V_j\} \in \mathbb{S}_3$ and $V_i \in \text{Pa}^{\mathcal{G}}(V_j)$.

**Proof Sketch.** For (1), if $\{V_i, V_j\} \in \mathbb{S}_1$ and $V_i \in \text{Pa}^{\mathcal{G}}(V_j)$, we can prove independence based on Lem. 1(1); otherwise, for each possible case, we can prove dependence based on Lem. 1(2,3). For (2), "only if" can be readily derived from the definitions of $\mathbb{S}_1$, $\mathbb{S}_2$, and $\mathbb{S}_3$ while "not if" can be proven by an example, which is $\{O_7, O_8\}$ in Fig. 3. For (3), if $\{V_i, V_j\} \in \mathbb{S}_3$ and $V_i \in \text{Pa}^{\mathcal{G}}(V_j)$, letting $V$ be the common parent of $V_i, V_j$ and $V_k, V_l$ be respective generalized descendants of $V$'s any two pure children, we can prove the quintuple constraint is satisfied based on Lem. 2(2.a); otherwise, for each possible case, we can prove it is not satisfied based on Lem. 2(1).

*Proof.* Cond. 1 indicates that $\forall V \in \mathbf{V}_a, \text{Ch}^{\mathcal{H}_1}(V) \subseteq \text{PCh}^{\mathcal{G}}(V)$.

- "If" of (1): Clearly, $V_{i_1} = a_{i_1 i}V_i + \epsilon_{V_{i_1}}$, $V_j = a_{ji}V_i + \epsilon_{V_j}$ where $(\epsilon_{V_{i_1}}, \epsilon_{V_j}) \perp (V_i, V_{i_2})$, so we can reach the conclusion based on Lem. 1(1).

- "Only if" of (1): We prove this part by contradiction.

    - Suppose (i) $\{V_i, V_j\} \in \mathbb{S}_1$ and $V_j \in \text{Pa}^{\mathcal{G}}(V_i)$, or (ii) $\{V_i, V_j\} \in \mathbb{S}_2$, or (iii) $\{V_i, V_j\} \in \mathbb{S}_3$ and $V_j \in \text{Pa}^{\mathcal{G}}(V_i)$. Since $m_{i_1 i}m_{i_2 i} \neq 0$ and $m_{ji} = 0$, $\text{R}(V_{i_1}, V_j|V_{i_2}) \not\perp V_{i_2}$ based on Lem. 1(2), which leads to contradiction.

    - Suppose $\{V_i, V_j\} \in \mathbb{S}_3$ and $V_i \in \text{Pa}^{\mathcal{G}}(V_j)$. Let $\text{Pa}^{\mathcal{H}_2}(V_i) = \{V_k\}$. Clearly, $m_{i_1 i}m_{i_1 k}m_{ji}m_{jk} \neq 0$. Since there exist two non-intersecting directed paths from $\{V_i, V_k\}$ to $\{V_{i_1}, V_j\}$ (e.g., $V_i \rightarrow V_{i_1}$ and $V_k \rightarrow V_j$), $m_{i_1 i}/m_{i_1 k} \neq m_{ji}/m_{jk}$. Also, $m_{i_2 i}m_{i_2 k} \neq 0$, so $\text{R}(V_{i_1}, V_j|V_{i_2}) \not\perp V_{i_2}$ based on Lem. 1(3), which leads to contradiction.

- "Only if" of (2): This follows the definitions of $\mathbb{S}_1$, $\mathbb{S}_2$, and $\mathbb{S}_3$.

- "Not if" of (2): An example is $\{O_7, O_8\}$ in Fig. 3.

- "If" of (3): Let $\mathrm{Pa}^{\mathcal{H}_2}(V_i) = \{V_h\}$. If $V_h \in \mathbf{V}_a$, let $\{V_k, V_l\} \subseteq \mathrm{Ch}^{\mathcal{H}_1}(V_h)$. Otherwise, $V_h \in \mathbf{V}_c \subseteq \mathbf{L}$, let $\{V_{h_1}, V_{h_2}\} \subseteq \mathrm{PCh}^{\mathcal{G}}(V_h)$. Similarly to fn. 5, we can obtain $\mathrm{GDe}^{\mathcal{H}_2}_{\mathbf{V}_a}(V_{h_1}) \setminus \{V_i, V_j\} \neq \emptyset$ and $\mathrm{GDe}^{\mathcal{H}_2}_{\mathbf{V}_a}(V_{h_2}) \setminus \{V_i, V_j\} \neq \emptyset$, let $V_k \in \bigcup \mathrm{Ch}^{\mathcal{H}_1}(\mathrm{GDe}^{\mathcal{H}_2}_{\mathbf{V}_a}(V_{h_1}))$ and $V_l \in \bigcup \mathrm{Ch}^{\mathcal{H}_1}(\mathrm{GDe}^{\mathcal{H}_2}_{\mathbf{V}_a}(V_{h_2}))$. In both cases, $V_{i_1}, V_{i_2}, V_j, V_k$ can be expressed as

$$V_{i_1} = a_{ih} a_{i_1 i} V_h + a_{i_1 i} \epsilon_{V_i} + \epsilon_{V_{i_1}} \tag{14}$$

$$V_{i_2} = a_{ih} a_{i_2 i} V_h + a_{i_2 i} \epsilon_{V_i} + \epsilon_{V_{i_2}} \tag{15}$$

$$V_j = (a_{ih} a_{ji} + a_{jh}) V_h + a_{ji} \epsilon_{V_i} + \epsilon_{V_j}, \tag{16}$$

$$V_k = \lambda_k V_h + e_{V_k}, \tag{17}$$

where $V_h, \epsilon_{V_i}, \epsilon_{V_{i_1}}, \epsilon_{V_{i_2}}, \epsilon_{V_j}, e_{V_k}$ are independent of each other, $V_l \perp\!\!\!\perp (\epsilon_{V_i}, \epsilon_{V_{i_1}}, \epsilon_{V_{i_2}}, \epsilon_{V_j}, e_{V_k})$ and $\mathrm{Cov}(V_l, V_h) \neq 0$, so $(V_{i_1}, V_{i_2}, V_j, V_k, V_l)$ satisfies the quintuple constraint based on Lem. 2(2.a).

- "Only if" of (3): We prove this part by contradiction. Suppose (i) $\{V_i, V_j\} \in \mathbb{S}_3$ and $V_j \in \mathrm{Pa}^{\mathcal{H}_2}(V_i)$ or (ii) $\{V_i, V_j\} \in \mathbb{S}_2$. Clearly, $m_{i_1 i} m_{i_2 i} \neq 0$, $m_{ji} = 0$, and for any $V_k$, $m_{ki} = 0$, so $(V_{i_1}, V_{i_2}, V_j, V_k, V_l)$ does not satisfy the quintuple constraint based on Lem. 2(1), which leads to contradiction.

$\square$

**Remark.** For the proof of (1) and (3), it can be readily verified that if both $V_{i_1}$ and $V_{i_2}$ are identical to $V_i$ (i.e., $a_{i_1 i} = a_{i_2 i} = 1$ and $\epsilon_{V_{i_1}} = \epsilon_{V_{i_2}} = 0$). Clearly, when $V_i$ refers to any $O_j \in \mathbf{O}_0$, $\{V_{i_1}, V_{i_2}\}$ refers to $\{O'_j, O''_j\} \subseteq \mathbf{O}_1$, which are two created children of $O_j$. This means that in the practical implementation, we can create $O'_j$ and $O''_j$ by directly making two copies of $O_j$, without the need to add independent non-Gaussian noises into $O_j$.

**Theorem 3.** (1) $\forall \{V_i, V_j\} \in \mathbb{S}_2, \bigcap \mathrm{Pa}^{\mathcal{G}}(\{V_i, V_j\}) \subseteq \mathbf{V}_a$ iff $\exists \{V_{i'}, V_{j'}\} \in \mathbb{S}_1$ s.t. $\{V_i, V_j\} \cap \{V_{i'}, V_{j'}\} \neq \emptyset$. (2) $\forall \{\{V_i, V_j\}, \{V_{i'}, V_{j'}\}\} \subseteq \mathbb{S}_2, \bigcap \mathrm{Pa}^{\mathcal{G}}(\{V_i, V_j\}) = \bigcap \mathrm{Pa}^{\mathcal{G}}(\{V_{i'}, V_{j'}\})$ iff $\exists \{V_{i''}, V_{j''}\} \in \mathbb{S}_2$ s.t. $\{V_i, V_j\} \cap \{V_{i''}, V_{j''}\} \neq \emptyset$ and $\{V_{i'}, V_{j'}\} \cap \{V_{i''}, V_{j''}\} \neq \emptyset$.

*Proof.* This can be readily derived from the definitions of $\mathbb{S}_1$ and $\mathbb{S}_2$. $\square$

**Proposition 2.** *Cond. 1 is still valid after update.*

*Proof.* We denote $\mathbf{V}_b, \mathbf{V}_a, \mathbf{V}_c, \mathcal{H}_1, \mathcal{H}_2, \mathbb{S}$ after update by $\mathbf{V}'_b, \mathbf{V}'_a, \mathbf{V}'_c, \mathcal{H}'_1, \mathcal{H}'_2, \mathbb{S}'$ respectively. While Cond. 1 is valid for $\mathbf{V}'_b \cap \mathbf{V}_b$ and $\mathbf{V}'_a \cap \mathbf{V}_a$ trivially, we focus on $\mathbf{V}'_b \setminus \mathbf{V}_b$ and $\mathbf{V}'_a \setminus \mathbf{V}_a$.

- If $V_j \in \mathbf{V}'_b \setminus \mathbf{V}_b$, then $V_j \in \mathbf{V}_a$ and there exists $V_i \in \mathbf{V}_a$ s.t. (i) $\{V_i, V_j\} \in \mathbb{S}_1$ and $V_i \in \mathrm{Pa}^{\mathcal{H}_2}(V_j)$ or (ii) $\{V_i, V_j\} \in \mathbb{S}_2$. In both cases, $|\mathrm{Pa}^{\mathcal{H}'_1}(V_j)| = 1$ because $\mathrm{Pa}^{\mathcal{H}'_1}(V_j) = \{V_i\}$ in case (i) and $\mathrm{Pa}^{\mathcal{H}'_1}(V_j) = \bigcap \mathrm{Pa}^{\mathcal{H}_2}(\{V_i, V_j\})$ in case (ii). Also, based on the definitions of $\mathbb{S}_1$ and $\mathbb{S}_2$, $V_j$ has no pure child in $\mathcal{H}_2$, so $\mathrm{Ch}^{\mathcal{H}_1}(V_j) \subseteq \mathrm{PCh}^{\mathcal{G}}(V_j)$ reduces to $\mathrm{Ch}^{\mathcal{H}_1}(V_j) = \mathrm{PCh}^{\mathcal{G}}(V_j)$, then $\mathrm{Ch}^{\mathcal{H}'_1}(V_j) = \mathrm{Ch}^{\mathcal{H}_1}(V_j) = \mathrm{PCh}^{\mathcal{G}}(V_j)$.

- If $V_j \in \mathbf{V}'_a \setminus \mathbf{V}_a$, then $\forall \{V_k, V_l\} \subseteq \mathrm{Ch}^{\mathcal{H}'_1}(V_j)$, $\{V_k, V_l\} \in \mathbb{S}_2$ and $\bigcup \mathrm{Pa}^{\mathcal{H}_2}(\{V_k, V_l\}) = \{V_j\}$, so $\mathrm{Ch}^{\mathcal{H}'_1}(V_j) \subseteq \mathrm{PCh}^{\mathcal{G}}(V_j)$. Besides, it is trivial that $\mathrm{Pa}^{\mathcal{H}'_1}(V_j) = \emptyset$.

$\square$

**Theorem 4.** If $\mathbb{S}_1 \cup \mathbb{S}_2 = \emptyset$, $\mathbf{V}_c = \emptyset$.

*Proof.* We prove this by contradiction. Suppose $\mathbf{V}_c \neq \emptyset$ and $V_i \in \mathbf{V}_c \subseteq \mathbf{L}$.

- If $\bigcup \mathrm{Ch}^{\mathcal{H}_2}(\mathrm{PCh}^{\mathcal{G}}(V_i)) = \emptyset$, then $\mathrm{PCh}^{\mathcal{G}}(V_i) \subseteq \mathbf{V}_a$. Conversely, if $\mathrm{PCh}^{\mathcal{G}}(V_i) \nsubseteq \mathbf{V}_a$, there exists $V \in \mathrm{PCh}^{\mathcal{G}}_{\mathbf{V}_c}(V_i) \subseteq \mathbf{L}$ and $\mathrm{Ch}^{\mathcal{G}}(V) = \mathrm{Ch}^{\mathcal{H}_2}(V) = \emptyset$, contradicting Asmp. 1. For each $\{V_j, V_k\} \subseteq \mathrm{PCh}^{\mathcal{G}}(V_i)$, it is trivial that $\{V_j, V_k\} \in \mathbb{S}_2$, that is, $\mathbb{S}_2 \neq \emptyset$, which leads to contradiction.

- If $\bigcup \mathrm{Ch}^{\mathcal{H}_2}(\mathrm{PCh}^{\mathcal{G}}(V_i)) \neq \emptyset$, let $\mathbf{V}' = \{V | V \in \mathrm{PDe}^{\mathcal{G}}(V_i), \mathrm{Ch}^{\mathcal{H}_2}(V) \neq \emptyset, \bigcup \mathrm{Ch}^{\mathcal{H}_2}(\mathrm{Ch}^{\mathcal{H}_2}(V_j)) = \emptyset\}$ and $V_j \in \mathbf{V}'$, then $\mathrm{Ch}^{\mathcal{H}_2}(V_j) \subseteq \mathbf{V}_a$ as proven by contradiction above. If $V_j \in \mathbf{V}_a$, then for any $V_k \in \mathrm{Ch}^{\mathcal{H}_2}(V_j)$, it is trivial that $\{V_j, V_k\} \in \mathbb{S}_1$, that is, $\mathbb{S}_1 \neq \emptyset$. Otherwise, $V_j \in \mathbf{V}_c \subseteq \mathbf{L}$, for each $\{V_k, V_l\} \subseteq \mathrm{Ch}^{\mathcal{H}_2}(V_j)$, it is trivial that $\{V_k, V_l\} \in \mathbb{S}_2$, that is, $\mathbb{S}_2 \neq \emptyset$. Both cases lead to contradiction.

$\square$

### C.2.2 PROOF OF THEORETICAL RESULTS IN SEC. 3.2

**Condition 2.** (1) $\forall V \in \mathbf{U}_a, \mathrm{De}^{\mathcal{H}_2}(V) \subseteq \mathbf{U}_a$. (2) $\forall V_i \in \mathbf{U}_a, X_{2i-1}, X_{2i}$ can be written as

$$X_{2i-1} = c_{2i-1} \sum_{V_j \in \mathbf{U}_a} m_{ij} \epsilon_{V_j} + e_{X_{2i-1}} + e'_{X_{2i-1}}, \quad X_{2i} = c_{2i} \sum_{V_j \in \mathbf{U}_a} m_{ij} \epsilon_{V_j} + e_{X_{2i}} + e'_{X_{2i}}, \quad (18)$$

where $\epsilon_{V_1}, ..., \epsilon_{V_n}, e_{X_1}, ..., e_{X_{2n}}, (e'_{X_1}, e'_{X_3}, ..., e'_{X_{2n-1}}), (e'_{X_2}, e'_{X_4}, ..., e'_{X_{2n}})$ are mutually independent. Without loss of generality, we assume each $c_{2i-1}$ is positive and each $\epsilon_{V_j}$ has variance 1.

**Theorem 5.** $\forall V_i \in \mathbf{U}_a, \mathrm{An}^{\mathcal{H}_2}(V_i) \cap \mathbf{U}_a = \emptyset$ iff $\forall V_j \in \mathbf{U}_a \backslash \{V_i\}, \mathrm{R}(X_{2j-1}, X_{2i-1} | X_{2i}) \perp\!\!\!\perp X_{2i}$.

*Proof.* When $|\mathbf{U}_a| = 1$, the proof is trivial, we focus on the case $|\mathbf{U}_a| > 1$.

"Only if": As $\mathrm{An}^{\mathcal{H}_2}(V_i) \cap \mathbf{U}_a = \emptyset$, $X_{2i-1}$ and $X_{2i}$ can be written as

$$X_{2i-1} = c_{2i-1} \epsilon_{V_i} + e_{X_{2i-1}} + e'_{X_{2i-1}}, \quad X_{2i} = c_{2i} \epsilon_{V_i} + e_{X_{2i}} + e'_{X_{2i}}. \quad (19)$$

- If $\mathrm{Cov}(X_{2j-1}, X_{2i}) = 0$, then $m_{ji} = 0$, so $\mathrm{R}(X_{2j-1}, X_{2i-1} | X_{2i}) = X_{2j-1} \perp\!\!\!\perp X_{2i}$.

- If $\mathrm{Cov}(X_{2j-1}, X_{2i}) \neq 0$, then $m_{ji} \neq 0$. $X_{2j-1}$ can be written as

$$X_{2j-1} = c_{2j-1} m_{ji} \epsilon_{V_i} + c_{2j-1} \sum_{V_k \in \mathbf{U}_a \backslash \{V_i\}} m_{jk} \epsilon_{V_k} + e_{X_{2j-1}} + e'_{X_{2j-1}}, \quad (20)$$

where $(e_{X_{2i-1}} + e'_{X_{2i-1}}, c_{2j-1} \sum_{V_k \in \mathbf{U}_a \backslash \{V_i\}} m_{jk} \epsilon_{V_k} + e_{X_{2j-1}} + e'_{X_{2j-1}}) \perp\!\!\!\perp (\epsilon_{V_i}, X_{2i})$, so $\mathrm{R}(X_{2j-1}, X_{2i-1} | X_{2i}) \perp\!\!\!\perp X_{2i}$ based on Lem. 1(1).

"If": We prove this part by contradiction. Let $V_j \in \mathrm{An}^{\mathcal{H}_2}_{\mathbf{U}_a}(V_i)$. Clearly, $X_{2i-1}, X_{2i}, X_{2j-1}$ are correlated to each other. Since $X_{2i-1}$ and $X_{2i}$ both contain $\epsilon_{V_i}$ while $X_{2j-1}$ does not contain $\epsilon_{V_i}$, so $\mathrm{R}(X_{2j-1}, X_{2i-1} | X_{2i}) \not\perp\!\!\!\perp X_{2i}$ based on Lem. 1(2), which leads to contradiction. $\square$

**Theorem 6.** If $V_i$ satisfies Thm. 5, then $\mathrm{Cov}(X_{2i-1}, X_{2i}) = c_{2i-1} c_{2i}$ and $\forall V_j \in \mathbf{U}_a \backslash \{V_i\}$,

$$\mathrm{sgn}(m_{ji}) = \mathrm{sgn}(\frac{\mathrm{Cov}(X_{2i-1}, X_{2j})}{\mathrm{Cov}(X_{2j-1}, X_{2j})}), \quad \mathrm{Cov}(X_{2i-1}, X_{2j}) \mathrm{Cov}(X_{2i}, X_{2j-1}) = c_{2i-1} c_{2i} c_{2j-1} c_{2j} m_{ji}^2. \quad (21)$$

Besides, $\forall V_k \in \mathbf{V}_a \backslash \mathbf{U}_a, m_{ki} = 0$.

*Proof.* It is trivial that $\forall V_j \in \mathbf{V}_a \backslash \mathbf{U}_a, m_{ji} = 0$ because $V_i \notin \mathrm{An}^{\mathcal{H}_2}(V_j)$ based on Cond. 2(1). Since we assume each $\epsilon_{V_i}$ has variance 1 and each $c_{2i-1}$ is positive without loss of generality, $\forall V_j \in \mathbf{U}_a \backslash \{V_i\}$,

$$\frac{\mathrm{Cov}(X_{2i-1}, X_{2j})}{\mathrm{Cov}(X_{2j-1}, X_{2j})} = \frac{c_{2i-1} m_{ji}}{c_{2j-1} \mathrm{Var}(\sum_{V_k \in \mathbf{U}_a} m_{jk} \epsilon_{V_k})}, \quad (22)$$

so $\mathrm{sgn}(m_{ji}) = \mathrm{sgn}(\frac{\mathrm{Cov}(X_{2i-1}, X_{2j})}{\mathrm{Cov}(X_{2j-1}, X_{2j})})$. Besides, it is trivial that

$$\mathrm{Cov}(X_{2i-1}, X_{2j}) \mathrm{Cov}(X_{2i}, X_{2j-1}) = c_{2i-1} c_{2i} c_{2j-1} c_{2j} m_{ji}^2, \quad \mathrm{Cov}(X_{2i-1}, X_{2i}) = c_{2i-1} c_{2i}. \quad (23)$$

Finally, $\forall V_k \in \mathbf{V}_a \backslash \mathbf{U}_a, V_k \notin \mathrm{De}^{\mathcal{G}}(V_i)$, so $m_{ki} = 0$. $\square$

**Proposition 3.** *Cond. 2 is still valid after removal.*

*Proof.* Based on Thm. 5, Cond. 2(1) holds trivially. Besides,

$$R(X_{2j-1}, X_{2i-1}|X_{2i}) = c_{2j-1} \sum_{V_k \in \mathbf{U}_a \setminus \{V_i\}} m_{jk} \epsilon_{V_k} + e_{X_{2j-1}} + \underbrace{e'_{X_{2j-1}} - \frac{m_{ji} c_{2j-1}}{c_{2i-1}} (e_{X_{2i-1}} + e'_{X_{2i-1}})}_{\text{updated } e'_{X_{2j-1}}},$$

(24)

$$X_{2j} = c_{2j} \sum_{V_k \in \mathbf{U}_a} m_{jk} \epsilon_{V_k} + e_{X_{2j}} + e'_{X_{2j}} = c_{2j} \sum_{V_k \in \mathbf{U}_a \setminus \{V_i\}} m_{jk} \epsilon_{V_k} + e_{X_{2j}} + \underbrace{e'_{X_{2j}} + c_{2j} m_{ji} \epsilon_{V_i}}_{\text{updated } e'_{X_{2j}}}, \quad (25)$$

so Cond. 2(2) is still valid. □

### C.2.3 PROOF OF THEORETICAL RESULTS IN SEC. 3.3

**Theorem 7.** Suppose the observed variables are generated by a LiNGAM with latent variables satisfying the bottleneck-faithfulness assumption and Asmp. 1, in the limit of infinite data, our algorithm correctly identifies the underlying complete causal structure.

*Proof.* In Stage 1, our algorithm sequentially identifies latent variables and their pure children. During this process, $\mathcal{H}_1$ records all identified causal relations. According to the theoretical results in Sec. 3.1, causal relations in $\mathcal{H}_1$ are correct. In Stage 2, with $\mathcal{H}_1$ fixed, our algorithm recovers $\mathcal{H}_2$. According to the theoretical results in Sec. 3.2, causal relations in $\mathcal{H}_2$ are correctly revealed. Combining $\mathcal{H}_1$ and $\mathcal{H}_2$, our algorithm correctly identifies the underlying complete causal structure. □

### C.3 PROOF OF THEORETICAL RESULTS IN SEC. 4

**Definition 5.** (Paired pseudo-pure children) We say $\{V_2, V_3\}$ is a pair of pseudo-pure children of $V_1$, denoted by $\{V_2, V_3\} \in P^3Ch(V_1)$, if (i) $\bigcap Pa(\{V_2, V_3\}) = \{V_1\}$, (ii) $\bigcup Pa(\{V_2, V_3\}) \setminus \{V_1, V_2, V_3\} = \emptyset$, (iii) $V_2 \in Ne(V_3)$, and (iv) $\forall V \in \bigcup De(\{V_2, V_3\}) \setminus \{V_2, V_3\}, |Pa(V)| = 1$.

> **Remark.** Based on this definition, if $\{V_2, V_3\} \in P^3Ch(V_1)$ and $V_2 \in Pa(V_3)$, $Ch(V_2) = PCh(V_2) \cup \{V_3\}$ and $Ch(V_3) = PCh(V_3)$.

**Assumption 2.** (1) $\forall L \in \mathbf{L}, V \in \mathbf{V}_0 \setminus \{L\}, Ch^{\mathcal{G}_0}(L) \not\subseteq Ch^{\mathcal{G}_0}(V) \cup \{V\}$. (2) $\forall V \in \mathbf{V}_0, Ch^{\mathcal{G}_0}(V)$ is the unique minimal bottleneck from $Ch^{\mathcal{G}_0}(V)$ to $\mathbf{O}_0$. (3) $\forall L \in \mathbf{L}, L$ is not a PV.

Trivially, if Asmp. 2 holds, then (1) $\forall L \in \mathbf{L}, V \in \mathbf{V} \setminus \{L\}, Ch^{\mathcal{G}}(L) \not\subseteq Ch^{\mathcal{G}}(V) \cup \{V\}$. (2) $\forall V \in \mathbf{V}, Ch^{\mathcal{G}}(V)$ is the unique minimal bottleneck from $Ch^{\mathcal{G}}(V)$ to $\mathbf{O}$. (3) $\forall L \in \mathbf{L}, L$ is not a PV.

### C.3.1 PROOF OF THEORETICAL RESULTS IN SEC. 4.1

**Definition 9.** *(Pseudo-pure descendant) We say $V_2$ is a pseudo-pure descendant of $V_1$, denoted by $V_2 \in P^2De(V_1)$, if $V_2 \in De(V_1)$ and there exists no common cause between $V_1$ and $V_2$.*

> **Example.** In Fig. 2(a), $P^2De(O_2) = \{L_1, ..., L_4, O_1, O_3, ..., O_{16}\}$.

**Condition 3.** (1) $\forall V \in \mathbf{V}_b, |Pa^{\mathcal{H}_1}(V)| = 1$ and $Ch^{\mathcal{H}_1}(V) = PCh^{\mathcal{G}}(V)$; (2) $\forall V \in \mathbf{V}_a, Pa^{\mathcal{H}_1}(V) = \emptyset$ and $|Ch^{\mathcal{H}_1}(V)| \geq 2$. If $Ch^{\mathcal{H}_1}(V) \not\subseteq PCh^{\mathcal{G}}(V)$, then $\{Ch^{\mathcal{H}_1}(V)\} = P^3Ch^{\mathcal{G}}(V)$ and $\nexists \{V_i, V_j\} \subseteq \mathbf{V}_a \setminus \{V\}$ s.t. $V_i \in P^2De^{\mathcal{G}}(V), V_j \not\perp V$, and $V_i \perp\!\!\!\perp V_j|V$.

Before proving theoretical results in the main text one by one, we first introduce two corollaries (Cors. 3 and 4) readily derived from Cond. 3.

**Corollary 3.** *(1) $\forall V \in \mathbf{V}_b$, (i) $Pa^{\mathcal{G}}(V) = Pa^{\mathcal{H}_1}(V)$ and $Ch^{\mathcal{G}}(V) = Ch^{\mathcal{H}_1}(V)$, or (ii) $\exists V' \in \mathbf{V}_b \setminus \{V\}$ s.t. $Pa^{\mathcal{G}}(V) = Pa^{\mathcal{H}_1}(V) \cup \{V'\}$ and $Ch^{\mathcal{G}}(V) = Ch^{\mathcal{H}_1}(V)$, or (iii) $Pa^{\mathcal{G}}(V) = Pa^{\mathcal{H}_1}(V)$ and $\exists V' \in \mathbf{V}_b \setminus \{V\}$ s.t. $Ch^{\mathcal{G}}(V) = Ch^{\mathcal{H}_1}(V) \cup \{V'\}$; (2) $\forall V \in \mathbf{V}_c, Ch^{\mathcal{G}}(V) = Ch^{\mathcal{H}_2}(V)$ and $Pa^{\mathcal{G}}(V) = Pa^{\mathcal{H}_2}(V)$; (3) $\forall V \in \mathbf{V}_a, Ch^{\mathcal{G}}(V) = Ch^{\mathcal{H}_1}(V) \cup Ch^{\mathcal{H}_2}(V)$ and $Pa^{\mathcal{G}}(V) = Pa^{\mathcal{H}_2}(V)$.*

**Remark.** This is a variant of Cor. 1 in App. C.2.1. While (2,3) here are identical to (2,3) in Cor. 1, (1) here is slightly different from (1) in Cor. 1 in the sense that for each $V \in \mathbf{V}_b$, $\mathrm{Ch}^{\mathcal{G}}(V)$ or $\mathrm{Pa}^{\mathcal{G}}(V)$ might contain one more variable in $\mathbf{V}_b$ than $\mathrm{Ch}^{\mathcal{H}_1}(V)$ or $\mathrm{Pa}^{\mathcal{H}_2}(V)$. This corollary is widely used in the following proofs. To maintain fluency, we will use it without further citation.

*Proof.* First, if $V_i \in \mathbf{V}_b$, then $\mathrm{Ch}^{\mathcal{H}_1}(V_i) = \mathrm{PCh}^{\mathcal{G}}(V_i)$ and there exists $V_j \in \mathbf{V}_a \cup \mathbf{V}_b$ s.t. $\mathrm{Pa}^{\mathcal{H}_1}(V_i) = \{V_j\}$ based on Cond. 3(1). Moreover, since (i) $\mathrm{Ch}^{\mathcal{H}_1}(V_j) = \mathrm{PCh}^{\mathcal{G}}(V_j)$ if $V_j \in \mathbf{V}_b$ based on Cond. 1(1) and (ii) $\mathrm{Ch}^{\mathcal{H}_1}(V_j) \subseteq \mathrm{PCh}^{\mathcal{G}}(V_j)$ or $\{\mathrm{Ch}^{\mathcal{H}_1}(V_j)\} = \mathrm{P}^3\mathrm{Ch}^{\mathcal{G}}(V_j)$ if $V_j \in \mathbf{V}_a$ based on Cond. 1(2), there is (i) $V_i \in \mathrm{PCh}^{\mathcal{G}}(V_j)$ or (ii) there exists $V_i' \in \mathbf{V}_b\backslash\{V_i\}$ s.t. $\mathrm{P}^3\mathrm{Ch}^{\mathcal{G}}(V_j) = \{\{V_i, V_i'\}\}$. Therefore, we can conclude that (i) $\mathrm{Ch}^{\mathcal{G}}(V_i) = \mathrm{PCh}^{\mathcal{G}}(V_i) = \mathrm{Ch}^{\mathcal{H}_1}(V_i)$ and $\mathrm{Pa}^{\mathcal{G}}(V_i) = \{V_j\} = \mathrm{Pa}^{\mathcal{H}_1}(V_i)$ if $V_i \in \mathrm{PCh}^{\mathcal{G}}(V_j)$; or (ii) $\mathrm{Ch}^{\mathcal{G}}(V_i) = \mathrm{PCh}^{\mathcal{G}}(V_i) = \mathrm{Ch}^{\mathcal{H}_1}(V_i)$ and $\mathrm{Pa}^{\mathcal{G}}(V_i) = \{V_j, V_i'\} = \mathrm{Pa}^{\mathcal{H}_1}(V_i) \cup \{V_i'\}$ if $\mathrm{P}^3\mathrm{Ch}^{\mathcal{G}}(V_j) = \{\{V_i, V_i'\}\}$ and $V_i' \in \mathrm{Pa}^{\mathcal{G}}(V_i)$; or (iii) $\mathrm{Ch}^{\mathcal{G}}(V_i) = \mathrm{PCh}^{\mathcal{G}}(V_i) \cup \{V_i'\} = \mathrm{Ch}^{\mathcal{H}_1}(V_i) \cup \{V_i'\}$ and $\mathrm{Pa}^{\mathcal{G}}(V_i) = \{V_j\} = \mathrm{Pa}^{\mathcal{H}_1}(V_i)$ if $\mathrm{P}^3\mathrm{Ch}^{\mathcal{G}}(V_j) = \{\{V_i, V_i'\}\}$ and $V_i' \in \mathrm{Ch}^{\mathcal{G}}(V_i)$. This completes the proof of (1).

The proofs for (2,3) are similar to Cor. 1. $\qquad\square$

**Corollary 4.** $\forall V \in \mathbf{V}_c$, $|\mathrm{De}^{\mathcal{H}_2}_{\mathbf{V}_a}(V)| \geq 2$. *If* $|\mathrm{De}^{\mathcal{H}_2}_{\mathbf{V}_a}(V)| = 2$, $\mathrm{De}^{\mathcal{H}_2}_{\mathbf{V}_a}(V) = \mathrm{Ch}^{\mathcal{G}}(V)$.

**Remark.** This is a variant of Cor. 2 in App. C.2.1, where "pure descendants" and "pure children" in Cor. 2 degenerate to "descendants" and "children" here. Although this is not sufficient to identify variables in $\mathbf{V}_c$, we can still infer some of their properties through analyzing variables in $\mathbf{V}_a$.

*Proof.* It is trivial that $\mathbf{V}_a$ is a bottleneck from $\mathbf{V}_a \cup \mathbf{V}_c$ to $\mathbf{V}_a \cup \mathbf{V}_b$, so for any $\mathbf{V}' \subseteq \mathbf{V}_a \cup \mathbf{V}_c$, $\bigcup \mathrm{GDe}^{\mathcal{G}}_{\mathbf{V}_a}(\mathbf{V}')$ is a bottleneck from $\mathbf{V}'$ to $\mathbf{O}$ given that $\mathbf{O} \subseteq \mathbf{V}_a \cup \mathbf{V}_b$. For any $V \in \mathbf{V}_c$, $\mathrm{Ch}^{\mathcal{G}}(V) = \mathrm{Ch}^{\mathcal{H}_2}(V) \subseteq \mathbf{V}_a \cup \mathbf{V}_c$, so $\mathrm{De}^{\mathcal{H}_2}_{\mathbf{V}_a}(V) = \bigcup \mathrm{GDe}^{\mathcal{G}}_{\mathbf{V}_a}(\mathrm{Ch}^{\mathcal{G}}(V))$ is a bottleneck from $\mathrm{Ch}^{\mathcal{G}}(V)$ to $\mathbf{O}$. Based on Asmp. 2(1,2), we have $|\mathrm{De}^{\mathcal{H}_2}_{\mathbf{V}_a}(V)| \geq |\mathrm{Ch}^{\mathcal{G}}(V)| \geq 2$. Furthermore, the first "$\geq$" becomes "$=$" iff $\mathrm{De}^{\mathcal{H}_2}_{\mathbf{V}_a}(V) = \mathrm{Ch}^{\mathcal{G}}(V)$ because of Asmp. 2(2). $\qquad\square$

**Theorem 8.** $\forall\{V_i, V_j\} \subseteq \mathbf{V}_a$, $\{V_i, V_j\} \in \mathbb{S}$ iff there exists $V_k \in \mathbf{V}_a\backslash\{V_i, V_j\}$ s.t. $\mathrm{Cov}(V_i, V_j)$ $\mathrm{Cov}(V_i, V_k)\mathrm{Cov}(V_j, V_k) \neq 0$ and for each such $V_k$, $\mathrm{R}(V_i, V_j|V_k) \perp\!\!\!\perp \mathbf{V}_a \backslash \{V_i, V_j\}$.

*Proof.* "Only if"

(1) Suppose $\{V_i, V_j\} \in \mathbb{S}_1$. The proof is similar to case (1) of "only if" part in proof of Thm. 1, except that we obtain $\mathrm{GDe}^{\mathcal{H}_2}_{\mathbf{V}_a}(V_k)\backslash\{V_i, V_j\} \neq \emptyset$ in a different way[6].

(2) Suppose $\{V_i, V_j\} \in \mathbb{S}_2$. The proof is similar to case (2) of "only if" part in proof of Thm. 1, except that we obtain $\mathrm{GDe}^{\mathcal{H}_2}_{\mathbf{V}_a}(V_l)\backslash\{V_i, V_j\} \neq \emptyset$ similarly to fn. 6.

(3) Suppose $\{V_i, V_j\} \in \mathbb{S}_3$. The proof is similar to case (3) of "only if" part in proof of Thm. 1, except that we obtain $\mathrm{GDe}^{\mathcal{H}_2}_{\mathbf{V}_a}(V_k)\backslash\{V_i, V_j\} \neq \emptyset$ in a different way[7].

"If". We prove this part by contradiction. Suppose $\{V_i, V_j\} \notin \mathbb{S}$.

---

[6]There are three possible cases. (i) If $V_k \in \mathbf{V}_a$, we have $V_k \in \mathrm{GDe}^{\mathcal{H}_2}_{\mathbf{V}_a}(V_k)\backslash\{V_i, V_j\} \neq \emptyset$. (ii) If $V_k \in \mathbf{V}_c$ and $\{V_i, V_j\} \nsubseteq \mathrm{De}^{\mathcal{H}_2}(V_k)$, based on Cor. 4, we have $\mathrm{GDe}^{\mathcal{H}_2}_{\mathbf{V}_a}(V_k)\backslash\{V_i, V_j\} \neq \emptyset$. (iii) If $V_k \in \mathbf{V}_c$ and $\{V_i, V_j\} \subseteq \mathrm{De}^{\mathcal{H}_2}(V_k)$, since $V_j \notin \mathrm{Ch}^{\mathcal{G}}(V_k)$, based on Cor. 4, we have $\mathrm{GDe}^{\mathcal{H}_2}_{\mathbf{V}_a}(V_k)\backslash\{V_i, V_j\} \neq \emptyset$.

[7]If $V_k \in \mathbf{V}_a$, then $V_k \in \mathrm{GDe}^{\mathcal{H}_2}_{\mathbf{V}_a}(V_k)\backslash\{V_i, V_j\} \neq \emptyset$. If $V_k \in \mathbf{V}_c \subseteq \mathbf{L}$, suppose $\mathrm{GDe}^{\mathcal{H}_2}_{\mathbf{V}_a}(V_k)\backslash\{V_i, V_j\} = \emptyset$, base on Cor. 4, $\mathrm{Ch}^{\mathcal{G}}(V_k) = \{V_i, V_j\}$, which leads to $\mathrm{Ch}^{\mathcal{G}}(V_k) \subseteq \mathrm{Ch}^{\mathcal{G}}(V_i) \cup \{V_i\}$, contradicting Asmp. 2(1).

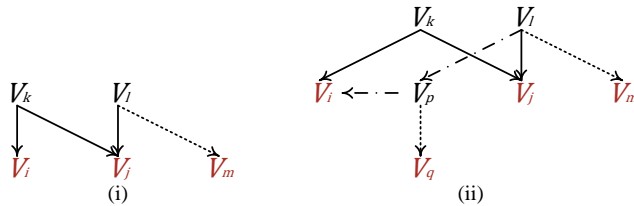

Figure 12: Illustration of case (1.b.ii) of "if" part in proof of Thm. 8. A dotted arrow from $V_1$ to $V_2$ means that $V_2 \in \mathrm{GDe}^{\mathcal{H}_2}(V_1)$. A dot-dash arrow from $V_1$ to $V_2$ means that $V_2 \in \mathrm{De}^{\mathcal{H}_2}(V_1)$. $V$ is marked in red if $V \in \mathbf{V}_a$.

(1) Suppose $V_i \notin \mathrm{Ne}^{\mathcal{H}_2}(V_j)$. All possible cases are as follows.

    (a) Suppose $\mathrm{Ch}^{\mathcal{H}_2}(V_i) \neq \emptyset$ or $\mathrm{Ch}^{\mathcal{H}_2}(V_j) \neq \emptyset$. The proof is similar to case (1.a) of "if" part in proof of Thm. 1, except that we obtain $\mathrm{GDe}^{\mathcal{H}_2}_{\mathbf{V}_a}(V_k)\backslash\{V_i, V_j\} \neq \emptyset$ similarly to fn. 6.

    (b) Suppose $\mathrm{Ch}^{\mathcal{H}_2}(V_i) = \mathrm{Ch}^{\mathcal{H}_2}(V_j) = \emptyset$, $\mathrm{Pa}^{\mathcal{H}_2}(V_i) \neq \emptyset$ and $\mathrm{Pa}^{\mathcal{H}_2}(V_j) \neq \emptyset$. Since $\{V_i, V_j\} \notin \mathbb{S}_2$, there are two possible cases.

       (i) Suppose $\mathrm{Pa}^{\mathcal{H}_2}(V_i) = \mathrm{Pa}^{\mathcal{H}_2}(V_j) = \{V_k\}$ where $V_k \in \mathbf{V}_c$ and $\mathrm{Ne}^{\mathcal{H}_2}(V_k)\backslash\{V_i, V_j\} = \emptyset$[8]. Clearly, for any $V \in \mathbf{V}_a\backslash\{V_i, V_j\}$, $\mathrm{Cov}(V, V_i) = 0$, which leads to contradiction.

       (ii) Suppose there exists $\{V_k, V_l\} \subseteq \mathbf{V}_a \cup \mathbf{V}_c\backslash\{V_i, V_j\}$ s.t. $V_k \in \mathrm{Pa}^{\mathcal{H}_2}(V_i)$ and $V_l \in \mathrm{Pa}^{\mathcal{H}_2}(V_j)$. If $\mathrm{GDe}^{\mathcal{H}_2}_{\mathbf{V}_a}(V_k)\backslash\{V_i, V_j\} \neq \emptyset$ and $\mathrm{GDe}^{\mathcal{H}_2}_{\mathbf{V}_a}(V_l)\backslash\{V_i, V_j\} \neq \emptyset$, the proof is similar to case (1.b) of "if" part in proof of Thm. 1. Otherwise, let $\mathrm{GDe}^{\mathcal{H}_2}_{\mathbf{V}_a}(V_k)\backslash\{V_i, V_j\} = \emptyset$ without loss of generality, then we have $V_k \in \mathbf{V}_c \subseteq \mathbf{L}$ and $\mathrm{Ch}^{\mathcal{G}}(V_k) = \{V_i, V_j\}$ based on Cor. 4. Also, based on Asmp. 2(1), $\mathrm{Ch}^{\mathcal{G}}(V_k) \not\subseteq \mathrm{Ch}^{\mathcal{G}}(V_l)$, so $V_i \notin \mathrm{Ch}^{\mathcal{H}_2}(V_l)$, we have $\mathrm{GDe}^{\mathcal{H}_2}_{\mathbf{V}_a}(V_l)\backslash\{V_i, V_j\} \neq \emptyset$ and let $V_m \in \mathrm{GDe}^{\mathcal{H}_2}_{\mathbf{V}_a}(V_l)\backslash\{V_i, V_j\}$. An illustrative example is shown as Fig. 12(i).

         • Suppose $V_i \notin \mathrm{De}^{\mathcal{H}_2}(V_l)$. As $m_{jl}m_{ml} \neq 0$ and $m_{il} = 0$, for any $V$ s.t. $V, V_i, V_j$ are correlated to each other, $\mathrm{R}(V_i, V_j|V) \not\perp V_m$, which leads to contradiction.

         • Suppose $V_i \in \mathrm{De}^{\mathcal{H}_2}(V_l)$, besides $V_i \notin \mathrm{Ch}^{\mathcal{H}_2}(V_l)$ as mentioned above, we can also derive $V_k \notin \mathrm{Ch}^{\mathcal{H}_2}(V_l)$[9], so there exists $V_p \neq V_k$ s.t. $V_p \in \mathrm{De}^{\mathcal{H}_2}(V_l)$ and $V_i \in \mathrm{De}^{\mathcal{H}_2}(V_p)$. Based on Asmp. 2(1), $\mathrm{Ch}^{\mathcal{G}}(V_k) \not\subseteq \mathrm{Ch}^{\mathcal{G}}(V_p)$, so we have $\mathrm{GDe}^{\mathcal{H}_2}_{\mathbf{V}_a}(V_p)\backslash\{V_i, V_j\} \neq \emptyset$ similarly to fn. 6. Let $V_q \in \mathrm{GDe}^{\mathcal{H}_2}_{\mathbf{V}_a}(V_p)\backslash\{V_i, V_j\}$ (It is possible that $V_m = V_q$). An illustrative example is shown as Fig. 12(ii). If $V_j \notin \mathrm{De}^{\mathcal{H}_2}(V_p)$, $m_{ip}m_{qp} \neq 0$ and $m_{jp} = 0$, so for any $V$ s.t. $V, V_i, V_j$ are correlated to each other, $\mathrm{R}(V_i, V_j|V) \not\perp V_q$, which leads to contradiction. Otherwise, $m_{il}m_{ip}m_{jl}m_{jp} \neq 0$. Since there exist two non-intersecting directed paths from $\{V_l, V_p\}$ to $\{V_i, V_j\}$ (e.g., $V_l \to V_j$ and $V_p \to \dots \to V_i$), $m_{il}/m_{ip} \neq m_{jl}/m_{jp}$. Also, $m_{ml}m_{qp} \neq 0$, so for any $V$ s.t. $V, V_i, V_j$ are correlated to each other, $\mathrm{R}(V_i, V_j|V) \not\perp V_m$ or $\mathrm{R}(V_i, V_j|V) \not\perp V_q$, which leads to contradiction.

    (c) Suppose $\mathrm{Ne}^{\mathcal{H}_2}(V_i) = \emptyset$ or $\mathrm{Ne}^{\mathcal{H}_2}(V_j) = \emptyset$. The proof is similar to case (1.c) of "if" part in proof of Thm. 1.

(2) Suppose $V_i \in \mathrm{Pa}^{\mathcal{H}_2}(V_j)$ or $V_j \in \mathrm{Pa}^{\mathcal{H}_2}(V_i)$, we take the former as an example without loss of generality. All possible cases are as follows.

    (a) Suppose $\mathrm{Ch}^{\mathcal{H}_2}(V_j) \neq \emptyset$. The proof is similar to case (2.a) of "if" part in proof of Thm. 1, except that we obtain $\mathrm{GDe}^{\mathcal{H}_2}_{\mathbf{V}_a}(V_k)\backslash\{V_i, V_j\} \neq \emptyset$ similarly to fn. 6.

---

[8]With Asmp. 1 valid, this case is impossible as $|\mathrm{Ne}^{\mathcal{G}}(V_k)| \geq 3$, so we do not discuss this case in the proof of Thm. 1.

[9]We can prove this by contradiction. Suppose $V_k \in \mathrm{Ch}^{\mathcal{H}_2}(V_l)$, then $\mathrm{Ch}^{\mathcal{G}}(V_l) \cup \{V_i\}\backslash\{V_k\}$ is a bottleneck from $\mathrm{Ch}^{\mathcal{G}}(V_l)$ to $\mathbf{O}$ and $|\mathrm{Ch}^{\mathcal{G}}(V_l)| = |\mathrm{Ch}^{\mathcal{G}}(V_l) \cup \{V_i\}\backslash\{V_k\}|$, contradicting Asmp. 2(2).

(b) Suppose $\text{Pa}^{\mathcal{H}_2}(V_j)\backslash\{V_i\} = \emptyset$ and $\text{Ch}^{\mathcal{H}_2}(V_j) = \emptyset$. The proof is similar to case (2.b) of "if" part in proof of Thm. 1.

(c) Suppose $\text{Ch}^{\mathcal{H}_2}(V_i)\backslash\{V_j\} \neq \emptyset$, $\text{Pa}^{\mathcal{H}_2}(V_j)\backslash\{V_i\} \neq \emptyset$, and $\text{Ch}^{\mathcal{H}_2}(V_j) = \emptyset$. The proof is similar to case (2.c) of "if" part in proof of Thm. 1, except that we obtain $\text{GDe}^{\mathcal{H}_2}_{\mathbf{V}_a}(V_k)\backslash\{V_i, V_j\} \neq \emptyset$ similarly to fn. 6 and $\text{GDe}^{\mathcal{H}_2}_{\mathbf{V}_a}(V_l)\backslash\{V_i, V_j\} \neq \emptyset$ similarly to fn. 7.

(d) Suppose $\text{Pa}^{\mathcal{H}_2}(V_i) \neq \emptyset$, $\text{Ch}^{\mathcal{H}_2}(V_i)\backslash\{V_j\} = \emptyset$, $\text{Pa}^{\mathcal{H}_2}(V_j)\backslash\{V_i\} \neq \emptyset$, and $\text{Ch}^{\mathcal{H}_2}(V_j) = \emptyset$. The proof is similar to case (2.d) of "if" part in proof of Thm. 1, except that we obtain both $\text{GDe}^{\mathcal{H}_2}_{\mathbf{V}_a}(V_k)\backslash\{V_i, V_j\} \neq \emptyset$ and $\text{GDe}^{\mathcal{H}_2}_{\mathbf{V}_a}(V_l)\backslash\{V_i, V_j\} \neq \emptyset$ similarly to fn. 7.

(e) Suppose $\text{Pa}^{\mathcal{H}_2}(V_i) = \emptyset$, $\text{Ch}^{\mathcal{H}_2}(V_i)\backslash\{V_j\} = \emptyset$, $\text{Pa}^{\mathcal{H}_2}(V_j)\backslash\{V_i\} \neq \emptyset$, and $\text{Ch}^{\mathcal{H}_2}(V_j) = \emptyset$. The proof is similar to case (2.e) of "if" part in proof of Thm. 1

$\square$

**Lemma 3.** $\forall\{V_i, V_j\} \in \mathbb{S}$, $\text{Ch}^{\mathcal{H}_1}(V_i) \subseteq \text{PCh}^{\mathcal{G}}(V_i)$ and $\text{Ch}^{\mathcal{H}_1}(V_j) \subseteq \text{PCh}^{\mathcal{G}}(V_j)$.

**Remark.** This lemma means that if a variable $V$ is in an identifiable pair, then there is $\text{Ch}^{\mathcal{H}_1}(V) \subseteq \text{PCh}^{\mathcal{G}}(V)$, which is consistent with the case where Asmp. 1 holds. This significantly simplifies the complexity of the proof of some following theoretical results. For instance, with this lemma, most proof strategies employed in the proof of Thm. 2 can be directly adapted to prove Thm. 9.

*Proof.* The proofs are as follows.

- Suppose $\{V_i, V_j\} \in \mathbb{S}_1$ and $V_i \in \text{Pa}^{\mathcal{G}}(V_j)$. First, we can easily derive that $\text{Ch}^{\mathcal{H}_1}(V_j) \subseteq \text{PCh}^{\mathcal{G}}(V_j)$[10]. Second, we suppose $\text{Ch}^{\mathcal{H}_1}(V_i) \nsubseteq \text{PCh}^{\mathcal{G}}(V_i)$ and let $V_k \in \text{Ne}^{\mathcal{H}_2}(V_i)\backslash\{V_j\}$. Similarly to fn. 6, we have $\text{GDe}^{\mathcal{H}_2}_{\mathbf{V}_a}(V_k)\backslash\{V_i, V_j\} \neq \emptyset$ and let $V_l \in \text{GDe}^{\mathcal{H}_2}_{\mathbf{V}_a}(V_k)\backslash\{V_i, V_j\}$. Clearly, there is $V_j \in \text{P}^2\text{De}^{\mathcal{G}}(V_i)$, $V_l \not\!\perp V_i$, and $V_j \perp\!\!\!\perp V_l|V_i$, contradicting Cond. 3(2).

- Suppose $\{V_i, V_j\} \in \mathbb{S}_2$, similarly to fn. 10, we can easily derive that $\text{Ch}^{\mathcal{H}_1}(V_i) \subseteq \text{PCh}^{\mathcal{G}}(V_i)$ and $\text{Ch}^{\mathcal{H}_1}(V_j) \subseteq \text{PCh}^{\mathcal{G}}(V_j)$.

- Suppose $\{V_i, V_j\} \in \mathbb{S}_3$ and $V_i \in \text{Pa}^{\mathcal{G}}(V_j)$. First, we can easily derive that $\text{Ch}^{\mathcal{H}_1}(V_j) \subseteq \text{PCh}^{\mathcal{G}}(V_j)$ similarly to fn. 10. Second, suppose $\text{Ch}^{\mathcal{H}_1}(V_i) \nsubseteq \text{PCh}^{\mathcal{G}}(V_i)$, based on Cond. 3(2), we have $\{\text{Ch}^{\mathcal{H}_1}(V_i)\} = \text{P}^3\text{Ch}^{\mathcal{G}}(V_i)$, so $V_i \notin \mathbf{O}$, that is, $V_i \in \mathbf{L}$. Clearly, $V_i$ is a I-PV. This leads to contradiction to Asmp. 2(3).

$\square$

**Theorem 9.** $\forall\{V_i, V_j\} \in \mathbb{S}$, let $\{V_{i_1}, V_{i_2}\} \subseteq \text{Ch}^{\mathcal{H}_1}(V_i)$.

(1) $\text{R}(V_{i_1}, V_j|V_{i_2}) \perp\!\!\!\perp V_{i_2}$ iff $\{V_i, V_j\} \in \mathbb{S}_1$ and $V_i \in \text{Pa}^{\mathcal{G}}(V_j)$.

(2) Suppose $\{V_i, V_j\} \notin \mathbb{S}_1$, $\exists\{V_{i'}, V_{j'}\} \in \mathbb{S}\backslash\{\{V_i, V_j\}\}$ s.t. $\{V_{i'}, V_{j'}\} \cap \{V_i, V_j\} \neq \emptyset$ only if (but not if) $\{V_i, V_j\} \in \mathbb{S}_2$.

(3) Suppose $\{V_i, V_j\} \notin \mathbb{S}_1$, $\exists\{V_k, V_l\} \subseteq \bigcup\text{Ch}^{\mathcal{H}_1}(\mathbf{V}_a\backslash\{V_i, V_j\})$ s.t. $(V_{i_1}, V_{i_2}, V_j, V_k, V_l)$ satisfies the quintuple constraint only if (but not if) $\{V_i, V_j\} \in \mathbb{S}_3$ and $V_i \in \text{Pa}^{\mathcal{G}}(V_j)$

*Proof.* Combined with Lem. 3, the proofs of (1), (2), and "only if" part of (3) are similar to Thm. 2. Here we focus on "not if" part of (3).

---

[10]Suppose $\text{Ch}^{\mathcal{H}_1}(V_j) \nsubseteq \text{PCh}^{\mathcal{G}}(V_j)$, based on Cond. 3(2), we have $\{\text{Ch}^{\mathcal{H}_1}(V_j)\} = \text{P}^3\text{Ch}^{\mathcal{G}}(V_j)$, so $V_j \notin \mathbf{O}$, that is, $V_j \in \mathbf{L}$. Let $\text{Ch}^{\mathcal{H}_1}(V_j) = \{V_k, V_l\}$ where $V_k \in \text{Pa}^{\mathcal{G}}(V_l)$, since $\text{Ch}^{\mathcal{H}_2}(V_j) = \emptyset$, we have $\text{Ch}^{\mathcal{G}}(V_j) = \{V_k, V_l\}$. There is $\text{Ch}^{\mathcal{G}}(V_j) \subseteq \{V_k\} \cup \text{Ch}^{\mathcal{G}}(V_k)$, contradicting Asmp. 2(1).

Suppose $\mathcal{G}_0$ is shown as Fig. 8(c), at the first iteration when $\mathbf{V}_a = \{O_2, ..., O_6\}$, $\{O_2, O_3\} \in \mathbb{S}_3$ and $O_2 \in \mathrm{Pa}^{\mathcal{G}}(O_3)$. Let $\{O_{2_1}, O_{2_2}\} \subseteq \mathrm{Ch}^{\mathcal{H}_1}(O_2)$, then $O_{2_1}, O_{2_2}, O_3$ can be expressed as

$$O_{2_1} = a_{21}a_{2_12}L_1 + a_{2_12}\epsilon_{O_2} + \epsilon_{O_{2_1}}, \tag{26}$$

$$O_{2_2} = a_{21}a_{2_22}L_1 + a_{2_22}\epsilon_{O_2} + \epsilon_{O_{2_2}}, \tag{27}$$

$$O_3 = (a_{21}a_{32} + a_{31})L_1 + a_{32}\epsilon_{O_2} + \epsilon_{O_3}. \tag{28}$$

Because $a_{31} \neq 0$, $a_{21}a_{2_12}/(a_{21}a_{32} + a_{31}) \neq a_{2_12}/a_{32}$. For any $\{O_i, O_j\} \subseteq \bigcup \mathrm{Ch}^{\mathcal{H}_1}(\mathbf{V}_a \backslash \{O_2, O_3\})$, they can be expressed as

$$O_i = \lambda_i L_1 + (\gamma_i \epsilon_{O_4} + e_i'), \quad O_j = \lambda_j L_1 + (\gamma_j \epsilon_{O_4} + e_j'), \tag{29}$$

where $L_1, \epsilon_{O_2}, \epsilon_{O_{2_1}}, \epsilon_{O_{2_2}}, \epsilon_{O_3}, \gamma_i \epsilon_{O_4} + e_i'$ are independent of each other, $O_j \perp\!\!\!\perp (\epsilon_{O_3}, \epsilon_{O_{2_1}}, \epsilon_{O_{2_2}}, \epsilon_{O_3})$ and $\mathrm{Cov}(O_j, L_1)\mathrm{Cov}(O_j, \gamma_i \epsilon_{O_4} + e_i') \neq 0$. Based on Lem. 2(2.b), $(O_{2_1}, O_{2_2}, O_3, O_i, O_j)$ does not satisfy the quintuple constraint. $\qquad \square$

**Corollary 5.** *Given* $\{V_i, V_j\} \in \mathbb{S}_3$ *and* $V_i \in \mathrm{Pa}^{\mathcal{G}}(V_j)$, *let* $\mathrm{Pa}^{\mathcal{H}_2}(V_i) = \{V_h\}$ *and* $\{V_{i_1}, V_{i_2}\} \subseteq \mathrm{Ch}^{\mathcal{H}_1}(V_i)$.

*(1) If* $V_h \in \mathbf{V}_a$, *then* $\exists \{V_k, V_l\} \subseteq \mathrm{Ch}^{\mathcal{H}_1}(\mathbf{V}_a \backslash \{V_i, V_j\})$ *s.t.* $(V_{i_1}, V_{i_2}, V_j, V_k, V_l)$ *satisfies the quintuple constraint.*

*(2) If* $\exists \{V_m, V_n\} \subseteq \mathbf{V}_a \backslash \{V_i, V_j\}$ *s.t.* $V_m \in \mathrm{P}^2\mathrm{De}^{\mathcal{G}}(V_h)$, $V_n \not\perp\!\!\!\perp V_h$, *and* $V_m \perp\!\!\!\perp V_n | V_h$, *then* $\exists \{V_k, V_l\} \subseteq \mathrm{Ch}^{\mathcal{H}_1}(\mathbf{V}_a \backslash \{V_i, V_j\})$ *s.t.* $(V_{i_1}, V_{i_2}, V_j, V_k, V_l)$ *satisfies the quintuple constraint.*

> **Remark.** Given $\{V_i, V_j\} \in \mathbb{S}_3$ and $V_i \in \mathrm{Pa}^{\mathcal{G}}(V_j)$, based on Rem. of Thm. 9 in the main text, both $\{V_i, V_j\} \in \tilde{\mathbb{S}}_2$ and $\{V_i, V_j\} \in \tilde{\mathbb{S}}_3$ are possible. This corollary provides two sufficient conditions that $\exists \{V_k, V_l\} \subseteq \mathrm{Ch}^{\mathcal{H}_1}(\mathbf{V}_a \backslash \{V_i, V_j\})$ s.t. $(V_{i_1}, V_{i_2}, V_j, V_k, V_l)$ satisfies the quintuple constraint, that is, $\{V_i, V_j\} \in \tilde{\mathbb{S}}_3$. The proof of the following Thm. 10 highly relies on this corollary.

*Proof.* The proofs are as follows.

(1) We first prove $\mathrm{Ch}^{\mathcal{H}_1}(V_h) \subseteq \mathrm{PCh}^{\mathcal{G}}(V_h)$ by contradiction. Suppose $\mathrm{Ch}^{\mathcal{H}_1}(V_h) \not\subseteq \mathrm{PCh}^{\mathcal{G}}(V_h)$, then based on Cond. 3(2), $\{\mathrm{Ch}^{\mathcal{H}_1}(V_h)\} = \mathrm{P}^3\mathrm{Ch}^{\mathcal{G}}(V_h)$. In addition, since $\{V_i, V_j\} \in \mathbb{S}_3$, we have $\mathrm{Ch}^{\mathcal{H}_1}(V_i) \subseteq \mathrm{PCh}^{\mathcal{G}}(V_i)$ and $\mathrm{Ch}^{\mathcal{H}_1}(V_j) \subseteq \mathrm{PCh}^{\mathcal{G}}(V_j)$ based on Lem. 3, so $\{V_i, V_j\} \in \mathrm{P}^3\mathrm{Ch}^{\mathcal{G}}(V_h) = \{\mathrm{Ch}^{\mathcal{H}_1}(V_h)\} \subseteq \mathbf{V}_b$, which leads to contradiction, thus $\mathrm{Ch}^{\mathcal{H}_1}(V_h) \subseteq \mathrm{PCh}^{\mathcal{G}}(V_h)$. Let $\{V_k, V_l\} \subseteq \mathrm{Ch}^{\mathcal{H}_1}(V_h)$, then the proof is similar to "'if' of (3)" part in proof of Thm. 2.

(2) Let $V_k \in \mathrm{Ch}^{\mathcal{H}_1}(V_m)$ and $V_l \in \mathrm{Ch}^{\mathcal{H}_1}(V_n)$, then the proof is similar to "'if' of (3)" part in proof of Thm. 2.

$\qquad \square$

**Theorem 10.** *(1)* $\forall \{V_i, V_j\} \in \tilde{\mathbb{S}}_2$, $\bigcap \mathrm{Pa}^{\mathcal{G}}(\{V_i, V_j\}) \subseteq \mathbf{V}_a$ *iff* $\exists \{V_{i'}, V_{j'}\} \in \tilde{\mathbb{S}}_1$ *s.t.* $\{V_i, V_j\} \cap \{V_{i'}, V_{j'}\} \neq \emptyset$. *(2)* $\forall \{\{V_i, V_j\}, \{V_{i'}, V_{j'}\}\} \subseteq \tilde{\mathbb{S}}_2$, $\bigcap \mathrm{Pa}^{\mathcal{G}}(\{V_i, V_j\}) = \bigcap \mathrm{Pa}^{\mathcal{G}}(\{V_{i'}, V_{j'}\})$ *iff* $\exists \{V_{i''}, V_{j''}\} \in \tilde{\mathbb{S}}_2$ *s.t.* $\{V_i, V_j\} \cap \{V_{i''}, V_{j''}\} \neq \emptyset$ *and* $\{V_{i'}, V_{j'}\} \cap \{V_{i''}, V_{j''}\} \neq \emptyset$.

*Proof.* As mentioned in Rem. of Thm. 9 in the main text, $\tilde{\mathbb{S}}_1 = \mathbb{S}_1$, $\tilde{\mathbb{S}}_2 \supseteq \mathbb{S}_2$, $\tilde{\mathbb{S}}_3 \subseteq \mathbb{S}_3$.

- For any $\{V_i, V_j\} \in \mathbb{S}_2 \subseteq \tilde{\mathbb{S}}_2$, (1) can be derived from the definitions of $\mathbb{S}_1$ and $\mathbb{S}_2$.

- For $\{V_i, V_j\} \in \tilde{\mathbb{S}}_2 \backslash \mathbb{S}_2$, the proof of (1) is as follows. Clearly, $\{V_i, V_j\} \in \mathbb{S}_3$. First, based on the definition of $\mathbb{S}_1$, $\forall \{V_{i'}, V_{j'}\} \in \tilde{\mathbb{S}}_1 = \mathbb{S}_1$, $\{V_i, V_j\} \cap \{V_{i'}, V_{j'}\} = \emptyset$. Second, we have $\bigcap \mathrm{Pa}^{\mathcal{H}_2}(\{V_i, V_j\}) \not\subseteq \mathbf{V}_a$, otherwise $\{V_i, V_j\} \in \tilde{\mathbb{S}}_3$ based on Cor. 5(1), which leads to contradiction. This finishes the proof.

- For any $\{\{V_i, V_j\}, \{V_{i'}, V_{j'}\}\} \subseteq \mathbb{S}_2 \subseteq \tilde{\mathbb{S}}_2$, (2) can be derived from the definition of $\mathbb{S}_2$.

- For $\{V_i, V_j\} \in \tilde{\mathbb{S}}_2 \backslash \mathbb{S}_2$, the proof of (2) is as follows. Clearly, $\{V_i, V_j\} \in \mathbb{S}_3$. First, based on the definitions of $\mathbb{S}_2$ and $\mathbb{S}_3$, $\forall \{V_{i'}, V_{j'}\} \in \tilde{\mathbb{S}}_2 \backslash \{\{V_i, V_j\}\} \subseteq (\mathbb{S}_2 \cup \mathbb{S}_3) \backslash \{\{V_i, V_j\}\}$, $\{V_i, V_j\} \cap \{V_{i'}, V_{j'}\} = \emptyset$. Second, we prove $\forall \{V_{i'}, V_{j'}\} \in \tilde{\mathbb{S}}_2 \backslash \{\{V_i, V_j\}\}$, $\bigcap \mathrm{Pa}^{\mathcal{H}_2}(\{V_i, V_j\}) \neq \bigcap \mathrm{Pa}^{\mathcal{H}_2}(\{V_{i'}, V_{j'}\})$ by contradiction. Let $\bigcap \mathrm{Pa}^{\mathcal{H}_2}(\{V_i, V_j\}) = \{V_h\}$ and suppose $\bigcap \mathrm{Pa}^{\mathcal{H}_2}(\{V_{i'}, V_{j'}\}) = \{V_h\}$. First, we have $V_h \in \mathbf{V}_c \subseteq \mathbf{L}$, otherwise $\{V_i, V_j\} \in \tilde{\mathbb{S}}_3$ based on Cor. 5(1), which leads to contradiction. Second, we also have $\{V_{i'}, V_{j'}\} \in \mathbb{S}_3$, otherwise $V_{i'} \in \mathrm{P}^2\mathrm{De}^{\mathcal{G}}(V_h)$, $V_{j'} \not\perp V_h$, and $V_{i'} \perp\!\!\!\perp V_{j'} | V_h$, so $\{V_i, V_j\} \in \tilde{\mathbb{S}}_3$ based on Cor. 5(2), which leads to contradiction. Third, because $V_h$ is not a II-PV based on Asmp. 2(3), there exists $V_k \in \mathrm{Ne}^{\mathcal{H}_2}(V_h) \backslash \{V_i, V_j, V_{i'}, V_{j'}\}$ and we can derive $\mathrm{GDe}_{\mathbf{V}_a}^{\mathcal{H}_2}(V_k) \backslash \bigcup \mathrm{GDe}_{\mathbf{V}_a}^{\mathcal{H}_2}(\{V_i, V_j, V_{i'}, V_{j'}\}) \neq \emptyset$[11]. Let $V_m \in \{V_{i'}, V_{j'}\}$ and $V_n \in \mathrm{GDe}_{\mathbf{V}_a}^{\mathcal{H}_2}(V_k) \backslash \bigcup \mathrm{GDe}_{\mathbf{V}_a}^{\mathcal{H}_2}(\{V_i, V_j, V_{i'}, V_{j'}\})$, since $V_m \in \mathrm{P}^2\mathrm{De}^{\mathcal{G}}(V_k)$, $V_n \not\perp V_k$, and $V_m \perp\!\!\!\perp V_n | V_k$, $\{V_i, V_j\} \in \tilde{\mathbb{S}}_3$ based on Cor. 5(2), which leads to contradiction.

$\square$

**Proposition 4.** *Cond. 3 is still valid after update.*

*Proof.* We denote $\mathbf{V}_b, \mathbf{V}_a, \mathbf{V}_c, \mathcal{H}_1, \mathcal{H}_2, \mathbb{S}$ after update by $\mathbf{V}_b', \mathbf{V}_a', \mathbf{V}_c', \mathcal{H}_1', \mathcal{H}_2', \mathbb{S}'$ respectively. While Cond. 3 is valid for $\mathbf{V}_b' \cap \mathbf{V}_b$ and $\mathbf{V}_a' \cap \mathbf{V}_a$ trivially, we focus on $\mathbf{V}_b' \backslash \mathbf{V}_b$ and $\mathbf{V}_a' \backslash \mathbf{V}_a$.

- If $V_j \in \mathbf{V}_b' \backslash \mathbf{V}_b$, then $V_j \in \mathbf{V}_a$ and there exists $V_i \in \mathbf{V}_a$ s.t. (1) $\{V_i, V_j\} \in \tilde{\mathbb{S}}_1$ and $V_i \in \mathrm{Pa}^{\mathcal{H}_2}(V_j)$ or (2) $\{V_i, V_j\} \in \tilde{\mathbb{S}}_2$. Based on Lem. 3, $\mathrm{Ch}^{\mathcal{H}_1}(V_j) \subseteq \mathrm{PCh}^{\mathcal{G}}(V_j)$, then the proof is similar to part 1 in proof of Prop. 2.

- If $V_j \in \mathbf{V}_a' \backslash \mathbf{V}_a$, then $V_j \in \mathbf{V}_c$ and $\forall \{V_k, V_l\} \subseteq \mathrm{Ch}^{\mathcal{H}_1'}(V_j)$, $\{V_k, V_l\} \in \tilde{\mathbb{S}}_2$ and $\bigcap \mathrm{Pa}^{\mathcal{H}_2}(\{V_k, V_l\}) = \{V_j\}$. Also, $\forall V \in \mathrm{Ch}^{\mathcal{H}_1'}(V_j)$, $\mathrm{Ch}^{\mathcal{H}_1}(V) \subseteq \mathrm{PCh}^{\mathcal{G}}(V)$ based on Lem. 3. Furthermore, based on Thm. 10(2), there are two possible cases.

  - $\forall \{V_k, V_l\} \subseteq \mathrm{Ch}^{\mathcal{H}_1'}(V_j)$, $\{V_k, V_l\} \in \mathbb{S}_2$. In this case, it is trivial that $\mathrm{Ch}^{\mathcal{H}_1'}(V_j) \subseteq \mathrm{PCh}^{\mathcal{G}}(V_j)$.

  - $\mathrm{Ch}^{\mathcal{H}_1'}(V_j) \in \mathbb{S}_3$. In this case, it is trivial that $\mathrm{Ch}^{\mathcal{H}_1'}(V_j) \in \mathrm{P}^3\mathrm{Ch}^{\mathcal{G}}(V_j)$. Now we prove $\{\mathrm{Ch}^{\mathcal{H}_1'}(V_j)\} = \mathrm{P}^3\mathrm{Ch}^{\mathcal{G}}(V_j)$ by contradiction. Let $\mathrm{Ch}^{\mathcal{H}_1'}(V_j) = \{V_{j_1}, V_{j_2}\}$ and suppose there exists $\{V_k, V_l\} \in \mathrm{P}^3\mathrm{Ch}^{\mathcal{G}}(V_j) \backslash \{\{V_{j_1}, V_{j_2}\}\}$. As $\{V_{j_1}, V_{j_2}\} \cap \mathrm{De}^{\mathcal{H}_2}(V_k) = \emptyset$, we have $\mathrm{GDe}_{\mathbf{V}_a}^{\mathcal{H}_2}(V_k) \backslash \{V_{j_1}, V_{j_2}\} \neq \emptyset$ and let $V_m \in \mathrm{GDe}_{\mathbf{V}_a}^{\mathcal{H}_2}(V_k) \backslash \{V_{j_1}, V_{j_2}\}$. Because $V_j$ is not a II-PV based on Asmp. 2(3), there exists $V_i \in \mathrm{Ne}^{\mathcal{H}_2}(V_j) \backslash \{V_{j_1}, V_{j_2}, V_k, V_l\}$. Also, there exists $V_n \in \mathrm{GDe}_{\mathbf{V}_a}^{\mathcal{H}_2}(V_i) \backslash \bigcup \mathrm{GDe}_{\mathbf{V}_a}^{\mathcal{H}_2}(\{V_{j_1}, V_{j_2}, V_k, V_l\}) \neq \emptyset$ similarly to fn. 11. Clearly, $V_m \in \mathrm{P}^2\mathrm{De}^{\mathcal{G}}(V_j)$, $V_n \not\perp V_j$, and $V_m \perp\!\!\!\perp V_n | V_j$, so $\{V_{j_1}, V_{j_2}\} \in \tilde{\mathbb{S}}_3$ based on Cor. 5(2), which leads to contradiction. Therefore, $\{\mathrm{Ch}^{\mathcal{H}_1'}(V_j)\} = \mathrm{P}^3\mathrm{Ch}^{\mathcal{G}}(V_j)$. Likewise, we can conclude that $\nexists \{V_k, V_l\} \subseteq \mathbf{V}_a' \backslash \{V_j\}$ s.t. $V_k \in \mathrm{P}^2\mathrm{De}^{\mathcal{G}}(V_j)$, $V_l \not\perp V_j$, and $V_k \perp\!\!\!\perp V_l | V_j$, otherwise there is also $\{V_{j_1}, V_{j_2}\} \in \tilde{\mathbb{S}}_3$, which leads to contradiction.

  Finally, it is trivial that $\mathrm{Pa}^{\mathcal{H}_1'}(V_j) = \emptyset$.

$\square$

---

[11] There are three possible cases. (i) If $V_k \in \mathbf{V}_a$, then $V_k \in \mathrm{GDe}_{\mathbf{V}_a}^{\mathcal{H}_2}(V_k) \backslash \bigcup \mathrm{GDe}_{\mathbf{V}_a}^{\mathcal{H}_2}(\{V_i, V_j, V_{i'}, V_{j'}\}) \neq \emptyset$. (ii) If $V_k \in \mathbf{V}_c$ and $V_k \in \mathrm{Ch}^{\mathcal{H}_2}(V_h)$, then $\mathrm{De}^{\mathcal{H}_2}(V_k) \cap \bigcup \mathrm{GDe}_{\mathbf{V}_a}^{\mathcal{H}_2}(\{V_i, V_j, V_{i'}, V_{j'}\}) = \emptyset$, so $\mathrm{GDe}_{\mathbf{V}_a}^{\mathcal{H}_2}(V_k) \backslash \bigcup \mathrm{GDe}_{\mathbf{V}_a}^{\mathcal{H}_2}(\{V_i, V_j, V_{i'}, V_{j'}\}) \neq \emptyset$ based on Cor. 4. (iii) If $V_k \in \mathbf{V}_c$ and $V_k \in \mathrm{Pa}^{\mathcal{H}_2}(V_h)$, then $(\mathrm{De}_{\mathbf{V}_a}^{\mathcal{H}_2}(V_k) \cup \{V_h\}) \backslash \bigcup \mathrm{GDe}_{\mathbf{V}_a}^{\mathcal{H}_2}(\{V_i, V_j, V_{i'}, V_{j'}\})$ is a bottleneck from $\mathrm{Ch}^{\mathcal{G}}(V_k)$ to $\mathbf{O}$, so $|(\mathrm{De}_{\mathbf{V}_a}^{\mathcal{H}_2}(V_k) \cup \{V_h\}) \backslash \bigcup \mathrm{GDe}_{\mathbf{V}_a}^{\mathcal{H}_2}(\{V_i, V_j, V_{i'}, V_{j'}\})| \geq |\mathrm{Ch}^{\mathcal{G}}(V_k)| \geq 2$ based on Asmp. 2(1,2), so $\mathrm{GDe}_{\mathbf{V}_a}^{\mathcal{H}_2}(V_k) \backslash \bigcup \mathrm{GDe}_{\mathbf{V}_a}^{\mathcal{H}_2}(\{V_i, V_j, V_{i'}, V_{j'}\}) \neq \emptyset$.

**Theorem 11.** If Asmp. 1 is invalid, when $\tilde{\mathbb{S}}_1 \cup \tilde{\mathbb{S}}_2 = \emptyset$, $\mathbf{V}_c \neq \emptyset$ or there exists $L \in \mathbf{V}_a$ s.t. $\mathrm{Ch}^{\mathcal{H}_1}(L) \nsubseteq \mathrm{PCh}^{\mathcal{G}}(L)$.

*Proof.* The proofs are as follows.

- Suppose $\exists L_i \in \mathbf{L}$ s.t. $|\mathrm{PCh}^{\mathcal{G}}(L_i)| < 2$. If $\mathrm{P}^3\mathrm{Ch}^{\mathcal{G}}(L_i) = \emptyset$, there is $L_i \in \mathbf{V}_c$ trivially. Otherwise,

    - It is possible that $L_i \in \mathbf{V}_c$, e.g., if no pair in $\mathrm{P}^3\mathrm{Ch}^{\mathcal{G}}(L_i)$ is incorporated into $\tilde{\mathbb{S}}_2$.
    - It is possible that $L_i \in \mathbf{V}_a$, an example is shown as Fig. 8(d). In this case, there must be $\mathrm{Ch}^{\mathcal{H}_1}(L_i) \nsubseteq \mathrm{PCh}^{\mathcal{G}}(L_i)$ because $|\mathrm{PCh}^{\mathcal{G}}(L_i)| < 2$ but $|\mathrm{Ch}^{\mathcal{H}_1}(L_i)| \geq 2$ based on Cond. 3(2).
    - It is impossible that $L_i \in \mathbf{V}_b$. Suppose $L_i \in \mathbf{V}_b$, it is trivial that $\mathrm{Ch}^{\mathcal{H}_1}(L_i) \nsubseteq \mathrm{PCh}^{\mathcal{G}}(L_i)$, which leads to contradiction to Cond. 3(1).

- Suppose $\forall L \in \mathbf{L}$, $|\mathrm{PCh}^{\mathcal{G}}(L)| \geq 2$. Since Asmp. 1 is invalid, $\exists L_i \in \mathbf{L}$ s.t. $|\mathrm{Ne}^{\mathcal{G}}(L_i)| < 3$, that is, $|\mathrm{PCh}^{\mathcal{G}}(L_i)| = 2$ and $\mathrm{Ne}^{\mathcal{G}}(L_i) \backslash \mathrm{PCh}^{\mathcal{G}}(L_i) = \emptyset$. According to the definition of $\mathbb{S}$, there is always $\mathrm{PCh}^{\mathcal{G}}(L_i) \notin \mathbb{S}$, so there is always $L_i \in \mathbf{V}_c$.

$\square$

**Corollary 6.** *If Asmp. 1 is invalid, when $\tilde{\mathbb{S}}_1 \cup \tilde{\mathbb{S}}_2 = \emptyset$, (1) $\forall V_i \in \mathbf{V}_c$, $|\mathrm{De}_{\mathbf{V}_a}^{\mathcal{H}_2}(V_i)| \geq 2$. (2) $\forall V_i \in \mathbf{V}_a$ s.t. $\mathrm{Ch}^{\mathcal{H}_1}(V_i) \nsubseteq \mathrm{PCh}^{\mathcal{G}}(V_i)$, $|\mathrm{De}_{\mathbf{V}_a}^{\mathcal{H}_2}(V_i)| \geq 1$.*

> **Remark.** This lemma means that at the end of stage 1, (1) for any $V_i \in \mathbf{V}_a$, $V_i$ has at least two descendants in $\mathbf{V}_a$ and (2) for any $\forall V_i \in \mathbf{V}_a$ s.t. $\mathrm{Ch}^{\mathcal{H}_1}(V_i) \nsubseteq \mathrm{PCh}^{\mathcal{G}}(V_i)$, $V_i$ has at least one descendant in $\mathbf{V}_a$. This corollary is important for the proofs of the following Thms. 12 and 13.

*Proof.* The proofs are as follows.

(1) This directly follows Cor. 4.

(2) Based on Cond. 3(2), if $\mathrm{Ch}^{\mathcal{H}_1}(V_i) \nsubseteq \mathrm{PCh}^{\mathcal{G}}(V_i)$, then $\{\mathrm{Ch}^{\mathcal{H}_1}(V_i)\} = \mathrm{P}^3\mathrm{Ch}^{\mathcal{G}}(V_i)$, so $V_i \notin \mathbf{O}$, that is, $V_i \in \mathbf{L}$. Let $\mathrm{Ch}^{\mathcal{H}_1}(V_i) = \{V_{i_1}, V_{i_2}\}$ and $V_{i_1} \in \mathrm{Pa}^{\mathcal{G}}(V_{i_2})$, we can prove $\mathrm{Ch}^{\mathcal{H}_2}(V_i) \neq \emptyset$ by contradiction. Specifically, suppose $\mathrm{Ch}^{\mathcal{H}_2}(V_i) = \emptyset$, then $\mathrm{Ch}^{\mathcal{G}}(V_i) = \{V_{i_1}, V_{i_2}\}$, note that $\mathrm{Ch}^{\mathcal{G}}(V_i) \subseteq \mathrm{Ch}^{\mathcal{G}}(V_{i_1}) \cup \{V_{i_1}\}$, this leads to contradiction to Asmp. 2(1). Let $V_j \in \mathrm{Ch}^{\mathcal{H}_2}(V_i)$, it is trivial that $\mathrm{GDe}_{\mathbf{V}_a}^{\mathcal{H}_2}(V_j) \neq \emptyset$, so $\mathrm{De}_{\mathbf{V}_a}^{\mathcal{H}_2}(V_i) \neq \emptyset$.

$\square$

### C.3.2 PROOF OF THEORETICAL RESULTS IN SEC. 4.2

**Condition 4.** (1) $\forall V \in \mathbf{V}_a \backslash \mathbf{U}_a$, $\mathrm{Ch}^{\mathcal{H}_1}(V_i) \subseteq \mathrm{PCh}^{\mathcal{G}}(V_i)$. (2) $\forall V \in \mathbf{U}_a \cup \mathbf{V}_c$, $\mathrm{De}^{\mathcal{H}_2}(V) \subseteq \mathbf{U}_a \cup \mathbf{V}_c$. (3) $\forall V_i \in \mathbf{U}_a$, $X_{2i-1}, X_{2i}$ can be written as

$$X_{2i-1} = c_{2i-1} \sum_{V_j \in \mathbf{U}_a \cup \mathbf{V}_c} m_{ij}\epsilon_{V_j} + e_{X_{2i-1}} + e'_{X_{2i-1}}, \quad X_{2i} = c_{2i} \sum_{V_j \in \mathbf{U}_a \cup \mathbf{V}_c} m_{ij}\epsilon_{V_j} + e_{X_{2i}} + e'_{X_{2i}}, \quad (30)$$

where $\epsilon_{V_1}, ..., \epsilon_{V_n}, (e_{X_1}, e_{X_2})..., (e_{X_{2n-1}}, e_{X_{2n}}), (e'_{X_1}, e'_{X_3}, ..., e'_{X_{2n-1}}), (e'_{X_2}, e'_{X_4}, ..., e'_{X_{2n}})$ are mutually independent and $\forall j$ s.t. $\mathrm{Ch}^{\mathcal{H}_1}(V_j) \subseteq \mathrm{PCh}^{\mathcal{G}}(V_j)$, $e_{X_{2j-1}} \perp\!\!\!\perp e_{X_{2j}}$. Without loss of generality, we assume each $c_{2i-1}$ is positive and each $\epsilon_{V_j}$ has variance 1.

**Theorem 12.** $\forall V_i \in \mathbf{U}_a$, $\mathrm{An}^{\mathcal{H}_2}(V_i) \cap (\mathbf{U}_a \cup \mathbf{V}_c) = \emptyset$ and $\mathrm{Ch}^{\mathcal{H}_1}(V_i) \subseteq \mathrm{PCh}^{\mathcal{G}}(V_i)$ iff $\forall V_j \in \mathbf{U}_a \backslash \{V_i\}$, $\mathrm{R}(X_{2j-1}, X_{2i-1}|X_{2i}) \perp\!\!\!\perp X_{2i}$.

*Proof.* When $|\mathbf{U}_a| = 1$, there is $\mathbf{V}_c = \emptyset$ and the only $V_i \in \mathbf{U}_a$ satisfies $\mathrm{Ch}^{\mathcal{H}_1}(V_i) \subseteq \mathrm{PCh}^{\mathcal{G}}(V_i)$, otherwise we can derive contradiction to Cor. 6. We focus on the case where $|\mathbf{U}_a| > 1$.

"Only if": The proof is similar to "only if" part in Thm. 5.

"If": We prove this part by contradiction. All possible cases are as follows.

- Suppose $\text{Ch}^{\mathcal{H}_1}(V_i) \not\subseteq \text{PCh}^{\mathcal{G}}(V_i)$. Based on Cond. 3(2), $\{\text{Ch}^{\mathcal{H}_1}(V_i)\} = \text{P}^3\text{Ch}^{\mathcal{G}}(V_i)$. Let $\text{Ch}^{\mathcal{H}_1}(V_i) = \{V_{i_1}, V_{i_2}\}$ where $V_{i_1} \in \text{Pa}^{\mathcal{G}}(V_{i_2})$. Combining Cor. 6(2) with Cond. 4(2), $\text{De}_{\mathbf{U}_a}^{\mathcal{H}_2}(V_i) \neq \emptyset$ and let $V_j \in \text{De}_{\mathbf{U}_a}^{\mathcal{H}_2}(V_i)$. Clearly, $X_{2i-1}, X_{2i}, X_{2j-1}$ are correlated to each other. Since both $X_{2i-1}$ and $X_{2i}$ contain $\epsilon_{V_{i_1}}$ and $X_{2j-1}$ does not contain $\epsilon_{V_{i_1}}$, $\text{R}(X_{2j-1}, X_{2i-1}|X_{2i}) \not\perp\!\!\!\perp X_{2i}$ based on Lem. 1(2), which leads to contradiction.

- Suppose $\text{Ch}^{\mathcal{H}_1}(V_i) \subseteq \text{PCh}^{\mathcal{G}}(V_i)$ and $\text{An}_{\mathbf{U}_a}^{\mathcal{H}_2}(V_i) \neq \emptyset$, let $V_j \in \text{An}_{\mathbf{U}_a}^{\mathcal{H}_2}(V_i)$. The proof is similar to "if" part in proof of Thm. 5.

- Suppose $\text{Ch}^{\mathcal{H}_1}(V_i) \subseteq \text{PCh}^{\mathcal{G}}(V_i)$, $\text{An}_{\mathbf{U}_a}^{\mathcal{H}_2}(V_i) = \emptyset$, and $\text{An}_{\mathbf{V}_c}^{\mathcal{H}_2}(V_i) \neq \emptyset$, let $V_k \in \text{Pa}_{\mathbf{V}_c}^{\mathcal{H}_2}(V_i)$. Combining Cor. 6(1) with Cond. 4(2), $\text{De}_{\mathbf{U}_a}^{\mathcal{H}_2}(V_k)\backslash\{V_i\} \neq \emptyset$. Besides, there exists $V_j \in \text{De}_{\mathbf{U}_a}^{\mathcal{H}_2}(V_k)\backslash\{V_i\}$ s.t. there exists a directed path from $V_k$ to $V_j$ which does not include $V_i$. Conversely, if for each $V \in \text{De}_{\mathbf{U}_a}^{\mathcal{H}_2}(V_k)\backslash\{V_i\}$, every directed path from $V_k$ to $V$ includes $V_i$, then $\{V_i\}$ is a bottleneck from $\text{Ch}^{\mathcal{G}}(V_k)$ to $\mathbf{O}$, which leads to contradiction to Asmp. 2(1,2). Clearly, $X_{2i-1}, X_{2i}, X_{2j-1}$ are correlated to each other. If $V_j \notin \text{De}^{\mathcal{H}_2}(V_i)$, then $X_{2i-1}$ and $X_{2i}$ both contain $\epsilon_{V_i}$ while $X_{2j-1}$ does not contain $\epsilon_{V_i}$, so $\text{R}(X_{2j-1}, X_{2i-1}|X_{2i}) \not\perp\!\!\!\perp X_{2i}$ based on Lem. 1(2), which leads to contradiction. Otherwise, $X_{2i-1}, X_{2i}$ and $X_{2j-1}$ all contain $\epsilon_{V_i}$ and $\epsilon_{V_k}$, and there exist two non-intersecting paths from $\{V_k, V_i\}$ to $\{X_{2j-1}, X_{2i-1}\}$ (e.g., $V_k \rightarrow ... \rightarrow V_j \rightarrow ... \rightarrow X_{2j-1}$ and $V_i \rightarrow ... \rightarrow X_{2i-1}$), so $\text{R}(X_{2j-1}, X_{2i-1}|X_{2i}) \not\perp\!\!\!\perp X_{2i}$ based on Lem. 1(3), which leads to contradiction.

□

**Proposition 5.** *If we can find $V \in \mathbf{U}_a$ satisfying Thm. 12, Cond. 4 is still valid after removal.*

*Proof.* Based on Thm. 12, Cond. 4(1,2) holds trivially. The remaining proof is similar to the proof of Prop. 3. □

## C.4 Proof of Theoretical Results in Sec. 4.3

**Theorem 13.** Suppose the observed variables are generated by a LiNGAM with latent variables satisfying the bottleneck-faithfulness assumption and Asmp. 2, if Asmp. 1 is invalid, in the limit of infinite data, our algorithm raises an error signal.

*Proof.* Based on Thm. 11, at the end of stage 1, denote $\{L \in \mathbf{V}_a | \text{Ch}^{\mathcal{H}_1}(L) \not\subseteq \text{PCh}^{\mathcal{G}}(L)\}$ by $\mathbf{V}_a'$, if Asmp. 1 is invalid, $\mathbf{V}_c \cup \mathbf{V}_a' \neq \emptyset$. Based on Cor. 6, $|\bigcup \text{GDe}_{\mathbf{V}_a}^{\mathcal{H}_2}(\mathbf{V}_c \cup \mathbf{V}_a')| \geq 2$. Based on Cond. 4(1,2), throughout stage 2, there is always $\bigcup \text{GDe}_{\mathbf{V}_a}^{\mathcal{H}_2}(\mathbf{V}_c \cup \mathbf{V}_a') \subseteq \mathbf{U}_a$. When $\mathbf{U}_a = \bigcup \text{GDe}_{\mathbf{V}_a}^{\mathcal{H}_2}(\mathbf{V}_c \cup \mathbf{V}_a')$, there exists no $V_i \in \mathbf{U}_a$ s.t. $\text{An}^{\mathcal{H}_2}(V_i) \cap (\mathbf{U}_a \cup \mathbf{V}_c) = \emptyset$ and $\text{Ch}^{\mathcal{H}_1}(V_i) \subseteq \text{PCh}^{\mathcal{G}}(V_i)$, that is, we cannot find a $V_i \in \mathbf{U}_a$ satisfying the independence condition in Thm. 12. Therefore, before $\mathbf{U}_a$ becomes an empty set, our algorithm raises an error signal. □

# D Real-world Data

The ground-truth causal graph of multitasking behavior model is shown as Fig. 13(a), it satisfies Asmp. 1, on which our algorithm yields a correct result. Moreover, we add some edges (marked in red in Fig. 13(b)) into the ground-truth graph by replacing some single variable with the sum of multiple variables, the modified graph violates Asmp. 1, on which our algorithm raises an error signal.

# E Algorithm

The details of our proposed algorithm are provided in Alg. 3 and 4.

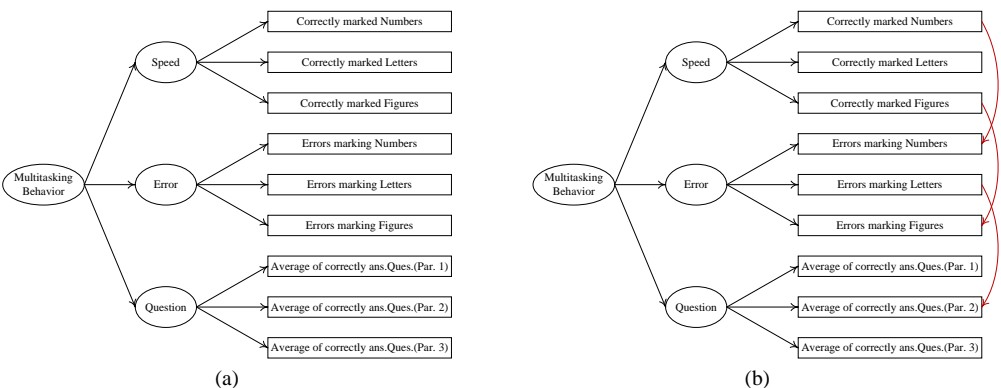

Figure 13: (a) ground-truth causal graph and (b) modified causal graph of multitasking behavior model. Rectangles represent observed variables while circles represent latent variables.

---

**Algorithm 3:** Stage 1: Identifying latent variables (detailed)

---

**Input:** Observed variables $\mathbf{O}_0$ and $\mathbf{O}_1$
**Output:** $\mathbf{V}_a$, $\mathbf{V}_b$, and $\mathcal{H}_1$

1   Initialize $\mathbf{V}_a$ as $\mathbf{O}_0$, $\mathbf{V}_b$ as $\mathbf{O}_1$, and let $V_1 \in \mathrm{Pa}^{\mathcal{H}_1}(V_2)$ iff $V_1 \in \mathrm{Pa}^{\mathcal{G}}(V_2)$, $V_1 \in \mathbf{O}_0$, and $V_2 \in \mathbf{O}_1$.

2   **while** *the current $\mathbf{V}_a$ is not identical to the previous $\mathbf{V}_a$* **do**

3      **Assert** Cond. 1 holds if Asmp. 1 holds.

4      $\mathbb{S} := \emptyset$.

5      **for** $\{V_i, V_j\} \subseteq \mathbf{V}_a$ **do**

6          **if** $\exists V_k \in \mathbf{V}_a \backslash \{V_i, V_j\}$ *s.t.* $\mathrm{Cov}(V_i, V_j)\,\mathrm{Cov}(V_i, V_k)\mathrm{Cov}(V_j, V_k) \neq 0$ *and* $\mathrm{R}(V_i, V_j | V_k) \perp\!\!\!\perp \mathbf{V}_a \backslash \{V_i, V_j\}$ **then**

7             $\mathbb{S} := \mathbb{S} \cup \{\{V_i, V_j\}\}$.

8          **end**

9      **end**

10     **Assert** $\mathbb{S}$ consists of all identifiable pairs satisfying Def. 2 if Asmp. 1 holds.

11     $\mathbb{S}_1 := \emptyset, \mathbb{S}_2 := \emptyset, \mathbb{S}_3 := \emptyset$.

12     **for** $\{V_i, V_j\} \in \mathbb{S}$ *where* $\{V_{i_1}, V_{i_2}\} \subseteq \mathrm{Ch}^{\mathcal{H}_1}(V_i)$ *and* $\{V_{j_1}, V_{j_2}\} \subseteq \mathrm{Ch}^{\mathcal{H}_1}(V_j)$ **do**

13        **if** $\mathrm{R}(V_{i_1}, V_j | V_{i_2}) \perp\!\!\!\perp V_{i_2}$ **then**

14           $\mathbb{S}_1 := \mathbb{S}_1 \cup \{\{V_i, V_j\}\}$ and $V_i \in \mathrm{Pa}^{\mathcal{G}}(V_j)$.

15        **else if** $\mathrm{R}(V_{j_1}, V_i | V_{j_2}) \perp\!\!\!\perp V_{j_2}$ **then**

16           $\mathbb{S}_1 := \mathbb{S}_1 \cup \{\{V_i, V_j\}\}$ and $V_j \in \mathrm{Pa}^{\mathcal{G}}(V_i)$.

17        **else if** $\exists \{V_{i'}, V_{j'}\} \in \mathbb{S} \backslash \{\{V_i, V_j\}\}$ *s.t.* $\{V_{i'}, V_{j'}\} \cap \{V_i, V_j\} \neq \emptyset$ **then**

18           $\mathbb{S}_2 := \mathbb{S}_2 \cup \{\{V_i, V_j\}\}$.

19        **else if** $\exists \{V_k, V_l\} \subseteq \bigcup \mathrm{Ch}^{\mathcal{H}_1}(\mathbf{V}_a \backslash \{V_i, V_j\})$ *s.t.* $(V_{i_1}, V_{i_2}, V_j, V_k, V_l)$ *satisfies the quintuple constraint* **then**

20           $\mathbb{S}_3 := \mathbb{S}_3 \cup \{\{V_i, V_j\}\}$ and $V_i \in \mathrm{Pa}^{\mathcal{G}}(V_j)$.

21        **else if** $\exists \{V_k, V_l\} \subseteq \bigcup \mathrm{Ch}^{\mathcal{H}_1}(\mathbf{V}_a \backslash \{V_i, V_j\})$ *s.t.* $(V_{j_1}, V_{j_2}, V_i, V_k, V_l)$ *satisfies the quintuple constraint* **then**

22           $\mathbb{S}_3 := \mathbb{S}_3 \cup \{\{V_i, V_j\}\}$ and $V_j \in \mathrm{Pa}^{\mathcal{G}}(V_i)$.

23        **else**

24           $\mathbb{S}_2 := \mathbb{S}_2 \cup \{\{V_i, V_j\}\}$.

25        **end**

26     **end**

27     **Assert** $\mathbb{S}_1$ consists of all type-1 identifiable pairs satisfying Def. 2(1) if Asmp. 1 holds.

28     **Assert** $\mathbb{S}_2$ consists of all type-2 identifiable pairs satisfying Def. 2(2) if Asmp. 1 holds.

29     **Assert** $\mathbb{S}_3$ consists of all type-3 identifiable pairs satisfying Def. 2(3) if Asmp. 1 holds.

30     $\mathbb{P} := \emptyset$.

31     **for** $\{V_i, V_j\} \in \mathbb{S}_2$ **do**

32        **if** $\exists \{V_{i'}, V_{j'}\} \in \mathbb{S}_1$ *s.t.* $\{V_i, V_j\} \cap \{V_{i'}, V_{j'}\} \neq \emptyset$ **then**

33           $\mathbb{P} := \mathbb{P} \cup \{\{V_i, V_j\}\}$.

34        **end**

35     **end**

36     **Assert** $\forall \{V_i, V_j\} \in \mathbb{P}, \bigcap \mathrm{Pa}^{\mathcal{G}}(\{V_i, V_j\}) \subseteq \mathbf{V}_a$ if Asmp. 1 holds.

37     $\mathbb{S}_2 := \mathbb{S}_2 \backslash \mathbb{P}, m := 0$.

38     **while** $\mathbb{S}_2 \neq \emptyset$ **do**

39        Arbitrarily choose $\{V_i, V_j\} \in \mathbb{S}_2$.

40        $m := m + 1, \mathbb{Q}_m := \{\{V_i, V_j\}\}$.

41        **for** $\{V_{i'}, V_{j'}\} \in \mathbb{S}_2 \backslash \{\{V_i, V_j\}\}$ **do**

42           **if** $\exists \{V_{i''}, V_{j''}\} \in \mathbb{S}_2$ *s.t.* $\{V_i, V_j\} \cap \{V_{i''}, V_{j''}\} \neq \emptyset$ *and* $\{V_{i'}, V_{j'}\} \cap \{V_{i''}, V_{j''}\} \neq \emptyset$ **then**

43             $\mathbb{Q}_m := \mathbb{Q}_m \cup \{\{V_{i'}, V_{j'}\}\}$.

44           **end**

45        **end**

46        $\mathbb{S}_2 := \mathbb{S}_2 \backslash \mathbb{Q}_m$.

47     **end**

48     **Assert** $\forall k \in \{1, ..., m\}$ and $\{\{V_i, V_j\}, \{V_{i'}, V_{j'}\}\} \subseteq \mathbb{Q}_k, \bigcap \mathrm{Pa}^{\mathcal{G}}(\{V_i, V_j\}) = \bigcap \mathrm{Pa}^{\mathcal{G}}(\{V_{i'}, V_{j'}\}) \subseteq \mathbf{V}_c$ if Asmp. 1 holds.

49     **for** $\{V_i, V_j\} \in \mathbb{S}_1$ *where* $V_i \in \mathrm{Pa}^{\mathcal{G}}(V_j)$ **do**

50        $\mathbf{V}_a := \mathbf{V}_a \backslash \{V_j\}, \mathbf{V}_b := \mathbf{V}_b \cup \{V_j\}$, and $\mathcal{H}_1 := \mathcal{H}_1 \cup \{V_i \to V_j\}$.

51     **end**

52     **for** $k = 1 : m$ **do**

53        Introduce a new latent variable $L$.

54        $\mathbf{V}_a := \mathbf{V}_a \cup \{L\} \backslash \bigcup \mathbb{Q}_k, \mathbf{V}_b := \mathbf{V}_b \cup \bigcup \mathbb{Q}_k$, and $\mathcal{H}_1 := \mathcal{H}_1 \cup \{L \to V | V \in \bigcup \mathbb{Q}_k\}$.

55     **end**

56   **end**

57   **Assert** $\mathbf{V}_c = \emptyset$ if Asmp. 1 holds.

---

---

**Algorithm 4:** Stage 2: Inferring causal relations (detailed)

---

**Input:** $\mathbf{V}_a$, $\mathbf{V}_b$, and $\mathcal{H}_1$ output by Alg. 3

**Output:** a complete causal structure $\mathcal{G}$

1   Initialize $\mathbf{U}_a$ as $\mathbf{V}_a$ and assign two observed surrogates $X_{2i-1}, X_{2i}$ for each $V_i \in \mathbf{U}_a$.

2   **while** $|\mathbf{U}_a| > 0$ **do**

3      **Assert** Cond. 2 holds if Asmp. 1 holds.

4      **if** *There exists $V_i \in \mathbf{U}_a$ s.t. $\forall V_j \in \mathbf{U}_a \setminus \{V_i\}, \mathrm{R}(X_{2j-1}, X_{2i-1}|X_{2i}) \perp\!\!\!\perp X_{2i}$* **then**

5         **Assert** $V_i$ is a root variable among $\mathbf{U}_a$ if Asmp. 1 holds.

6         $\forall V_j \in \mathbf{U}_a \setminus \{V_i\}$, calculate $\mathrm{sgn}(\mathrm{m}_{ji})$, $c_{2i-1}c_{2i}c_{2j-1}c_{2j}m_{ji}^2$, and $c_{2i-1}c_{2i}$.

7         $\forall V_k \in \mathbf{V}_a \setminus \mathbf{U}_a, m_{ki} := 0$.

8         $\mathbf{U}_a := \mathbf{U}_a \setminus \{V_i\}, X_{2j-1} := \mathrm{R}(X_{2j-1}, X_{2i-1}|X_{2i}), X_{2j} := X_{2j}$.

9      **else**

10         **raise error** // This happens iff Asmp. 1 fails

11      **end**

12   **end**

13   **for** $\{V_i, V_j\} \subseteq \mathbf{V}_a$ **do**

14      $m_{ji} := \mathrm{sgn}(\mathrm{m}_{ji})\sqrt{\dfrac{c_{2i-1}c_{2i}c_{2j-1}c_{2j}m_{ji}^2}{c_{2i-1}c_{2i}c_{2j-1}c_{2j}}}$ if $m_{ji} \neq 0$

15   **end**

16   $A := I - M^{-1}$ where $M$ is composed of $m_{ji}$.

17   $\mathcal{H}_2 := (\mathbf{V}_a, \emptyset)$.

18   **for** $\{V_i, V_j\} \subseteq \mathbf{V}_a$ **do**

19      **if** $a_{ji} \neq 0$ **then**

20         $\mathcal{H}_2 := \mathcal{H}_2 \cup \{V_i \to V_j\}$

21      **end**

22   **end**

23   $\mathcal{G} := \mathcal{H}_1 \cup \mathcal{H}_2$

---

