# OpenReview forum: "Efficient and Trustworthy Causal Discovery with Latent Variables and Complex Relations"
_ICLR.cc/2025/Conference — ICLR 2025 Poster_

### Official Review · Reviewer_SqCS · 2024-11-03

**Soundness:** 3
**Presentation:** 2
**Contribution:** 3
**Rating:** 6
**Confidence:** 4

**Summary:**

This paper considers the problem of causal discovery with latent variables, in a more general setting where the causal relations between observed variables are allowed, with some other structural assumptions (e.g., pure children). Moreover, those assumptions are claimed to be testable, i.e., when the assumptions are not satisfied, the algorithm can raise an error.

**Strengths:**

1. The idea to testify the graphical assumptions and give an error signal instead of giving possibly incorrect result is very significant for causal discovery with latent variables.

2. The generalization to allow edges in between observed variables is good.

**Weaknesses:**

Overall, the current draft's presentation **requires substantial improvement**. It's cluttered with theorem after theorem, while the core algorithm is scattered throughout and described with vague language. This makes it easy for readers (at least me) to lose track, unclear on the purpose of each theoretical statement, and difficult to assess the correctness of all the assertions. I totally understand that the work involves dense technical material and possibly complex algorithms, but this shouldn't excuse its current lack of clarity.

Here are my specific comments/questions:

1. In assumption 1., I guess what the authors meant is pure children or neighbors in the observed variables? Though the observed variables is not within the scope for definition 1.

2. For definition 2 (Identifiable pair, IP),
   - The definition heavily relies on H2, and the definition for H2 heavily relies on Vf. But so far (when definition 2 is presented) we only have "Vf is unknown", "H2 is unknown", which makes the understanding to definition 2 at that point very difficult, if not impossible.
   - Please explain every definition/theorem/algorithmic step's purpose before directly delving into the technical sides (same applies elsewhere). E.g., here what is the general purpose of H2 (instead of the technical definition on "induced subgraph of G over Vf and VC")? What does Vf's "launched in the future" mean? It's until pg.6 did I notice that the authors want to use Vf for latent variables..
   - Since VC and Vf can be updated, does the "identifiable pairs" changes in each epoch, based on different VC and Vf? Intuitively this shouldn't happen, as it seems describing something regarding the true structure. Then please show that there's no such dependence.
   - Why is it named "identifiable pairs"? Theorem 1 shows a way to sufficiently identify them from data, but are they necessarily all the pure children information that can be identified (e.g., using Adam's formulation of equivalence class)?
   - Instead of {V1, V2} ∈ S, please use e.g., (V1, V2) to indicate that they are ordered.

3. In definition 4's "Intuition", what are those "e1', e2', ..."? Are they referring to the noise added to get Oi' in O1 variables?

4. In theorem 2, what does "let {Vi1, Vi2}⊂ChH1(Vi)" mean? I understand that at the initialization epoch, ChH1(Vi) is just the corresponding two added O1 variables. But after that, does it mean "for any two variables in ChH1(Vi), the followings hold"?

5. In theorem 3, what does "∀Si ∈ S2" mean? Before this all the notations for enumerating items in S is in the form of {Vi, Vj} ∈ S. If the authors intend to express a same thing, please be consistent throughout.

Overall, for all the clarity/presentation issues that prevents me to further evaluate the work, I would appreciate it if the authors could provide:
   - A clear overview of the algorithm: What is the input? What are all the assumptions (except for the structural ones, e.g., at least there should be some faithfulness)? What is the output (to which equivalence class does it identify -- e.g., does it achieve Adam's? which parts are assumed to be correct? when (if and only if) will the algorithm give an error signal)?
   - A detailed pseudocode for the algorithm. For each step, instead of vague language (like those in the current "Updating the Active Set" paragraph), please use formal math/set language. For all the claims on the intermediate results (e.g., those reflected by condition 1, theorems 1, 2), please use "assert" lines.

---

Some other methodological questions:

6. Why do the authors need to add new simulated variables O1 into the system? Intuitively they couldn't give any more information than those already encoded in O0. In other words, all information found by O1 should be able to be found by O0. Could the authors please give an example where such intuition is incorrect? Otherwise the O1 seems only adding more unnecessary complexities.

7. What is the relationship between the two main constraints (Pseudo-residual and Quintuple constraint) used in this paper and the GIN condition? If they are special cases for GIN condition, why not directly use the GIN condition? Or can they identify something beyond GIN?

8. Regarding the at-least-2-pure-children assumption, can this algorithm be applied to violation cases e.g., measurement error?

9. Regarding the testability of the assumption, could the authors please give a brief review and compare to how other (linear non-Gaussian acyclic models with latents) work testify their assumptions?

**Questions:**

See above.

---

> ### Author Response · Authors · 2024-11-18
>
> # Part (1/4)
>
> Thank you very much for your careful reading and assessment of our work as well as the suggestions. We believe that there are several misunderstandings of our approach. To address this, we have improved the presentation of our manuscript substantially (where major revisions are marked in purple) and provide comprehensive clarification below, which we hope will resolve these misunderstandings. We slightly reordered your questions to match the sequence of concepts or theorems as they appear in the paper.
>
> > Q1: Algorithm overview for input, output, assumptions, etc.
>
> We totally agree that an algorithm overview makes it easier for readers to grasp the main workflow of our approach. We have added an overview version of our algorithm in the main text (Algorithms 1 and 2). The input of Algorithm 1 is observed variables $\mathbf{O} _0$ and $\mathbf{O} _1$. Algorithm 1 produces intermediate results that serves as the input of Algorithm 2, and Algorithm 2 outputs a complete causal structure (which is a directed acyclic graph that explicitly represents both observed and latent variables along with their causal relations) or raises an error finally.
>
> As a further supplement, we have added Section 3.3 and 4.3 to clearly characterize what results our algorithm can deliver under different sets of assumptions. Specifically,
> - Theorem 7 in Section 3.3 states that "Suppose the observed variables are generated by a linear latent non-Gaussian model satisfying the rank-faithfulness assumption and Assumption 1, in the limit of infinite data, our algorithm correctly identifies the underlying complete causal structure."
> - Theorem 13 in Section 4.3 states that "Suppose the observed variables are generated by a linear latent non-Gaussian model satisfying the rank-faithfulness assumption and Assumption 2, if Assumption 1 is invalid, in the limit of infinite data, our algorithm raises an error."
>
> > Q2: Detailed pseudo-code
>
> Thanks for your valuable comment. We have refined the detailed version of our algorithm (Algorithms 3 and 4) in Appendix, where each step is precisely described in math/set language, which eliminates ambiguity and ensures precise interpretation of each algorithmic step. Also, we use assert line to for intermediate results, which helps readers clearly understand what properties are guaranteed at each critical step of the algorithm.
>
> > Q3: Why introduce $\mathbf{O} _1$
>
> It is correct that $\mathbf{O} _1$ give no more information than those already encoded in $\mathbf{O} _0$. Instead, the purpose of introduce $\mathbf{O} _1$ is to streamline technical details in our paper and keep the presentation focused on the core ideas. The detailed explanation is as follows.
>
> In this paper, we consider the scenario with both latent and observed variables. As stated in footnote 1 in the revised manuscript, "While the values of observed variables are directly accessible for causal discovery, the causal relations of latent variables can only be inferred indirectly, e.g., through their pure children. By introducing $\mathbf{O}_1$ to create pure children for each observed variable, we can handle both types of variables through analyzing their pure children, thereby eliminating the need to repeatedly distinguish between treatments of latent and observed variables and keeping the core methodology clear."
>
> > Q4: Whether the pure children or neighbors in Assumption 1 must be observed variables
>
> Actually, the assumption allows both pure children and neighbors to be latent variables. In Example of Assumption 1, we have clarified that "the graph in Figure 2(a) satisfies this assumption, where $\mathrm{PCh}^{\mathcal{G}_0}(L_1) = \\{L_3, L_4\\}$ and $\mathrm{PCh}^{\mathcal{G}_0}(L_1) = \\{L_2, L_3, L_4, O_2, O_6\\}$."
>
> We would like to highlight that even if the latent variable $L _1$ in Figure 2(a) only have two latent pure children $L _3, L _4$, we can still identify it. This is clarified in § Repeating This Process in Section 3.1. Specifically, when we need to do independence/correlation test involving any latent variable, we can directly replace it with its any observed descendant in $\mathcal{H} _1$.

---

> ### Author Response · Authors · 2024-11-18
>
> # Part (2/4)
>
> > Q5: Definition of $\mathbf{V} _f, \mathcal{H} _2$
>
> In the revised manuscript, we have refined the definition, supplemented it with intuitive explanations, and provided additional illustrative examples to enhance comprehension.
>
> - Before defining $\mathbf{V} _f, \mathcal{H} _2$, at the beginning of § Initialization in Section 3.1, we first define $\mathbf{V}_c, \mathbf{V}_p, \mathcal{H}_1$ as two sets of variables and a graph with specific initialization and update rules. Immediately following this, we offer an intuitive explanation. "Intuitively, $\mathbf{V} _c$ consists of identified variables whose causal relations (i.e., both incoming and outgoing edges of the variable in the underlying causal graph) are not fully identified, $\mathbf{V} _p$ consists of identified variables whose causal relations are fully identified, and $\mathcal{H} _1$ consists of all identified causal relations. Considering the initial case (when $\mathbf{V} _c = \mathbf{O} _0$, $\mathbf{V} _p = \mathbf{O} _1$, and $\mathcal{H} _1$ consists of edges from $\mathbf{O} _0$ to $\mathbf{O} _1$), such intuitions become particularly apparent." Moreover, we include more illustrative figures (left subfigures of Figures 3, 4, 5) to display $\mathbf{V} _c, \mathbf{V} _p, \mathcal{H} _1$ at different iteration during stage 1.
>
> - After defining $\mathbf{V}_c, \mathbf{V}_p, \mathcal{H}_1$, we define $\mathbf{V} _f$ as $\mathbf{V} \backslash (\mathbf{V} _c \cup \mathbf{V} _p)$ and $\mathcal{H} _2$ as the induced subgraph of $\mathcal{G}$ over $\mathbf{V} _c \cup \mathbf{V} _f$. Immediately following this, we offer an intuitive explanation. "Intuitively, while $\mathbf{V} _c \cup \mathbf{V} _p$ consists of all identified variables, $\mathbf{V} _f$ consists of all unidentified variables. While $\mathcal{H} _1$ consists of all identified causal relations, $\mathcal{H} _2$ consists of all unidentified causal relations. Considering the initial case (Initially we have identified no latent variable and no causal relation in $\mathcal{G} _0$, when $\mathbf{V} _f = \mathbf{L}$ and $\mathcal{H} _2 = \mathcal{G} _0$), such intuitions become particularly apparent." Moreover, we include more illustrative figures (right subfigures of Figures 3, 4, 5) to display $\mathbf{V} _f, \mathcal{H} _2$ at different iteration during stage 1.
>
> > Q6: (1) Why the name "identifiable pairs" (2) Whether all pure children information can be identified
>
> (1) The definition of identifiable pairs relies on $\mathcal{H} _2$. Although $\mathcal{H} _2$ consists of unidentified causal relations, identifiable pairs can still be located from $\mathbf{V}_c$ via statistical analysis (Theorem 1), this is what ""identifiable" means. This explanation has been added to Remark of Definition 2.
>
> (2) Although not all pure children information can be identified from identifiable pairs, this is not a concern. Please note the goal of stage 1 is to identify all latent variables (rather than to identify all pure children information) and Theorem 4 guarantees that we can identify all latent variables at the end of stage 1. If some pure children information are not identified in stage 1, it will be identified in stage 2. For example, consider a simple case where there are no latent variable and only two observed variables $O _1, O _2$ where $O _1 \to O _2$. According to § Initialization in Section 3.1, we initialize $\mathbf{V} _c$ as $\\{ O _1, O _2 \\}$. Because there is no identifiable pair (Particularly, $\\{ O _1, O _2 \\} \notin \mathbb{S} _1$ because $\mathrm{Ne} ^{\mathcal{H} _2}(O _1) \backslash \\{O _2\\} = \emptyset$), we cannot determine that $O _2$ is $O _1$'s pure child in stage 1. Instead, we will discover $O _1 \to O _2$ in stage 2.
>
>
> > Q7: Whether identifiable pairs changes at each iteration
>
> Actually, identifiable pairs changes at each iteration. Any identifiable pair is composed of two variables in $\mathbf{V} _c$ according to Definition 2, as $\mathbf{V} _c$ changes at each iteration, identifiable pairs changes at each iteration naturally. We provide two examples as follows.
>
> - Suppose at some iteration, $\\{V _i, V _j\\} \in \mathbb{S} _2$, then according to the update rules, both $V _i$ and $V _j$ will be moved into $\mathbf{V} _p$, so $\\{V _i, V _j\\}$ will not be an identifiable pair at the next iteration.
>
> - Suppose at some iteration, $\\{V _i, V _j\\} \in \mathbb{S} _2$. It is entirely possible that at the previous iteration, $\\{V _i, V _j\\} \subset \mathbf{V} _f$. Clearly, $\\{V _i, V _j\\}$ was not an identifiable pair at the previous iteration.

---

> ### Author Response · Authors · 2024-11-18
>
> # Part (3/4)
>
> > Q8: Why not use $(V _i, V _j) \in \mathbb{S}$
>
> According to our Definition 2, $\mathbb{S} = \mathbb{S} _1 \cup \mathbb{S} _2 \cup \mathbb{S} _3$. Although the identifiable paris in $\mathbb{S} _1 \cup \mathbb{S} _3$ are ordered, those in $\mathbb{S} _2$ are not ordered, so it is not advisable to use the notation $(V_1, V_2) \in \mathbb{S}$.
>
> Also, even the notation $(V _i, V _j) \in \mathbb{S} _1$ may also cause some problems. Specifically, in Theorem 3, we need to check whether an identifiable pair in $\mathbb{S} _1$ and another in $\mathbb{S} _2$ has a common element. Suppose there is $(V_i, V_j) \in \mathbb{S} _1$ and $\\{V' _i, V' _j\\} \in \mathbb{S}$, the intersection operation between $(V_i, V_j)$ and $\\{V' _i, V' _j\\}$ is not well-defined.
>
> Based on these considerations, we think it might be more advisable to use $\\{V _i, V _j\\} \in \mathbb{S}$. For $\\{V _i, V _j\\} \in \mathbb{S} _1$ or $\\{V _i, V _j\\} \in \mathbb{S} _1$, we explicitly indicate whether $V _i \in \mathrm{Pa}(V _j)$ or $V _j \in \mathrm{Pa}(V _i)$ when necessary. We remain open to better alternatives and would gladly incorporate them.
>
> > Q9: Purposes of definitions / theorems / algorithmic steps.
>
> We have included more explanations to help readers grasp the purposes of definitions / theorems / algorithmic steps. Some examples are given as follows.
>
> - For $\mathbf{V} _f, \mathcal{H} _2$, the purpose of the definitions becomes readily apparent from the intuitive explanation. Specifically, while $\mathbf{V} _c \cup \mathbf{V} _p$ consists of all identified variables, we define $\mathbf{V} _f$ to represent all unidentified variables; while $\mathcal{H} _1$ consists of all identified causal relations, we define $\mathcal{H} _2$ to represent the graph consisting of all unidentified causal relations.
>
> - For Theorem 1, the purpose of the theorem becomes readily apparent from from its Remark. Specifically, Theorem 1 is used to provide a method for locating identifiable pairs from $\mathbf{V} _c$ via statistical analysis.
>
> - For the algorithmic step of identifying identifiable pairs from $\mathbf{V}_ c$, the purpose is clarified in § Locating Pure Children in Section 3.1. "Ideally, we want to locate pure children in a single step, but this is impossible because of the existence of complex causal relations. Instead, we first locate identifiable pairs from $\mathbf{V} _c$ and then locate pure children from these identifiable pairs."
>
> > Q10: What "let $\\{V_ {i_ 1}, V_ {i_ 1}\\} \subset \mathrm{Ch}^ {\mathcal{H}_ 1}(V_ i)$" in Theorem 2 means
>
> This just means $\\{V_ {i_ 1}, V_ {i_ 2}\\}$ are any two variables in $\mathrm{Ch}^ {\mathcal{H}_ 1}(V_ i)$. At the first iteration when $\mathbf{V}_ c = \mathbf{O}_ 0$, both $V_ {i_ 1}$ and $V_ {i_ 2}$ are variables in $\mathbf{O} _1$. But after that, they might be variables in $\mathbf{L}$ or $\mathbf{O} _0$.
>
> > Q11: Whether $e' _1$, $e' _2$, ... in Intuition of Definition 4 refer to noises used to construct $\mathbf{O} _1$
>
> In Intuition of Definition 4 (definition of the quintuple constraint), each of $e _i, e _j, e' _1, e' _2,...$ does not refer to any specific instance. Instead, the equation $V _1 = \lambda _1 e _i + \gamma _1 e _j + e' _1, V _2 = \lambda _2 e _i + \gamma _2 e _j + e' _2, ...$ means that $V _1$ can be expressed as the sum of three random variables $\lambda _1 e _i, \gamma _1 e_j, e'_1$, $V _2$ can be expressed as the sum of three random variables $\lambda _2 e _i, \gamma _2 e_j, e'_2$, ..., where $e _i, e_j, e' _1, e' _2, ...$ satisfies some constraints, e.g., $e _i, e_j, e' _1, e' _2, ...$ are mutually independent. By the way, this Intuition is moved to Lemma 2 in Appendix C.1 in the revised manuscript.
>
> From another perspective, as stated in response to Q10, $V_ {i_ 1}$ and $V_ {i_ 2}$ might not be variables in $\mathbf{O} _1$ after the first iteration. Suppose $e' _1$, $e' _2$, ... refer to noises used to construct $\mathbf{O} _1$, then the quintuple constraint cannot be applied to the case where $\\{V _{i _1}, V _{i _2}\\} \not \subset \mathbf{O} _1$, which leads to contradiction.
>
> > Q12: What $\mathcal{S} _i \in \mathbb{S} _2$ in Theorem 3 means
>
> This is equivalent to $\\{V _i, V _j\\} \in \mathbb{S} _2$. Following your advice, we replace $\mathcal{S} _i \in \mathbb{S} _2$ with $\\{V _i, V _j\\} \in \mathbb{S} _2$ to maintain consistency.

---

> ### Author Response · Authors · 2024-11-18
>
> # Part (4/4)
>
> > Q13: The relationship between two main constraints (pseudo-residual and quintuple constraint) and the GIN condition
>
> We first detail the relationship between pseudo-residual and GIN condition, then the relationship between quintuple constraint and the GIN condition.
>
> - Pseudo-residual and GIN condition are fundamentally different. The former is a specific linear combination of two variables while the latter consists of a set of independence relations. In Section 3.1, pseudo-residual always appears in independence relations. In principle, these independence relations can be replaced with GIN condition seamlessly, but we still use pseudo-residual because it is simpler in form and more accessible. In Section 3.2, pseudo-residual is also used to update $X_ {2j-1}$ in Equation (6) where $X_ {2j-1} := \mathrm{R}(X_ {2j-1}, X_ {2i-1} | X_ {2i})$, whereas GIN is typically not suitable for such use. While the authors of GIN [1] proposed a GIN-based method for inferring causal orders that requires no update of $X_ {2j-1}$ throughout, its time complexity is $\mathcal{O}(|\mathbf{V}_ c|^ 4)$, whereas ours has only $\mathcal{O}(|\mathbf{V}_ c|^ 3)$ time complexity.
>
> - Quintuple constraint is similar but not identical to a special case of GIN condition. Specifically, " $( V _{i _1}, V _{i _2}, V _j, V _k, V _l )$ satisfies quintuple constraint" is most similar to " $( \\{V _{i _2}, V _l\\}, \\{V _{i _1}, V _j, V _k\\} )$ satisfies GIN condition". The former implies that if there exists $\alpha, \beta$ s.t. $V _{i _1} + \alpha V _j + \beta V _k$ is uncorrelated to $V _{i _2}$ and $V _l$, then $V _{i _1} + \alpha V _j + \beta V _k$ is independent of $V _{i _2}$. The latter implies that if there exists $\alpha, \beta, \gamma$ s.t. $\alpha V _{i _1} + \beta V _j + \gamma V _k$ is uncorrelated to $V _{i _2}$ and $V _l$, then $\alpha V _{i _1} + \beta V _j + \gamma V _k$ is independent of $V _{i _2}$ and $V _l$. We have proven that the quintuple constraint is sufficient and necessary to identify identifiable pairs in $\mathbb{S} _3$ in Theorem 2, so we do not consider GIN condition that requires an additional independence test.
>
> In our opinion, the main advantage of GIN lies in its superior capability to handle n-factor models. In fact, we plan to extend our approach to more challenging scenarios using GIN.
>
> [1] Feng Xie et al. "Generalized independent noise condition for estimating latent variable causal graphs." NeurIPS 2020.
>
> > Q14: Whether this algorithm can be applied to violation cases such as measurement error
>
> It should be noted at the outset that our algorithm relies on the fact that in LiNGAMs, each variable can be expressed as a linear combination of its parents plus an independent non-Gaussian noise. Suppose the measurement error is an additive noise, this algorithm can be applied to the case where all observed variables have no child in the underlying causal graph (such as Case 1 and Case 2 in Figure 9), because each observed variables can still be expressed as a linear combination of their parents plus an independent term, which is the sum of the exogenous noise and the measurement error. However, this algorithm cannot be applied to the cases where some observed variables have descendants. For example, if $\mathrm{Pa}(O _2) = \\{O _1\\}$, then $O _2 = a _{21} O _{1} + \epsilon _{O _2}$. With measurement errors $e _1, e _2$, $\tilde{O} _1 = O _1 + e _1$ and $\tilde{O} _2 = a _{21} O _{1} + \epsilon _{O _2} + e _2$. In general, $\tilde{O} _2$ cannot be expressed as scaled $\tilde{O} _1$ plus a term independent of $\tilde{O} _1$ in this case.
>
> > Q15: How other work testify their assumptions
>
> As emphasized in the paper, to the best of our knowledge, no existing work on causal discovery with latent variables has rigorously discussed how to testify their assumptions, especially the widely-used pure children assumption. Without validating these assumptions, there is no guarantee that their recovered causal graph correctly reflects the true causal relations. This lack of verification could be potentially harmful in practical applications. For instance, in financial markets, a plausible but incorrect causal conclusion might mislead investors to make poor investment choices and cause significant financial losses. Even worse, users might not realize the unreliability of these results since the assumptions were never verified.
>
> Therefore, our proposed algorithm represents both an innovative advancement and a significant contribution to the field. Unlike previous methods that might silently return incorrect results when assumptions are violated, our trustworthy algorithm can actively detect invalid pure children assumptions and raise an error accordingly. This capability ensures that users are protected from acting on potentially incorrect causal conclusions, marking a crucial step toward more reliable causal discovery in practice.

---

> ### Author Response · Authors · 2024-11-25
> **Further Discussion**
>
> Dear Reviewer SqCS:
>
> We thank you again for your careful reading and assessment of our work. We have substantially improved clarification of our manuscript, e.g., we have provided an algorithm overview and more intuitive explanations for concepts and theorems. Moreover, we have taken our maximum effort to address your every concern.
>
> It is really important to us that you could kindly read our rebuttal and provide further questions if there are any. Thank you so much and hope you have a good day.
>
>
> Best,
>
> Authors

---

> ### Comment · Reviewer_SqCS · 2024-11-25
>
> Thank the authors for the detailed and prompt response. My concerns on presentation are relatively addressed. A few more questions:
>
> 1. Regarding "identifiable pairs", since it is defined in a rolling-based way, do we also have a static way that can determine whether an edge (or other graphical pattern) is globally identifiable from data? Would that be the same as the ones defined in [Adams]? How is the gap between the final results you could identify to the ones that can be maximally identified?
>
> 2. Can I under the quintuple constraint as a specific case of GIN condition, where by some graphical condition, one avoids to test between the linear combination's independence to all of another set of variables. "saving another time of CI test" is not that accurate since for HSIC test, one-dim or multi-dim variables are almost the same and can be checked once. But my biggest curiosity is that, e.g., what if using GIN as in [Jins] so that more conditions (than quintuple constraints) can be used? Will this work identify more or that Assumption 1 can be further relaxed?
>
> 3. I appreciate the authors' pseudocode. It would be better if the authors could provide an executable code of the algorithm in the appendix, where e.g., CI test, constraint checker can be done in oracle, and the expect outputs' correctness is checked in an `assert' way, for arbitrary input graph structures. This is because for a highly technical paper that is workflow/algorithm driven with different graphical patterns, it is not easy for reviewers to grasp all the intuition/motivation, and thus it would be hard to evaluate its correctness. However, for an algorithm based paper, such correctness check is always necessary.

---

> ### Author Response · Authors · 2024-11-26
>
> # Part (1/2)
>
> Dear Reviewer SqCS:
>
> Thanks for your response. We are glad to answer your further questions as follows.
>
> > Q16: Static way & gap
>
> First, we would like to highlight that research on causal discovery typically encompasses two aspects: identifiability result that establishes whether the underlying causal structure can be identified from data under certain conditions, and identification method that describes how to actually identify the underlying causal structure from data if it is identifiable. In the following, we systematically analyze both our work and that of Adams et al. from these two aspects. Finally, we provide answers to your questions.
>
> **1. Identifiability result.** Given observed variables generated by a LiNGAM with latent variables satisfying the rank-faithfulness assumption, the identifiability result of Adams et al. is that the underlying causal structure can be identified if it satisfies both the bottleneck condition and the strong non-redundancy conditions presented in their paper; our identifiability result is that the underlying causal structure can be identified if it satisfies Assumption 1 presented in our paper. Details of these conditions are omitted here due to space limit, we refer interested readers to the original papers.
>
> **2. Discussion on identifiability result.** Both the work of Adams et al. and our work imply that the underlying causal structure can be identified if it satisfies some conditions. All the conditions, including their bottleneck and strong non-redundancy conditions and our Assumption 1, are defined in a static way. Particularly, our identifiability result itself (excluding intermediate results) relies solely on Assumption 1, without involving the concept of identifiable pairs defined in a rolling-based way. Furthermore, Adams et al. prove that their identifiability result is exactly the theoretical maximum identifiability. There indeed exists a gap between their identifiability result and ours. For instance, our identifiability result requires that each latent variable has at least two pure children while theirs does not impose such a requirement. More specifically, their identifiability result covers Case 5 and Case 6 shown as Figure 10, which fall outside the scope of our identifiability result. The identifiability result in the work of Adams et al. is superior to ours.
>
> **3. Identification method.** The identification method in Adams et al. first estimates the mixing matrix from the noise terms to the observed variables, and then recovers the causal adjacency matrix from it. Our identification method first sequentially identifies latent variables from leaves to roots, and then sequentially infers causal relations from roots to leaves. Technical details are omitted here due to space limit, we refer interested readers to the original papers.
>
> **4. Discussion on identification method.** Our identification method is a rolling-based method while that proposed by Adams et al. is a static one. Although the latter can work in theory, it is not advisable in practice as acknowledged by Adams et al. themselves. Specifically, the procedure of estimating the mixing matrix requires the number of latent variables as prior knowledge and is computationally intractable, because it is based on overcomplete independent component analysis (OICA). Moreover, the procedure of recovering the causal adjacency matrix from the estimated mixing matrix is rather sensitive to noise. In contrast, our algorithm is not only practical but also efficient. Our identification method is superior to that in the work of Adams et al.
>
> **5. Answers to your questions.** (1) static way: our identifiability result is static while our identification method is rolling-based. The identification method in the work of Adams et al. is static. Although it can work in theory, it is not advisable in practice as acknowledged by Adams et al. themselves. Please refer to point 4 for more details. (2) gap: The identifiability result in the work of Adams et al. is exactly the theoretical maximum identifiability, which is more general than our identifiability result. Please refer to point 2 for more details. As future work, we plan to explore how to relax Assumption 1 while maintaining algorithmic efficiency.

---

> ### Author Response · Authors · 2024-11-26
>
> # Part (2/2)
>
> > Q17: quintuple constraint
>
> First, we agree with you that the quintuple constraint can be considered as a special case of GIN condition, except for one distinction: the former avoids to test independence between the linear combination of a set of variables and all of another set of variables.
>
> Second, we choose quintuple constraint not because it can significantly improve algorithmic efficiency, but because it makes theory more concise. Specifically, with the quintuple constraint, we eliminate the need to prove whether an additional independence relationship holds or not, which simplifies the theoretical analysis.
>
> Third, as mentioned in response to Q16, there exists a gap between our current identifiability result and the theoretical maximum identifiability. Given the flexibility of the GIN condition, it has potential to uncover more information when properly applied. We believe it could lead to an algorithm that substantially narrows this gap while maintaining algorithmic efficiency. This is a promising direction that is beyond the scope of our present work. We will explore this direction further in our future research.
>
> > Q18: Executable code
>
> We have further refined our pseudo-codes in Appendix and also provide an executable code via this anonymous github link: https://anonymous.4open.science/r/Fveds1C055gvGWsdvs345 \
> (Updated: We find that the original link is not very stable, if it does not work for you, please use this backup link: https://anonymous.4open.science/r/Fveds1C055gvGWsdvs751)
>
> Thanks again for your active participation in discussion.
>
> Sincerely,\
> Authors

---

> ### Comment · Reviewer_SqCS · 2024-11-27
>
> Thank you the authors for providing thorough answers to my questions. I know they are a lot. My major concerns have been addressed. I have raised my score accordingly. Thank you.

---

> > ### Author Response · Authors · 2024-11-27
> >
> > Dear Reviewer SqCS,
> >
> > We sincerely thank you for your patience and valuable suggestions, which have significantly improved the quality of our manuscript. We are pleased to have addressed your concerns.
> >
> > Sincerely,\
> > Authors

---

### Official Review · Reviewer_d5Lc · 2024-11-04

**Soundness:** 3
**Presentation:** 3
**Contribution:** 2
**Rating:** 6
**Confidence:** 4

**Summary:**

This paper proposes an efficient and trustworthy causal discovery method for discovering latent variable structure. The main difference compared with the previous work is that it will raise an error rather than draw an incorrect causal conclusion when the purity assumption is not met.

**Strengths:**

- Some theoretical results are proposed along with an efficient causal discovery algorithm.
- The overall writing is clear and well-structured.

**Weaknesses:**

- Although this paper allows violating the purity assumption, the overall identification results and the algorithm mostly rely on the purity assumption, e.g., by locating the pure children like the previous work (e.g., Jin et al., 2024). Whether the purity for identifying the causal structure necessary? What is the complete identifiability under the impurity setting?
- What is the intuition in obtaining the efficient causal discovery compared with the other works, e.g., which step is the key step in providing a faster discovering ability?

**Questions:**

See the weaknesses above.

---

> ### Author Response · Authors · 2024-11-18
>
> # Part (1/2)
>
> We are grateful for your valuable comments. We revise our manuscript (where major revisions are marked in purple) and address your concerns in the following response.
>
> > Q1: The purity assumption.
>
> As explained in the first paragraph of Introduction, the purity assumption means that there is no edge between observed variables. The proof of **none** of our theoretical results utilizes this assumption. $\mathcal{G} _0$ shown as Figure 2(a) violates the purity assumption, but our algorithm can correctly identify it.
>
> We would like to highlight the difference between the purity assumption and the pure children assumption. While the purity assumption is not employed throughout, our identification results in Section 3 relies on the pure children assumption, which is exactly Assumption 1 in our paper. Following previous works[1, 2, 3], we make the pure children assumption such that we can identify latent variables and infers their causal relations through their pure children.
>
> Also, we investigates the scenarios where the pure children assumption is invalid in Section 4. We prove that our algorithm can raise an error rather than return a wrong result in such scenarios, ensuring trustworthiness. This trustworthy mechanism marks both a novel and crucial contribution to the field, as no previous work in causal discovery with latent variables has demonstrated such capability. The ability to systematically identify invalid assumptions, rather than silently producing potentially misleading results, represents a significant step forward in ensuring the reliability and validity of causal discovery methods
>
> **Reference**
>
> [1] Ruichu Cai, et al. "Triad constraints for learning causal structure of latent variables." NeurIPS 2019.
>
> [2] Feng Xie et al. "Generalized independent noise condition for estimating latent variable causal graphs." NeurIPS 2020.
>
> [3] Songyao Jin et al. "Structural estimation of partially observed linear non-gaussian acyclic model: A practical approach with identifiability." ICLR 2024.
>
>
> > Q2: Identification results under the impurity setting.
>
> We have added Section 3.3 and 4.3 to clearly characterize what results our algorithm can deliver under different sets of assumptions.
> - Theorem 7 in Section 3.3 states that "Suppose the observed variables are generated by a LiNGAM with latent variables satisfying the rank-faithfulness assumption and Assumption 1, in the limit of infinite data, our algorithm correctly identifies the underlying complete causal structure."
> - Theorem 13 in Section 4.3 states that "Suppose the observed variables are generated by a LiNGAM with latent variables satisfying the rank-faithfulness assumption and Assumption 2, if Assumption 1 is invalid, in the limit of infinite data, our algorithm raises an error."

---

> ### Author Response · Authors · 2024-11-18
>
> # Part (2/2)
>
> > Q3: Intuition in obtaining efficiency
>
> In this paper, we claim our algorithm is efficient because the only existing algorithm capable of handling complex causal relations, PO-LiNGAM proposed by Jin et al. 2024, has exponential time complexity with the number of variables while ours has only cubic time complexity. We intuitively explain why our algorithm is more efficient than theirs in the following. This part can also be found in line 514~520 in our paper.
>
> As mentioned in Introduction, PO-LiNGAM alternates between inferring causal relations and identifying latent variables from leaves to roots, whereas ours first identifies latent variables from leaves to roots and then infers causal relations from roots to leaves. The efficiency gap arises from distinct approaches for inferring causal relations. While we have provided an illustrative example in our paper, here we present a more typical case for reference.
>
> Consider a causal graph $\mathcal{G} _0$ with latent variables $\mathbf{L} = \emptyset$ and observed variables $\mathbf{O} _0 = \\{ O _1,..., O _n \\}$, where $O _1$ and $O _n$ are respectively the common parent and common child of $O _2, ..., O _{n-1}$, and there is no causal relation among $O _2, ..., O _{n-1}$.
> - PO-LiNGAM first identifies $O _n$ as a leaf node by finding a subset $\mathbf{P} \subset \mathbf{O} _0 \backslash \\{ O_n \\}$ s.t. a particular linear combination of $\mathbf{P} \cup \\{O_n\\}$ is independent of $\mathbf{O} _0 \backslash \\{ O_n \\}$, where $\mathbf{P}$ is just the parents of $O _n$. Clearly, PO-LiNGAM needs to traverse the power set of $\mathbf{O} _0 \backslash \\{O _n\\}$.
> - Assigning each observed variable $O _i \in \mathbf{O} _0$ with two surrogates $X _{2i-1}$ and $X _{2i}$, our algorithm first identifies $O _1$ as a root node because for any $O _i \in \mathbf{O} _0 \backslash \\{ O_1 \\}$, $\mathrm{R}(X _{2i-1}, X _1 | X _2) \perp X _2$. Clearly, our algorithm only needs to traverse $\mathbf{O} _0 \backslash \\{O _1\\}$ itself.
>
> Furthermore, we would like to explain why PO-LiNGAM cannot infer causal relations from roots to leaves while we can. PO-LiNGAM alternates between inferring causal relations and identifying latent variables, that is, when it infers causal relations, there is no guarantee that the root variable has been identified, so it cannot infers causal relations from roots to leaves. In contrast, Theorem 4 guarantees that our algorithm identifies all latent variables in stage 1, so when inferring causal relations in stage 2, it can do this from roots to leaves.

---

> > ### Comment · Reviewer_d5Lc · 2024-11-25
> >
> > Thanks for the response, which addresses my concerns. I raise my score accordingly.

---

> > > ### Author Response · Authors · 2024-11-25
> > >
> > > We deeply appreciate your prompt reply! It has been our pleasure to address your concerns.

---

> ### Author Response · Authors · 2024-11-25
> **Further Discussion**
>
> Dear Reviewer d5Lc:
>
> We really appreciate your constructive opinions that helped us improve this paper. If there is any concern unresolved, we would be glad to have further discussions.
>
> Thanks again for your time, looking forward to hearing from you soon.
>
> Best,
>
> Authors.

---

### Official Review · Reviewer_BQvB · 2024-11-04

**Soundness:** 3
**Presentation:** 1
**Contribution:** 2
**Rating:** 6
**Confidence:** 3

**Summary:**

The authors consider the problem of causal discovery in latent variable LiNGAM models under less restrictive graphical assumptions (described in Assumption 1). Specifically, they allow for some observed variables to be connected to each other and some latent variables to be children of observed variables. They propose a two-stage recovery algorithm that primarily relies on the properties of pseudo-residuals in linear non-Gaussian models. Finally, they consider the case where the graphical assumptions are violated and evaluate the performance of the algorithm through simulations.

**Strengths:**

The authors provide a detailed theoretical analysis of the proposed algorithm, along with illustrative examples. They also include detailed comparisons of their assumptions and conditions with those of existing work.

**Weaknesses:**

The main weakness lies in the presentation of the results. Some definitions and concepts are not sufficiently motivated or explained (see Q1 and Q2 below), making certain technical details difficult to understand. Additionally, it would be helpful to include more detailed descriptions of the assumptions and theorems. For instance, in Algorithm 1, Theorem 2 appears to be used solely for partitioning $\mathbb{S}$ into $\mathbb{S}_1$, $\mathbb{S}_2$ and $\mathbb{S}_3$, rather than adding new IPs to $\mathbb{S}$. It would be helpful if this kind of explanation (or a very sketched version of Alg 1 & 2) is provided.

**Questions:**

1. What is the exact definition of neighbor? The examples on line 179 rely on the fact that $\text{Ne}^{\mathcal{H}_2}(O_1)\setminus{O_3}=\emptyset$. This indicates that $\text{Ne}^{\mathcal{H}_2}(O_1)$ does not correspond to the sibling set (which is $L_1$) nor the neighbor set in the undirected graph (which is $(O_2, O_3, O_4, O_5)$).

2. How are the augmented nodes ($O'$, $O''$) used in the identification results and algorithms?
My understanding is that they are only used in the algorithm, where the algorithm duplicates each observed variable with two extra copies and add non-Gaussian noises to them.

3. Are there any identification guarantees for the recovery output? Specifically, does Theorems 1-6 imply that in linear non-Gaussian model, under Assumption 1 and rank-faithfulness, the underlying model can be uniquely identified? Similarly, doesn Theorems 7-11 imply that if Assumption 1 is not satisfied, then there will be no output?

---

> ### Author Response · Authors · 2024-11-18
>
> # Part (1/2)
>
> Thank you for your time and effort put into our work. We have substantially improved presentation of our manuscript (where major revisions are marked in purple) and also addressed your concerns as follows. We will be grateful if you can re-evaluate our work.
>
> > Q1: Motivation or explanation of concepts
>
> We have provided clearer motivation and explanation for key concepts in the revised manuscript. In the following, we give two representative examples.
>
> - For augmented nodes $\mathbf{O}_ 1$, we have explained the motivation behind it. In brief, we introduce $\mathbf{O}_ 1$ to streamline technical details in our paper and keep the presentation focused on the core ideas. According to the footnote 1 in the revised manuscript. "While the values of observed variables are directly accessible for causal discovery, the causal relations of latent variables can only be inferred indirectly, e.g., through their pure children. By introducing $\mathbf{O}_ 1$ to create pure children for each observed variable, we can uniformly handle both types of variables through analyzing their pure children, thereby eliminating the need to repeatedly distinguish between treatments of latent and observed variables and keeping the core methodology clear."
>
>
> - We fully understand your concerns about the difficulty in understanding the concepts of $\mathbf{V} _f$ and $\mathcal{H} _2$, as many related definitions were concentrated in Chapter 3 without sufficient explanation. To address this issue, we provide both intuitive explanations and illustrative examples. For $\mathbf{V} _f$ and $\mathcal{H} _2$, which are defined as $\mathbf{V} \backslash (\mathbf{V} _c \cup \mathbf{V} _p)$ and the induced subgraph of $\mathcal{G}$ over $\mathbf{V} _c \cup \mathbf{V} _f$ in § Initialization in Section 3.1, we have provided the intuitive explanation immediately after defining them. "Intuitively, while $\mathbf{V} _c \cup \mathbf{V} _p$ consists of all identified variables, $\mathbf{V} _f$ consists of all unidentified variables. While $\mathcal{H} _1$ consists of all identified causal relations, $\mathcal{H} _2$ consists of all unidentified causal relations. Considering the initial case (Initially, we have identified no latent variable and no causal relation in $\mathcal{G} _0$, when $\mathbf{V} _f = \mathbf{L}$ and $\mathcal{H} _2 = \mathcal{G} _0$), such intuitions become particularly apparent." Moreover, we include more illustrative figures (right subfigures of Figures 3, 4, 5) to display $\mathbf{V} _f, \mathcal{H} _2$ at different iteration during stage 1.
>
> > Q2: (1) Definition of neighbor (2) Whether $\\{ O_1, O_3 \\} \in \mathbb{S}_1$ relies on $\mathrm{Ne} ^{\mathcal{H} _2} (O _1) \backslash \\{ O _3 \\} = \emptyset$
>
> (1) Our definition of neighbor adheres to the standard terminology in graph theory. Specifically, $X$ is a neighbor of $Y$ iff there exists an edge $X \to Y$ or $Y \to X$. For instance, in the initial $\mathcal{H}_ 2$ shown on the right of Figure 3, $\mathrm{Ne} ^{\mathcal{H}_ 2} (O_ 1) = \\{ O_ 2, O_ 3, O_ 4, O_ 5, L_ 2\\}$.
>
> (2) Actually, $\\{ O_1, O_3 \\} \in \mathbb{S}_ 1$ does not rely on $\mathrm{Ne} ^{\mathcal{H}_ 2} (O_ 1) \backslash \\{ O_ 3 \\} = \emptyset$. In contrast, it relies on $\mathrm{Ne} ^{\mathcal{H}_ 2} (O_ 1) \backslash \\{ O_ 3 \\} \neq \emptyset$. According to our Definition 2(1), "..., $\mathrm{Ne}^ {\mathcal{H}_ 2} (V_1) \backslash \\{V_ 2\\} \neq \emptyset$, we denote this by $\\{V_1, V_2\\} \in \mathbb{S}_ 1$, ...". Please note that there is a $\neq$, not a $=$.
>
> > Q3: How $\mathbf{O} _1$ is applied in identification results
>
> In response to Q1, we have introduced the motivation of the augmented nodes $\mathbf{O} _1$. Here we use Theorem 2 as an example to illustrate how $\mathbf{O} _1$ is applied in identification results.
>
> According to § Initialization in Section 3.1, $\mathbf{V} _c$ is initialized as $\mathbf{O} _0$, $\mathbf{V} _p$ is initialized as $\mathbf{O} _1$, and variables in $\mathbf{O} _1$ are children of variables in $\mathbf{O} _0$ in the initialized $\mathcal{H} _1$. According to Theorem 2, given $\\{V_i, V_j\\} \in \mathbb{S}$, we need to leverage $\\{V _{i _1}, V _{i _2}\\} \subset \mathrm{Ch} ^{\mathcal{H} _1}(V _i)$ to determine whether $\\{V_i, V_j\\} \in \mathbb{S} _1$ and whether $\\{V_i, V_j\\} \in \mathbb{S} _3$. Clearly, at the first iteration, $V _i \in \mathbf{V} _c = \mathbf{O} _0$ and $\\{V _{i _1}, V _{i _2}\\} \subset \mathbf{V} _p = \mathbf{O} _1$.

---

> ### Author Response · Authors · 2024-11-18
>
> # Part (2/2)
>
> > Q4: More explanations or algorithm overview
>
> We totally agree that more explanations are helpful. We have added illustrative examples for assumptions and expanded Remarks of theoretical results. For example,
> - We have provided an example for Assumption 1 "The graph in Figure 2(a) satisfies this assumption, where $\mathrm{PCh}^{\mathcal{G}_0}(L_1) = \\{L_3, L_4\\}$ and $\mathrm{PCh}^{\mathcal{G}_0}(L_1) = \\{L_2, L_3, L_4, O_2, O_6\\}$."
> - We have added the content "This theorem provides a method for locating identifiable pairs from $\mathbf{V}_ c$ via statistical analysis" to Remark of Theorem 1, which helps readers quickly grasp the practical implication of this theorem.
> - We have added the content "This theorem provides a method to divide $\mathbb{S}$ into $\mathbb{S}_ 1, \mathbb{S}_ 2, \mathbb{S}_ 3$ via statistical analysis, that is, we can locate pure children from identifiable pairs." to Remark of Theorem 2, which explicitly specifies what task this theorem serves.
>
> Also, we have added an overview version of our algorithm into the main text (Algorithms 1 and 2), where we have explicitly linked each step to its corresponding theorem. The detailed version (Algorithms 3 and 4) is deferred to Appendix.
>
>
> > Q5: Identification guarantees
>
> We have added Section 3.3 and 4.3 to clearly characterize what results our algorithm can deliver under different sets of assumptions.
> - Theorem 7 in Section 3.3 states that "Suppose the observed variables are generated by a LiNGAM with latent variables satisfying the rank-faithfulness assumption and Assumption 1, in the limit of infinite data, our algorithm correctly identifies the underlying complete causal structure."
> - Theorem 13 in Section 4.3 states that "Suppose the observed variables are generated by a LiNGAM with latent variables satisfying the rank-faithfulness assumption and Assumption 2, if Assumption 1 is invalid, in the limit of infinite data, our algorithm raises an error."

---

> ### Author Response · Authors · 2024-11-25
> **Further Discussion**
>
> Dear Reviewer BQvB:
>
> We want to express our appreciation for your valuable suggestions, which greatly helped us improve the quality of this paper. We have taken our maximum effort to address your concerns on clarification. Could you please kindly re-evaluate our work?
>
> Your further opinions are very important for evaluating our revised paper and we are hoping to hear from you. Thank you so much.
>
> Best,
>
> Authors.

---

> > ### Comment · Reviewer_BQvB · 2024-11-26
> >
> > Sorry for the late follow-up. I thank the authors for the response, and would like to increase my score from 5 to 6. Three minor questions:
> > 1. I noticed that the authors introduced $V'_i$ and $V''_i$ in Theorems 2 & 3 of the updated draft. My understanding is that these notations do not share the same meaning as $O'_i$ and $O''_i$ (i.e., $\mathbf{O}_1$). If this interpretation is correct, I would suggest that the authors either use different notations or clarify the distinction more explicitly.
> > 2. It seems that samples of variables in $\mathbf{O}_1$ need to be synthetically generated in practice, as they are used in the Algorithm. Could the authors provide additional details on how the noise distributions and noise variances are selected?
> > 3. Is rank-faithfulness assumption equivalent to the bottleneck faithfulness assumption in (Adams et al., 2021)?

---

> > > ### Author Response · Authors · 2024-11-26
> > >
> > > Dear Reviewer BQvB,
> > >
> > > Thanks for your kind response, we are glad to answer your remaining questions.
> > >
> > > > Q6: $V_i'$ and $V_i''$ in Theorem 2&3 of the revised manuscript.
> > >
> > > We made this modification as it was requested by Reviewer SqCS. We apologize for causing you extra confusion, we have replaced $\\{V_ i', V_ i''\\}$ with $\\{V_ {i'}, V_ {i''}\\}$ in the updated manuscript.
> > >
> > > > Q7: Details about $\mathbf{O}_ 1$.
> > >
> > > First, the noises used to create $\mathbf{O}_ 1$ can be mutually independent random variables with any non-Gaussian distribution and any variance. For instance, they can be mutually independent random variables that all follow uniform distribution between [-1, 1].
> > >
> > > Second, as stated in response to your Q1, where we state the motivation behind $\mathbf{O}_ 1$, "we introduce $\mathbf{O}_ 1$ purely to streamline technical details in our paper and keep the presentation focused on the core ideas", and "the values of observed variables are directly accessible for causal discovery". Therefore, in actual implementation, we can use the values of $\mathbf{O}_ 0$ directly. More specifically, whenever we need to use $O_ i'$ or $O_ i''$, we can directly replace it with $O_ i$.
> > >
> > > > Q8: Connection between rank-faithfulness and bottleneck faithfulness.
> > >
> > > First, according to their respective definitions, rank-faithfulness implies bottleneck faithfulness, but bottleneck faithfulness may not imply rank-faithfulness.
> > >
> > > Second, in our paper, rather than working directly with the rank-faithfulness itself, we derive and utilize two properties that follow from it, which are stated in Intuition of Assumption 3 in App. C.
> > > - $m_ {ij} \neq 0$ iff $V_ j \in \mathrm{GAn}(V_ i)$.
> > > - Suppose $m_ {ik} m_ {jk} m_ {il} m_ {jl} \neq 0$, $m_ {ik} / m_ {jk} \neq m_ {il} / m_ {jl}$ iff there exists two non-intersecting paths from $\{V_ k, V_ l\}$ to $\{V_ i, V_ j\}$.
> > >
> > > These two properties can also be derived from bottleneck faithfulness. In other words, although rank-faithfulness is not strictly equivalent to bottleneck faithfulness, we can readily replace rank-faithfulness with bottleneck faithfulness in our work.
> > >
> > > Thanks again for your time and labor in helping us improve our manuscript.
> > >
> > > Sincerely,\
> > > Authors

---

> > > > ### Comment · Reviewer_BQvB · 2024-11-27
> > > >
> > > > I thank the author for the further clarification.
> > > > > Regarding $\mathbf{O}_1$
> > > >
> > > > My understanding is that variables in $\mathbf{O}_1$ are always needed in Algorithm 3, at least in the first iteration. Please correct me if I am wrong.
> > > >
> > > > Do you imply that here, by selecting $V_{i_1}$ and $V_{i_2}$ exactly the same as $V_{i}$, lines 13-16 and 19-25 in Algorithm 3 are not affected? If this is the case then I believe it is worth calling out.
> > > >
> > > > > Rank-faithfulness assumption
> > > >
> > > > The properties here correspond to the case when $|J|, |K| \leq 2$ in bottleneck faithfulness assumption. If these are the only two properties needed in the proof, then I would encourage the authors to explicitly state this connection.

---

> ### Author Response · Authors · 2024-11-27
>
> Dear Reviewer BQvB,
>
> Thanks for your interest in our work. We are pleased to provide further clarification.
>
> > Q9: Regarding $\mathbf{O} _1$
>
> First, you are right that $\mathbf{O} _1$ are always needed at the first iteration of Algorithm 3.
>
> Second, for any $O_ j \in \mathbf{O}_ 0$, denote its two created children by $\\{ O'_ j, O''_ j\\} \subset \mathbf{O}_ 1$, we do imply that if we let both $O'_ j$ and $O''_ j$ be identical to $O_ j$, line 13-16 and 19-25 in Algorithm 3 are not affected. Specifically, the validity of lines 13-16 relies on Theorem 2(1) while the validity of lines 19-25 relies on Theorem 2(3). In Page 22, we have added Remark after the proof of Theorem 2, which explicitly demonstrates that the proof remains valid if $V_ {i _ 1}$ and $V_ {i_ 2}$ are both identical to $V_ i$. It is clear that when $V_ i$ refers to $O_ j \in \mathbf{O}_ 0$, $\\{ V_ {i _ 1}, V_ {i _ 2} \\}$ exactly refers to $\\{ O'_ j, O''_ j\\}$. Therefore, we can actually create $O'_j$ and $O''_j$ by making two copies of $O_j$
>
> > Q10: Rank-faithfulness assumption
>
> Thanks for this suggestion. In Page 16, we have added Remark after Intuition of the rank faithfulness, which explicitly demonstrates that we only utilize these two properties rather than work directly with the rank faithfulness assumption. These two properties can also be derived from the bottleneck faithfulness assumption, so we can replace the rank faithfulness assumption with the bottleneck faithfulness assumption in our work.
>
> Thanks again for your careful reading.
>
> Sincerely,\
> Authors

---

### Official Review · Reviewer_qEDV · 2024-11-09

**Soundness:** 4
**Presentation:** 2
**Contribution:** 3
**Rating:** 6
**Confidence:** 4

**Summary:**

This paper is concerned with learning the structure of a graphical model that may contain latent variables. In contrast to some prior art which learns an equivalence class but does not explicitly represent latent variables, this procedure aims to both identify and model the relationships with latent variables. The author employ an additive noise assumption, and provide a simple and intuitive algorithm for discovering the latent variables via tree construction. Under assumptions on the generative structure and som restrictions on the purity of the children, theoretical results are provided which give nice recovery guarantees and are in general quite complete. A small experimental evaluation is provided as well.

**Strengths:**

Overall, I think this paper approaches a challenging problem and provides a nice and elegant solution. The authors do a nice job of making the necessary assumption clear and carefully presenting their results. Though I have some reservations (see below), overall I consider the technical quality of this work to be quite good. Further, the problem it addresses in loosening the structural assumptions required for provable recovery. This is especially important, in my view, since these structural assumptions are unverifiable and can have an arbitrarily bad impact on the learnt structure under violations of those assumptionsl.

**Weaknesses:**

I found the writing in the introduction to be very hard to parse, in particular the task definition. The authors are focused on the problem of learning the structure of graph including latent variables under additive noise constraints. The description of the work in the introduction makes these points hard to follow, it also isn't immediately clear from the initial description the delineation between this and FCI and learning of other structures such as ADMGs, which allow for the presence of latent variables without explicitly representing them. It would be very useful to more clearly describe the problem.

Most of the paper reads as a step by step walk through of the proof techniques. While this is interesting and useful, it limits the potential audience for the paper, and in many areas, obscures some of the underlying machinery. In my view, much of this should be moved to the supplement. For example, there's no central place in the text which walks through the algorithm as a whole, rather it is strewn out across the paper in analysis steps.

Many of the proofs in the supplement lack the necessary text to provide sufficient context and intuition for the result. For example, it was not immediately clear to me why proposition 1 in App.C.2.1 implied the statement given in the remark of the main text until working it through. The results should be presented with sufficient detail such that they are reasonably easy to ingest and contextualize within the context of the paper.

Experimental evidence is small and limited to very simple settings.

**Questions:**

Please see weaknesses above.

---

> ### Author Response · Authors · 2024-11-18
>
> # Part (1/2)
>
> Thank you for your time and effort reviewing our paper. We appreciate the thoughtful feedback. We revise our manuscript (where major revisions are marked in purple) and address your concerns point by point in the following.
>
> > Q1: Task definition
>
>
> Thanks for your valuable suggestion, we agree that clearly defining our task and highlighting its distinctions from prior work in Introduction helps readers immediately grasp our paper's unique positioning and contributions. This upfront clarity prevents potential confusion and allows readers to better follow our technical approach and appreciate our results in the proper context.
>
> First, we have added the task definition into Introduction. "Given observational data generated by a linear non-Gaussian acyclic model (LiNGAM) with latent variables, we aim to correctly identify the underlying complete causal structure, which is a directed acyclic graph (DAG) that explicitly represents both observed and latent variables along with their causal relations, in an important and challenging setting where latent and observed variables are interconnected through complex causal relations, where ``complex" means that none of the above three assumptions (measurement, purity, and non-triangle assumptions) is employed."
>
> Second, we have also clearly distinguished our work from prior studies on causal discovery with latent variables.
> - Although some previous works such as FCI allows the presence of latent variables, their results such as partial ancestral graphs (PAGs) and acyclic directed mixed graphs (ADMGs) are not informative of the number of latent variables and their causal relations. By utilizing linear models, some recent works can represent latent variables and their causal relations explicitly in their results, which is of significant importance in some fields such as psychology. For instance, responses to psychometric questionnaires (observed variables) are usually thought of as noisy views of various traits (latent variables), and the researcher is predominately interested in the causal relations between the latter. In this regard, our work aligns more closely with the latter works.
>
> - To identify latent variables and infer their causal relations, recent works often assume the absence of certain special causal relations to ensure a degree of simplicity, including the purity (there is no edge between observed variables), measurement (no observed variable is a parent of any latent variable), or no-triangle assumptions (there exists no three mutually adjacent variables). Unfortunately, these assumptions are invalid in many real-world scenarios, an example in business contexts where none of these three assumptions hold is provided in our paper. Our work can handle the case with complex causal relations, where none of these three assumptions is employed.
>
> > Q2: Algorithm overview
>
> We totally agree that adding an algorithm overview in the main text can help readers grasp the big picture. We have moved many technical details such as the proof sketch from the main text to Appendix. With the space saved, we have added an overview version of our algorithm into the main text (Algorithms 1 and 2), where we have explicitly linked each step to its corresponding theorem. The detailed version (Algorithms 3 and 4) is deferred to Appendix.

---

> ### Author Response · Authors · 2024-11-18
>
> # Part (2/2)
> > Q3: Explanations for theoretical results in Appendix
>
> To improve readibility, we have added explanations for intermediate theoretical results presented in Appendix. Since some theoretical results (such as Lemma 1 and Corollary 1) are necessarily detailed and techinical in their formal statements, the readers might get lost in the techincal details. To help readers grasp the key implications, we have added Remarks for Lemma 1 and Corollary 1 as follows.
>
> - Remark of Lemma 1. "(1) provides a sufficient condition for independence involving the pseudo-residual to hold while (2, 3) provides two sufficient conditions for independence involving the pseudo-residual to not hold."
>
> - Remark of Corollary 1. "This corollary reveals the properties of variables in $\mathbf{V}_p$, $\mathbf{V}_f$, and $\mathbf{V}_c$. (1) means that for each variable in $\mathbf{V}_p$, its parents and children in the underlying causal graph $\mathcal{G}$ are exactly its parents and children in $\mathcal{H}_1$. (2) means that for each variable in $\mathbf{V}_f$, its parents and children in the underlying causal graph $\mathcal{G}$ are exactly its parents and children in $\mathcal{H}_2$. (3) means that for each variable in $\mathbf{V}_c$, its children in the underlying causal graph $\mathcal{G}$ are the union of its children in $\mathcal{H}_1$ and its children in $\mathcal{H}_2$ while its parents in $\mathcal{G}$ are exactly its parents in $\mathcal{H}_2$. This corollary is widely used in the following proofs. To maintain fluency, we will use it without further citation."
>
> Also, to help readers better connect the theoretical results in Appendix with the main text, e.g., how Proposition 1 helps to reduce computational cost, we have added Remarks of Proposition 1 as follows.
>
> - Remark of Proposition 1 states that "Given $\\{V_ i, V_ j\\} \subset \mathbf{V}_ c$, denote $\\{V \in \mathbf{V}_ c \backslash \\{V_ i, V_ j\\} | \mathrm{Cov}(V_ i, V_ j) \mathrm{Cov}(V, V_ i) \mathrm{Cov}(V, V_ j) \neq 0\\}$ by $\mathbf{V}_ {ij}$, this proposition means that there exists no $\\{V_k, V_l\\} \subset \mathbf{V}_ {ij}$ s.t. $\mathrm{R}(V_i, V_j | V_k) \perp \mathbf{V}_ c \backslash \\{V_i, V_j\\}$ and $\mathrm{R}(V_i, V_j | V_l) \not \perp \mathbf{V}_ c \backslash \\{V_i, V_j\\}$. Therefore, if we want to know whether for each $V \in \mathbf{V}_ {ij}$, $\mathrm{R}(V_i, V_j | V) \perp \mathbf{V}_ c \backslash \\{V_i, V_j\\}$, we only need to consider any single $V_k \in \mathbf{V}_ {ij}$."
>
> We appreciate this valuable suggestion and believe our added Remarks can significantly enhance readability.
>
> > Q4: Experimental evidence
>
> Similar to most works in this line, the preliminary contribution of our work is theoretical, where the experiments serve as a proof-of-concept of how the algorithm derived from our theoretical results performs in applications. While the existing experiments have demonstrated the effectiveness of our algorithm, we acknowledge that more comprehensive experiments would be beneficial. Therefore, given the observational data generated by $\mathcal{G} _0$ shown as Figure 2(a) consisting of 4 latent variables, 16 observed variables, and many complex causal relations, the experimental results are summarized as follows. With sufficient (10k) samples, our algorithm can achieve the best Error in latent Variables and F1-Score, its Correct-Ordering Rate is only slightly lower than that of PO-LiNGAM, and it is far more efficient than PO-LiNGAM.
>
>
> | | Error in latent Variables | | | Correct-Ordering Rate | | | F1-Score | | | Running Time | | |
> |-|-|-|-|-|-|-|-|-|-|-|-|-|
> | | 2k | 5k | 10k | 2k | 5k | 10k | 2k | 5k | 10k | 2k | 5k | 10k |
> | GIN | 0.8±0.4 | 0.9±0.3 | 0.8±0.4 | 0.15±0.02 | 0.14±0.03 | 0.14±0.02 | 0.36±0.07 | 0.39±0.06 | 0.38±0.04 | **5.94±1.05** | **6.52±0.70** | **7.24±0.82** |
> | LaHME | 3.4±0.5 | 2.8±0.7 | 2.2±1.0 | 0.28±0.02 | 0.27±0.02 | 0.28±0.04 | 0.45±0.04 | 0.46±0.07 | 0.47±0.07 | 7.72±1.00 | 8.38±0.81 | 10.12±1.05 |
> | PO-LiNGAM | **1.0±0.6** | **0.5±0.7** | **0.2±0.4** | **0.83±0.12** | **0.85±0.23** | **0.96±0.08** | 0.42±0.09 | 0.61±0.19 | 0.68±0.10 | 661.25±214.86 | 768.70±261.47 | 925.68±325.07 |
> | Ours | 1.2±0.7 | 0.7±0.6 | **0.2±0.4** | 0.72±0.12 | 0.78±0.17 | 0.93±0.12 | **0.58±0.14** | **0.75±0.13** | **0.94±0.11** | 16.80±1.31 | 20.65±3.89 | 24.55±1.28 |

---

> ### Author Response · Authors · 2024-11-25
> **Further Discussion**
>
> Dear Reviewer qEDV,
>
> We really appreciate your efforts to help improve this paper. We have carefully addressed your concerns. It is really important to us that you could kindly read our rebuttal and provide further questions if there are any.
>
> Thank you so much and hope you have a good day.
>
> Best,
>
> Authors.

---

> ### Author Response · Authors · 2024-12-02
>
> Dear Reviewer qEDV:
>
> Thanks again for your valuable time and effort in reviewing our paper. As the discussion period approaches its end, we would like to confirm whether our rebuttal has adequately addressed all your concerns. If you have any remaining question or require further clarification, we are glad to provide explanations. Also, we would greatly appreciate any suggestions you might have for further improving the quality and presentation of our paper.
>
> Sincerely,\
> Authors

---

### Author Response · Authors · 2024-11-21
**Further Discussion**

We sincerely thank all reviewers for their insightful and valuable feedback on our manuscript. Their constructive comments have helped us significantly improve the quality of our work. We have carefully addressed each concern raised and made substantial improvements to the manuscript. The major revisions are summarized below.

1. We have clearly described our task in Introduction. As an example, we discuss this improvement in detail in our response to Q1 of Reviewer qEDV. This upfront clarity prevents potential confusion and allows readers to better follow our technical approach and appreciate our results in the proper context.

2. We have provided detailed motivations, intuitive explanations, and illustrative examples for key concepts. As an example, we discuss this improvement in detail in our response to Q1 of Reviewer BQvB. These additions enhance the paper's accessibility and help readers better understand the underlying principles.

3. We have provided more detailed interpretations and discussions for both theoretical results in the main text and those in Appendix. As examples, we discuss this improvement in detail in our response to Q3 of Reviewer qEDV and Q4 of Reviewer BQvB. This comprehensive revision helps readers better understand the theoretical contributions of our work.

4. We have added an overview version of our proposed algorithm in the main text and also a detailed version in Appendix. As examples, we discuss this improvement in detail in our response to Q1 and Q2 of Reviewer SqCS. This two-tier structure allows readers to grasp the core idea while having access to the full technical depth of our work.

5. We have added Section 3.3 and 4.3 to clearly characterize what results our algorithm can deliver under different sets of assumptions. As an example, we discuss this improvement in detail in our response to Q5 of Reviewer BQvB.

We welcome any additional feedback that would further enhance our manuscript and remain committed to addressing any remaining concerns to ensure the highest quality of our research.

Thank you again for your time and expertise in reviewing our work.

---

### Author Response · Authors · 2024-12-02
**Rebuttal Summary**

Dear reviewers, AC, and SAC:

We sincerely thank all reviewers for taking their valuable time to read our paper, provide constructive suggestions, and actively engage in discussions. We are also very grateful to AC and SAC for organizing high-quality review process. In the last couple of weeks, we have tried our best to address each concern of all reviewers and integrated these changes into our revised manuscript. The main revisions are summarized as follows.

1. To help readers better grasp the key concepts, we have enriched the manuscript with clear motivations, intuitive explanations, and illustrative examples. For example, we have added clear motivations of $\mathbf{O}_ 1$ in footnote 1, provided intuitive explanations for $\mathbf{V}_ f$ and $\mathcal{H}_ 2$ immediately after their definitions in line 183~185, and included illustrative examples for $\mathbf{V}_ f$ and $\mathcal{H}_ 2$ in Figures 3, 4, 5.

2. To help readers better comprehend the theoretical results, we have enhanced both the main theorems (e.g., Theorems 1, 2) and supporting results in the Appendix (e.g., Lemma 1, Corollary 1, Proposition 1) with detailed interpretations and discussions.

3. To help readers better follow our algorithm, we have provided an algorithm overview in the main text (Algorithms 1, 2), a detailed pseudo-code in Appendix (Algorithms 3, 4), and released our source code through an anonymous link (https://anonymous.4open.science/r/Fveds1C055gvGWsdvs345).

We believe these revisions have substantially improved our paper, and we will continue to refine our paper in the future. Also, we would like to highlight the main contributions of our paper below.

1. We investigate an understudied setting where latent and observed variables are interconnected through complex causal relations.

2. Under a pure children assumption, we develop a series of theoretical results, leading to an efficient algorithm which is the first one capable of handling the setting with both latent variables and complex relations within polynomial time.

3. We prove that our algorithm can raise an error rather than return an incorrect result when the pure children assumption is invalid, ensuring trustworthiness. To the best of our knowledge, no prior work on causal discovery with latent variables has provided such rigorous trustworthiness guarantees.

We sincerely hope our work could contribute to the community and advance the development of causal discovery. Thanks again for your efforts.

Sincerely,\
Submission 215 Authors.

---

### Author Response · Authors · 2024-12-03
**Backup link of our source code**

Dear all,

We find that the original link of our source code (https://anonymous.4open.science/r/Fveds1C055gvGWsdvs345) is not very stable, the webpage may sometime display "The requested file is not found." If it does not work for you, please use this backup link: https://anonymous.4open.science/r/Fveds1C055gvGWsdvs751

Sincerely,\
Authors

---

### Meta-Review · Area_Chair_KLsL · 2024-12-19

**Metareview:**

The paper provides novel ideas to learn the structure of complex linear latent variable models, under the non-Gaussianity assumption. One key point is that it can detect when the necessary sparsity conditions (a sufficient number of "pure children") fail and flag those accordingly. Reviewers were unanimous that the contributions of the paper are of interest, although the density of presentation and ideas needs further work. I believe these are solvable.

One key point though is that the authors seem to misunderstand some of the results in the literature, including Silva et al. (2006). The contribution states that

"We prove trustworthiness of our algorithm, meaning that when the pure children assumption
is invalid, it can raise an error rather than return an incorrect result, thus preventing potential
damage to downstream tasks. To the best of our knowledge, there is a lack of similar results in
the literature of causal discovery with latent variables"

The latter statement, about the lack of results on validity under the failure of pure children assumptions, is not true. As a matter of fact, Silva et al. does not assume the existence of pure children. Quite the opposite, the main idea of that paper exploits the fact that *if* pure children exist in some subset of the model, the proposed algorithm exploits them and returns a corresponding submodel of the causal generative process. It is typically the case that only a strict subset of the true model is returned, and it is often the case that an empty solution is returned (analogous to the PC algorithm not returning any oriented edges) simply because no "pure children" structure exists (arguably a more standard way of reporting causal discovery solutions than raising an "error"). This confusion may be common in the literature, and it would be useful for this paper to clarify it.

**Additional Comments On Reviewer Discussion:**

Most reviewers engaged productively with the authors. It is clear that the paper is dense, as witnessed by the dense set of replies provided by the authors. It is important that the authors recognize that the effort that they put in the replies should be carried out to the paper if this is to be published in ICLR.

---

### Decision · Program_Chairs · 2025-01-22

Accept (Poster)